# Optimal Block-wise Asymmetric Graph Construction for Graph-based Semi-supervised Learning

**Zixing Song**
The Chinese University of Hong Kong
New Territories, Hong Kong SAR
zxsong@cse.cuhk.edu.hk

**Yifei Zhang**
The Chinese University of Hong Kong
New Territories, Hong Kong SAR
yfzhang@cse.cuhk.edu.hk

**Irwin King**
The Chinese University of Hong Kong
New Territories, Hong Kong SAR
king@cse.cuhk.edu.hk

## Abstract

Graph-based semi-supervised learning (GSSL) serves as a powerful tool to model the underlying manifold structures of samples in high-dimensional spaces. It involves two phases: constructing an affinity graph from available data and inferring labels for unlabeled nodes on this graph. While numerous algorithms have been developed for label inference, the crucial graph construction phase has received comparatively less attention, despite its significant influence on the subsequent phase. In this paper, we present an optimal asymmetric graph structure for the label inference phase with theoretical motivations. Unlike existing graph construction methods, we differentiate the distinct roles that labeled nodes and unlabeled nodes could play. Accordingly, we design an efficient block-wise graph learning algorithm with a global convergence guarantee. Other benefits induced by our method, such as enhanced robustness to noisy node features, are explored as well. Finally, we perform extensive experiments on synthetic and real-world datasets to demonstrate its superiority to the state-of-the-art graph construction methods in GSSL.

## 1 Introduction

Graph-based semi-supervised learning (GSSL) is a burgeoning research field [52, 9, 14, 63, 65, 62]. As a subclass of semi-supervised learning (SSL), GSSL exhibits promise since it encapsulates the smoothness or manifold assumption, where samples with similar features are likely to share the same label. GSSL methods begin by constructing an affinity graph, wherein nodes represent samples, and weighted edges denote similarity between pairs of nodes. This process aligns with the manifold assumption, implying that nodes connected by edges with large weights tend to have the same label. Upon obtaining the affinity graph, various label inference algorithms such as label propagation [85, 82, 25, 65] can be executed to predict labels for the unlabeled nodes.

Preliminary empirical studies [16] suggest that the quality of the affinity graph significantly influences label prediction performance. However, the graph construction phase in GSSL has received less scrutiny compared to the subsequent label inference phase. Constructing a high-quality graph presents a challenge as its quality can only be assessed indirectly through postmortem verification via label inference performance. Classic solutions such as the Radial Basis Function (RBF) Kernel [85] , kNN graph [17], and $b$-matching [26], while simple, may exhibit low robustness against noise due to their simplicity. More recent and complex methods, typically framed as optimization problems to find

the optimal graph under the smoothness assumption, suffer from inefficient optimization techniques without fast convergence guarantees [18, 29, 63] or lack a solid theoretical foundation [58, 14, 23].

Consequently, a series of research questions arise: *1) What is the optimal graph structure for label inference algorithms in GSSL? 2) How can we optimize this optimal graph efficiently? 3) What kinds of benefits can this optimal graph bring?*

Most existing GSSL graph construction methods treat all nodes equally, disregarding label information, which results in a symmetric, undirected graph. However, we contend that an asymmetric, directed graph structure might be more beneficial for subsequent label inference due to the distinct roles labeled and unlabeled nodes can play. Intuitively, edges from labeled to unlabeled nodes would naturally facilitate supervision information propagation, while edges from unlabeled to labeled nodes may introduce inconsistency, potentially undermining the prediction accuracy for the labeled nodes.

Motivated by this key observation, we revisit a unified optimization framework that encompasses most label inference algorithms, assuming the affinity graph is already constructed. We then fix the downstream label inference result and position the graph weight matrix (or adjacency matrix) as the optimization variable. This shift allows us to define optimality in the graph construction step of GSSL concisely, addressing our first research question. We subsequently present a tailored optimal solution featuring an asymmetric graph structure that aligns with our proposed intuition. In response, we introduce the **B**lock-wise **A**ffinity **G**raph **L**earning (BAGL) algorithm by leveraging duality and the fast proximal gradient method, addressing our second research question. Finally, we demonstrate that BAGL ensures a sub-linear global convergence rate of $O(1/k)$ and can alleviate issues of noisy node features, addressing our third research question.

In summary, this work offers four main contributions. First, we provide a succinct definition of the optimality of the affinity graph in GSSL, and through rigorous derivation, propose an ad-hoc solution with an asymmetric structure. Second, we design a block-wise graph learning framework, BAGL, to infer the weights in the optimal graph structure. Third, we prove that a global sub-linear convergence rate is guaranteed for BAGL and analyze other benefits. Fourth, we perform extensive experiments on synthetic and real-world datasets to demonstrate the effectiveness and efficiency of BAGL.

## 2 Preliminary

### 2.1 Problem Formulation

We present a formulation for the GSSL problem, comprising two steps: graph construction and label inference [52]. Our study primarily investigates the optimal construction of the affinity graph in the first phase to facilitate enhanced performance in the second label inference phase.

Given a set of data points $\{\boldsymbol{x}_i\}_{i=1}^n$, where each $\boldsymbol{x}_i \in \mathbb{R}^d$ is sampled from a $d$ dimensional feature space. In this paper, we interchangeably use the terms *node*, *point*, and *sample* to refer to $\boldsymbol{x}_i$. Each sample $\boldsymbol{x}_i$ has a label $y_i \in \mathbb{N}_c$, where $\mathbb{N}_c = \{i \in \mathbb{N}^+ \mid 1 \leq i \leq c\}$ with $c$ being the number of classes. Given the labels of the $l$ samples $\{\boldsymbol{x}_i\}_{i=1}^l$ as $\{y_i\}_{i=1}^l$, the ultimate goal of general transductive SSL is to infer the labels $\{y_i\}_{i=l+1}^{l+u}$ for the remaining $u$ unlabeled samples $\{\boldsymbol{x}_i\}_{i=l+1}^{l+u}$ ($n = l + u$). As a subcategory of general SSL, GSSL methods first construct a graph $\mathcal{G} = \{\mathcal{V}, \mathcal{E}, \boldsymbol{W}\}$ based on all the training samples $\{\boldsymbol{x}_i\}_{i=1}^n$ and partially given labels $\{y_i\}_{i=1}^l$. Here, $\mathcal{V}$ is the node set with $|\mathcal{V}| = n$. Each node represents each sample $\boldsymbol{x}_i$. $\mathcal{E}$ is the edge set in which each edge $(i, j)$ is assigned with a weight $W_{ij}$ (the $i$-th row, $j$-th column entry in $\boldsymbol{W} \in \mathbb{R}^{n \times n}$) to reflect the affinity or similarity between the sample pair $(\boldsymbol{x}_i, \boldsymbol{x}_j)$. Generally, a larger weight indicates a higher level of similarity. $W_{ij} = 0$ indicates no edge between node $i$ and node $j$. Therefore, the key challenge in the first graph construction step is to generate $\boldsymbol{W}$ based on $\{\boldsymbol{x}_i\}_{i=1}^n$ and $\{y_i\}_{i=1}^l$ so that the underlying manifold of the data is properly encoded. In the second step, various label inference algorithms can be performed on $\mathcal{G}$ to propagate the given labels $\{y_i\}_{i=1}^l$ and make predictions $\{y_i\}_{i=l+1}^{l+u}$ for unlabeled nodes.

However, the performance of the label inference step is significantly contingent on the quality of the weight matrix $\boldsymbol{W}$ from the first graph construction step. This paper aims to address three critical research questions pertaining to the graph construction step in GSSL. First, what constitutes the overall structure of the optimal weight matrix, $\boldsymbol{W}^*$? (Sect. 3.1). We define the optimal weight matrix, $\boldsymbol{W}^*$, as the one that gives the best prediction results when used with the same label inference method

across different graphs $\boldsymbol{W}$. Second, how to find an efficient method for optimizing the entries in $\boldsymbol{W}^*$? (Sect. 3.2). Third, what kinds of benefits can $\boldsymbol{W}^*$ bring from the theoretical perspective? (Sect. 3.3)

## 2.2 Recap on the Unified Framework for Label Inference Step in GSSL

Before we delve into the proposed structure for the optimal affinity graph, we first revisit the unified framework of the label inference step, allowing for a seamless definition of the optimal affinity graph.

If we assume the affinity graph has been constructed, then numerous influential label inference methods [85, 82, 83, 4, 5, 9] can be performed over this graph to infer the labels for the unlabeled nodes. However, the majority of them can be incorporated into the following framework.

We first define the node feature matrix $\boldsymbol{X} \in \mathbb{R}^{n \times d}$ as $\boldsymbol{X} = [\boldsymbol{x}_1, \cdots, \boldsymbol{x}_n]^\mathsf{T}$ and the predicted soft label matrix $\boldsymbol{F} \in \mathbb{R}^{n \times c}$ as $\boldsymbol{F} = [\boldsymbol{f}_1, \cdots, \boldsymbol{f}_n]^\mathsf{T}$ with $\boldsymbol{f}_i \in \mathbb{R}^c (1 \le i \le n)$. Ground-truth label matrix is given as $\boldsymbol{Y} = [\boldsymbol{y}_1, \cdots, \boldsymbol{y}_n]^\mathsf{T} \in \{0, 1\}^{n \times c}$ with $\boldsymbol{y}_i \in \{0, 1\}^c (1 \le i \le n)$. Here, for the labeled nodes $\{\boldsymbol{x}_i\}_{i=1}^l$, $\boldsymbol{y}_i$ is a one-hot vector in which $Y_{ij} = 1$ if $\boldsymbol{x}_i$ belongs to class $j$ ($y_i = j$), and $Y_{ij} = 0$, otherwise. For the unlabeled nodes $\{\boldsymbol{x}_i\}_{i=l+1}^{l+u}$, $\boldsymbol{y}_i$ is an all-zero vector for initialization.

Driven by the geometry of the affinity graph, we can unify these label inference algorithms as a minimizer of the optimization problem as Problem (1).

$$\boldsymbol{F}^* = \arg\min_{\boldsymbol{F}} Q(\boldsymbol{F}) = \arg\min_{\boldsymbol{F}} \{\mathrm{Tr}\left(\boldsymbol{F}^\mathsf{T}\boldsymbol{S}\boldsymbol{F}\right) + \mathrm{Tr}\left((\boldsymbol{F} - \boldsymbol{Y})^\mathsf{T}\boldsymbol{\Lambda}(\boldsymbol{F} - \boldsymbol{Y})\right). \tag{1}$$

Note that the loss function $Q(\boldsymbol{F})$ consists of a quadratic variation term $\mathrm{Tr}\left(\boldsymbol{F}^\mathsf{T}\boldsymbol{S}\boldsymbol{F}\right)$ as the graph smoothness regularizer, and a quadratic Frobenius error norm $\|\boldsymbol{F} - \boldsymbol{Y}\|_F^2 = \mathrm{Tr}((\boldsymbol{F} - \boldsymbol{Y})^\mathsf{T}(\boldsymbol{F} - \boldsymbol{Y}))$, both of which should ideally be small subject to a trade-off parameter $\boldsymbol{\Lambda}$ between them. Here the smoothing matrix $\boldsymbol{S} = s(\boldsymbol{W}) \in \mathbb{S}_+^n$ in the first term, with $s \colon \mathbb{R}^{n \times n} \to \mathbb{S}_+^{n \times n}$, is positive semidefinite and determined by the weight matrix $\boldsymbol{W}$ of the graph to ensure the adjacent nodes share similar predictions. The second term measures the distance between predicted results and initial assignments, and restricts the output for labeled nodes from deviating too much compared with the ground-truth. $\boldsymbol{\Lambda}$ is a diagonal matrix with $\Lambda_{ii} \ge 0$ and $\boldsymbol{S} + \boldsymbol{\Lambda}$ must be invertible to avoid trivial solutions.

By the first-order optimality condition, we can easily obtain the optimal solution for Problem (1).

$$\boldsymbol{F}^* = (\boldsymbol{S} + \boldsymbol{\Lambda})^{-1}\boldsymbol{\Lambda}\boldsymbol{Y} = (s(\boldsymbol{W}) + \boldsymbol{\Lambda})^{-1}\boldsymbol{\Lambda}\boldsymbol{Y}. \tag{2}$$

Based on the optimal soft label matrix $\boldsymbol{F}^*$, the final predicted label for each node is given as $\hat{y}_i = \hat{y}(\boldsymbol{f}_i) = \arg\max_{1 \le j \le c} F_{ij}^*$. It is worth noting that the unified optimization framework in Problem (1) can admit most of the mainstream GSSL methods. For instance, if we set the smoothing matrix as the normalized graph Laplacian matrix $\boldsymbol{S} = \mathcal{L} = s(\boldsymbol{W}) = \boldsymbol{D}^{-\frac{1}{2}}(\boldsymbol{D} - \boldsymbol{W})\boldsymbol{D}^{-\frac{1}{2}}$ and $\boldsymbol{\Lambda} = \lambda\boldsymbol{I}$, we can easily recover one of the most popular label inference methods for GSSL, Local and Global Consistency (LGC) [82], as $\boldsymbol{F}^* = \arg\min_{\boldsymbol{F}} \{\frac{1}{2}\sum_{i,j=1}^n \left\|\frac{\boldsymbol{f}_i}{\sqrt{D_{ii}}} - \frac{\boldsymbol{f}_j}{\sqrt{D_{jj}}}\right\|_2^2 W_{ij} + \lambda\sum_{i=1}^n \|\boldsymbol{f}_i - \boldsymbol{y}_i\|_2^2\}$. Here, $\mathcal{L} \in \mathbb{S}_+^n$ is defined as $\mathcal{L} = \boldsymbol{D}^{-\frac{1}{2}}\boldsymbol{L}\boldsymbol{D}^{-\frac{1}{2}}$, where the combinatorial Laplacian matrix $\boldsymbol{L} \in \mathbb{S}_+^n$ is given as $\boldsymbol{L} = \boldsymbol{D} - \boldsymbol{W}$, and the degree matrix $\boldsymbol{D} \in \mathbb{R}^{n \times n}$ is a diagonal matrix defined as $D_{ii} = \sum_{j=1}^n W_{ij}$. We summarize several representative works under this unified framework in Table 5 with explicit forms of $\boldsymbol{S}$ (or $s(\boldsymbol{W})$) and $\boldsymbol{\Lambda}$ in Appendix B.2.

# 3 Methodology

## 3.1 Motivation: Optimal Affinity Graph Structure

### 3.1.1 Definition of the Optimality of the Affinity Graph

If we perform the same label inference algorithm from the above-mentioned generalized framework on all possible affinity graphs $\boldsymbol{W}$, the optimal graph $\boldsymbol{W}^*$ will enable the label inference step to obtain the most accurate predictions. Under the above-mentioned unified framework for label inference (Problem (1)), the weight matrix $\boldsymbol{W}$ is fixed while the soft label matrix $\boldsymbol{F}$ is the optimization variable. This is due to our goal of executing label inference to attain the optimal $\boldsymbol{F}^*$ given the affinity graph $\boldsymbol{W}$. Similarly, when we want to construct the optimal graph $\boldsymbol{W}^*$ given the label inference framework,

we consider the weight matrix $\boldsymbol{W}$ as the optimization variable, keeping the soft label matrix $\boldsymbol{F}$ fixed as the solution in Eq. (2). The form of $Q(\cdot)$ remains unchanged as it is related to the generalization bound for GSSL that would be discussed in Appendix D.2. This approach aids in identifying the "optimal" affinity graph for the label inference algorithms under the unified framework.

**Definition 1** (Optimality of the Affinity Graph). *The affinity graph $\mathcal{G}$ is optimal for label inference under the unified GSSL framework if its weight matrix $\boldsymbol{W}$ is the minimizer of Problem (3).*

$$\min_{\boldsymbol{W} \in \mathbb{R}^{n \times n}} \left\{ \mathrm{Tr}(\boldsymbol{F}^{\mathsf{T}} s(\boldsymbol{W}) \boldsymbol{F}) + \mathrm{Tr}\left((\boldsymbol{F} - \boldsymbol{Y})^{\mathsf{T}} \boldsymbol{\Lambda}(\boldsymbol{F} - \boldsymbol{Y})\right) \right\} \quad s.t. \quad \boldsymbol{F} = (s(\boldsymbol{W}) + \boldsymbol{\Lambda})^{-1} \boldsymbol{\Lambda} \boldsymbol{Y}, \quad (3)$$

Although the optimization Problem (3) is intractable in general due to the various forms of $s(\boldsymbol{W})$ in the constraint, we can present an ad-hoc optimal structure of $\boldsymbol{W}^*$ that is independent of $s(\boldsymbol{W})$, which motivates our proposed method in Sect. 3.2 from a theoretical perspective.

### 3.1.2 Structure of the Optimal Affinity Graph

We then present an equivalent proposition in Theorem 1, which provides a necessary and sufficient condition for the optimality of $\boldsymbol{W}^*$ in Problem (3). Motivated by some classic graph sharpening techniques [47, 15, 46], Theorem 1 helps to circumvent the challenges of directly dealing with Problem (3) and holds regardless of the forms of $s(\boldsymbol{W})$ (Appendix E.1).

**Theorem 1.** *$\boldsymbol{W}^*$ obtained by Definition 1 is optimal if and only if $\boldsymbol{Y}_l = \boldsymbol{F}_l$ holds. $\boldsymbol{Y}_l \in \{0, 1\}^{l \times c}$, $\boldsymbol{F}_l \in \mathbb{R}^{l \times c}$ are the ground-truth label matrix, and the soft label matrix for labeled nodes.*

To put it in a simpler way, Theorem 1 tells us that if the affinity graph is optimal, then after we perform the label inference algorithm for GSSL, the predicted soft label for the labeled nodes would coincide with the ground truth precisely. The converse of this observation also holds true. Unfortunately, it remains an open question to solve $\boldsymbol{Y}_l = \boldsymbol{F}_l$ by listing all possible classes of solutions due to the complex interconnection of $\boldsymbol{F}$ and $\boldsymbol{W}$. However, we can provide a simple ad-hoc solution for $\boldsymbol{Y}_l = \boldsymbol{F}_l$ in Proposition 1. For one thing, this asymmetric graph structure, in the theoretical sense, conforms to the intuition of the better affinity graph we discussed earlier. For another, it also sheds some light on the proposed optimization framework to infer the weights in this graph structure later.

**Proposition 1.** *If $\boldsymbol{W}^*$ can be expressed as (4), $\boldsymbol{W}^*$ is an optimal solution given by (3) in Definition 1.*

$$\boldsymbol{W}^* = \begin{pmatrix} \boldsymbol{O} & \boldsymbol{O} \\ \boldsymbol{W}_{ul} & \boldsymbol{W}_{uu} \end{pmatrix}, \tag{4}$$

*where $\boldsymbol{W}_{ul} \in \mathbb{R}^{u \times l}$, and $\boldsymbol{W}_{uu} \in \mathbb{R}^{u \times u}$ can be non-zero submatrices with arbitrary entries.*

Proposition 1 provides an ad-hoc solution to Problem (3) (Appendix E.2). If the constructed graph is asymmetrical with edges from unlabeled nodes to labeled nodes only, then it is optimal by Definition 1. This asymmetrical optimal graph structure answers the first research question in Sect. 2.1.

The optimal asymmetric graph structure presented in Proposition 1 offers meaningful interpretations and notable advantages. It effectively eliminates the influence from unlabeled nodes to labeled nodes by enforcing $\boldsymbol{W}_{ll}$ and $\boldsymbol{W}_{lu}$ to be zero matrices. This aligns with the intuition that label information should be propagated from labeled nodes to unlabeled nodes in the GSSL methods, rather than the other way around. More technically, when applying the classic label inference algorithm LGC [82] on this optimal graph structure, the soft label matrix for unlabeled nodes now becomes $\boldsymbol{F}_u = (\boldsymbol{I} - \mu \boldsymbol{W}_{uu})^{-1} \boldsymbol{W}_{ul} \boldsymbol{Y}_L$ with some constant $0 \leq \mu \leq 1$. This formulation indicates that the LGC algorithm spreads supervision information from labeled nodes to unlabeled nodes once through $\boldsymbol{W}_{ul} \boldsymbol{Y}_l$, followed by propagating this information solely among unlabeled nodes through $(\boldsymbol{I} - \mu \boldsymbol{W}_{uu})^{-1} = \boldsymbol{I} + \mu \boldsymbol{W}_{uu} + \mu^2 \boldsymbol{W}_{uu}^2 + \cdots$. By Theorem 1, this optimal asymmetric graph guarantees zero empirical risk on the labeled nodes. Moreover, many existing graph construction methods in GSSL primarily focus on node features while disregarding label information. Consequently, these methods may produce heterophilous edges that connect nodes with similar features but different labels. Such edges violate the manifold assumption in GSSL, where nodes with the same label tend to be linked. During the label inference step, the label information of these nodes connected by heterophilous edges confuses each other during propagation, resulting in misleading predictions. By setting $\boldsymbol{W}_{ll} = \boldsymbol{O}$, our method eliminates these heterophilous edges completely. As a result, it increases the edge homophily ratio of the constructed graph and enhances the robustness of subsequent label inference algorithms, as validated in Appendix D.1.

## 3.2 Implementation: Block-wise Graph Learning Algorithm

### 3.2.1 Framework

By differentiating the roles that labeled and unlabeled nodes would play, we arrive at an optimal structure for the affinity graph in Proposition 1. However, an efficient algorithm to infer the exact weights or entries for blocks in Eq. (4) is still in demand. Motivated by previous works [29, 30, 19], we make the following reasonable restrictions or assumptions on $W$ to obtain a more meaningful graph for GSSL. First, the features for two directly connected nodes should not vary too much, no matter whether they are labeled or not. This agrees with the manifold assumption in GSSL, where links tend to form between similar nodes. Violation of this fundamental assumption usually causes a significant performance drop in the label inference step. Second, the constructed graph should be well-connected in terms of both $W_{ul}$ and $W_{uu}$ to ensure the supervision information could propagate freely among all nodes. Otherwise, some disconnected nodes may never receive any label information. Third, it is desirable to control the sparsity of the graph, avoiding an overly sparse or overly dense graph. We will empirically demonstrate the effects of the sparsity control later. We propose two similar optimization frameworks for $W_{ul}$ and $W_{uu}$ in Eq. (4) as Problem (5) and (6).

$$\min_{W_{ul}} \|W_{ul} \odot Z_{ul}\|_1 - \alpha_1 \mathbf{1}^\mathsf{T} \log(W_{ul}\mathbf{1}) - \alpha_1 \mathbf{1}^\mathsf{T} \log(W_{ul}^\mathsf{T}\mathbf{1}) + \beta_1\|W_{ul}\|_F^2, \text{s.t.} W_{ul} \geq 0. \quad (5)$$

$$\min_{W_{uu}} \|W_{uu} \odot Z_{uu}\|_1 - \alpha_2 \mathbf{1}^\mathsf{T} \log(W_{uu}\mathbf{1}) - \alpha_2 \mathbf{1}^\mathsf{T} \log(W_{uu}^\mathsf{T}\mathbf{1}) + \beta_2\|W_{uu}\|_F^2, \text{s.t.} W_{uu} \geq 0. \quad (6)$$

Here, we define the pairwise distance matrix $Z = \begin{pmatrix} Z_{ll} & Z_{ul}^\mathsf{T} \\ Z_{ul} & Z_{uu} \end{pmatrix} \in \mathbb{R}_+^{n \times n}$ with $Z_{ij} = \|x_i - x_j\|_2^2$. $\odot$ is the Hadamard product and $\log(\cdot)$ is the element-wise logarithm operator. Since Problem (5) and (6) share the same form, we will only focus on Problem (5) as an example. The first term $\|W_{ul} \odot Z_{ul}\|_1 = \sum_{1 \leq i \leq u, 1 \leq j \leq l} W_{ij}\|x_i - x_j\|_2^2$ encourages similar nodes to be connected with larger weights, meeting the manifold assumption. The second and third logarithmic barrier term act on the out-degree and in-degree vectors to make sure that each unlabeled node is connected by at least one labeled node and vice versa, improving the overall connectivity of the graph. The last Frobenius norm term measures the sparsity of the graph. The parameters $\alpha_1, \alpha_2, \beta_1, \beta_2$ are positive.

### 3.2.2 Optimization

For convenience of presentation, we view the entries in $W_{ul}$ ($W_{uu}$) as a new vector $w = \text{vec}[W_{ul}] \in \mathbb{R}_+^{ul}$. Similarly, we have $z = \text{vec}[Z_{ul}] \in \mathbb{R}_+^{ul}$. Accordingly, a linear mapping matrix $T_1 \in \{0,1\}^{u \times ul}$ transforms the edge weights to the corresponding out-degree vector (i.e. $T_1 w = W_{ul}\mathbf{1}$). Similarly, we have $T_2 w = W_{ul}^\mathsf{T}\mathbf{1}$ for the in-degree vector. Further, if we let $T^\mathsf{T} = (T_1^\mathsf{T}, T_2^\mathsf{T}) \in \{0,1\}^{n \times ul}$, $\alpha = \alpha_1 = \alpha_2$, and $\beta = \beta_1$ we can easily transform Problem (5) into Problem (7) (primal) in a more compact way with an extra linear constraint.

$$\min_{w,v} \quad f(w) + g(v) \quad \text{s.t.} \quad v = Tw, \quad (7)$$

with $f(w) = w^\mathsf{T} z + \beta\|w\|_2^2 + \mathbb{I}_{\{w \geq 0\}}, \quad g(v) = -\alpha \mathbf{1}^\mathsf{T} \log(v)$.

The state-of-the-art method [61] applies the linearized alternating direction method of multipliers (ADMM) algorithm [8] directly to the primal problem with a similar structure in Problem (7), which lacks the theoretical guarantee on its convergence rate since the objective function in Problem (7) has no Lipschitz gradient. Motivated by the recent work [45], we circumvent this critical issue by applying the FISTA algorithm [2], a proximal gradient method, to the dual problem of (7) instead. Motivated by recent advances [3, 61], we can now provide a better convergence rate with rigorous theoretical analysis so that our proposed graph construction method is much more efficient with guarantees.

**Dual Problem Formation**   We construct the Lagrangian function by introducing the Lagrangian multipliers $\lambda \in \mathbb{R}^n$ as $\mathcal{L}(w, v, \lambda) = f(w) + g(v) - \langle \lambda, Tw - v \rangle$. We establish the corresponding dual problem as Problem (8) by introducing the conjugate functions $f^*, g^*$ for simpler notation (Appendix E.3).

$$\min_{\lambda} \quad F(\lambda) + G(\lambda), \quad (8)$$

with $F(\boldsymbol{\lambda}) = f^*(\boldsymbol{T}^\intercal\boldsymbol{\lambda})$, $G(\boldsymbol{\lambda}) = g^*(-\boldsymbol{\lambda})$. By Slater's condition, we know that strong duality holds as long as $\boldsymbol{v}$ resides in the range of $\boldsymbol{T}$. Therefore, the optimal values for Problem (7) and (8) are identical, and the optimal solution for Problem (8) can be attained. Consequently, we can apply the FISTA algorithm to the dual problem to generate the dual sequence that converges to the optimal solution, and construct the corresponding optimal solution back for the primal problem.

**Dual Problem with FISTA**    It is not hard to show that $F(\boldsymbol{\lambda})$ is differentiable and $\nabla F(\boldsymbol{\lambda}) = \boldsymbol{T}\boldsymbol{x}^*$, where $\boldsymbol{x}^* = \arg\max_x\{(\boldsymbol{T}^\intercal\boldsymbol{\lambda})^\intercal\boldsymbol{x} - f(\boldsymbol{x})\}$ (Appendix E.4). Therefore, the dual problem minimizes the sum of a differentiable convex function $F$ and a closed proper convex function $G$. This structure of the dual Problem (8) immediately paves the way for applying proximal gradient methods. Here, for the sake of a better convergence rate, we apply the FISTA algorithm with a fixed step size to the dual Problem (8), and the following iteration schemes are performed. We first choose any $\boldsymbol{\lambda}^0 = \boldsymbol{\lambda}^{-1}$ and fix the step size as $t = \frac{2\beta}{l+u}$. Henceforth, $k = 1, 2, \cdots$, we repeat the following two steps as

$$\boldsymbol{\mu}^k = \boldsymbol{\lambda}^{k-1} + \frac{k-2}{k+1}(\boldsymbol{\lambda}^{k-1} - \boldsymbol{\lambda}^{k-2}), \tag{9}$$

$$\boldsymbol{\lambda}^k = \mathrm{prox}_{tG}\left(\boldsymbol{\mu}^k - t\nabla F(\boldsymbol{\mu}^k)\right), \tag{10}$$

Hence, based on the above-mentioned properties of $\nabla F(\boldsymbol{\mu}^k)$, we have $\nabla F(\boldsymbol{\mu}^k) = \boldsymbol{T}\bar{\boldsymbol{w}}^k$ with $\bar{\boldsymbol{w}}^k$ set as $\bar{\boldsymbol{w}}^k = \arg\max_w\left\{(\boldsymbol{T}^\intercal\boldsymbol{\mu}^k)^\intercal\boldsymbol{w} - f(\boldsymbol{w})\right\} = \left[\frac{\boldsymbol{T}^\intercal\boldsymbol{\mu}^k-\boldsymbol{z}}{2\beta}\right]_+$ (Appendx E.5).

Let $\boldsymbol{p}^k = \boldsymbol{\mu}^k - t\boldsymbol{T}\bar{\boldsymbol{w}}^k$. By the extended Moreau decomposition [43], $\mathrm{prox}_{\gamma h}(z) + \gamma\,\mathrm{prox}_{\gamma^{-1}h^*}(z/\gamma) = z, \forall z$. We have $\mathrm{prox}_{tG}(\boldsymbol{p}^k) = \boldsymbol{p}^k - t\,\mathrm{prox}_{t^{-1}G^*}(t^{-1}\boldsymbol{p}^k) = \boldsymbol{\mu}^k - t(\boldsymbol{T}\bar{\boldsymbol{w}}^k - \bar{\boldsymbol{u}}^k)$. Here, $\bar{\boldsymbol{u}}^k = \mathrm{prox}_{t^{-1}g}(\boldsymbol{T}\bar{\boldsymbol{w}}^k - t^{-1}\boldsymbol{\mu}^k)$. Note that $g(\boldsymbol{v}) = -\alpha\boldsymbol{1}^\intercal\log(\boldsymbol{v})$ and by the definition of proximal mapping, it is easy to prove that $\mathrm{prox}_{t^{-1}g}(\boldsymbol{v}) = \frac{1}{2}(\boldsymbol{v} + \sqrt{\boldsymbol{v}\odot\boldsymbol{v} + 4\alpha t^{-1}\boldsymbol{1}})$. Therefore, we can simplify the updating step (10) as the following three steps (Appendix E.6).

$$\bar{\boldsymbol{w}}^k = \left[\frac{\boldsymbol{T}^\intercal\boldsymbol{\mu}^k - \boldsymbol{z}}{2\beta}\right]_+, \tag{11}$$

$$\bar{\boldsymbol{u}}^k = \frac{1}{2}(\boldsymbol{T}\bar{\boldsymbol{w}}^k - t^{-1}\boldsymbol{\mu}^k) + \frac{1}{2}\sqrt{(\boldsymbol{T}\bar{\boldsymbol{w}}^k - t^{-1}\boldsymbol{\mu}^k)\odot(\boldsymbol{T}\bar{\boldsymbol{w}}^k - t^{-1}\boldsymbol{\mu}^k) + 4\alpha t^{-1}\boldsymbol{1}}, \tag{12}$$

$$\boldsymbol{\lambda}^k = \boldsymbol{\mu}^k - t(\boldsymbol{T}\bar{\boldsymbol{w}}^k - \bar{\boldsymbol{u}}^k), \tag{13}$$

Finally, we arrive at Procedure `GWBI`, where the optimal graph weights are inferred for one block like $\boldsymbol{W}_{uu}$ in the optimal graph structure suggested in Proposition 1. Here, instead of directly optimizing the primal variable of the block graph weight vector $\boldsymbol{w}$, we consider its corresponding dual variable $\boldsymbol{\lambda}$ for a better convergence rate. In each iteration step, we first find an extrapolated point $\boldsymbol{\mu}$ based on the points $\boldsymbol{\lambda}$ from two previous steps (line 3). We then perform the proximal gradient update on this extrapolated point (lines 4-6) to obtain $\boldsymbol{\lambda}$ for the next iteration. Note that lines 4-6 are the detailed instantiation of Eq. (10). Finally, we can convert the dual variable $\boldsymbol{\lambda}$ back to the desired primal variable $\boldsymbol{w}$ based on line 4 after its convergence. With this core procedure in hand, we plug it into the proposed Algorithm 1, **B**lock-wise **A**ffinity **G**raph **L**earning (BAGL) algorithm, in Appendix C.

### 3.3   Theoretical Analysis

We prove that our proposed optimization method for Problem (7) in Procedure `GWBI` enjoys the guarantee of the global convergence rate, where the generated primal sequence converges to the global optimal solution at a rate of $O(\frac{1}{k})$ with a fixed step size. To begin with, it is well known that the FISTA algorithm enjoys the global convergence rate of $O(\frac{1}{k^2})$ [2]. For simplicity, we focus on the results when optimizing Problem (5), which can be viewed as a general case of Problem (6) when $l \neq u$. Therefore, $Q(\boldsymbol{\lambda}) \equiv F(\boldsymbol{\lambda}) + G(\boldsymbol{\lambda})$ converges to the dual optimal value $Q(\boldsymbol{\lambda}^*)$ at a rate of $O(\frac{1}{k^2})$ due to this well-known fact [2], which yields Theorem 2.

**Theorem 2.** *Let $\{\boldsymbol{\lambda}^k\}$ be the dual sequence generated by Procedure `GWBI`, then*

$$Q(\boldsymbol{\lambda}^k) - Q(\boldsymbol{\lambda}^*) \leq \frac{l+u}{\beta(k+1)^2}\|\boldsymbol{\lambda}^0 - \boldsymbol{\lambda}^*\|_2^2. \tag{14}$$

---

**Procedure** GraphWeightBlockInference($\boldsymbol{z}$,$\boldsymbol{T}$,$\alpha$,$\beta$,$t$,$\epsilon$)

---

**Input:** Distance vector $\boldsymbol{z}$, linear mapping matrix $\boldsymbol{T}$, balancing parameters $\alpha$ and $\beta$, step size $t$, error tolerance parameter $\epsilon$.

**Output:** Block graph weight vector $\hat{\boldsymbol{w}}$.

1 Initialize $\boldsymbol{\lambda}^0 = \boldsymbol{\lambda}^{-1}$ at random and set $k = 1$;

2 **do**

3     $\boldsymbol{\mu}^k = \boldsymbol{\lambda}^{k-1} + \frac{k-2}{k+1}(\boldsymbol{\lambda}^{k-1} - \boldsymbol{\lambda}^{k-2})$;

4     $\bar{\boldsymbol{w}}^k = \left[ \frac{\boldsymbol{T}^\mathsf{T}\boldsymbol{\mu}^k - \boldsymbol{z}}{2\beta} \right]_+$;

5     $\bar{\boldsymbol{u}}^k = \frac{1}{2}(\boldsymbol{T}\bar{\boldsymbol{w}}^k - t^{-1}\boldsymbol{\mu}^k) + \frac{1}{2}\sqrt{(\boldsymbol{T}\bar{\boldsymbol{w}}^k - t^{-1}\boldsymbol{\mu}^k) \odot (\boldsymbol{T}\bar{\boldsymbol{w}}^k - t^{-1}\boldsymbol{\mu}^k) + 4\alpha t^{-1}\mathbf{1}}$;

6     $\boldsymbol{\lambda}^k = \boldsymbol{\mu}^k - t(\boldsymbol{T}\bar{\boldsymbol{w}}^k - \bar{\boldsymbol{u}}^k)$;

7     $k \leftarrow k + 1$;

8 **while** $\|\boldsymbol{\lambda}^k - \boldsymbol{\lambda}^{k-1}\|_\infty > \epsilon$;

9 **return** $\hat{\boldsymbol{w}} = \bar{\boldsymbol{w}}^k$;

---

We then consider the corresponding primal sequence $\{\boldsymbol{w}^k\}$ and its convergence rate. Since the strong duality holds and a dual optimal solution $\boldsymbol{\lambda}^*$ exists, any primal optimal point $(\boldsymbol{w}^*, \boldsymbol{v}^*)$ is also a minimizer of $\mathcal{L}(\boldsymbol{w}, \boldsymbol{v}, \boldsymbol{\lambda}^*)$. This motivates us to construct the primal sequence $\{\boldsymbol{w}^k\}$ based on the dual sequence $\{\boldsymbol{\lambda}^k\}$ as $\boldsymbol{w}^k = \arg\min_{\boldsymbol{w}} \mathcal{L}(\boldsymbol{w}, \boldsymbol{v}, \boldsymbol{\lambda}^k) = \arg\max_{\boldsymbol{w}}\{\langle \boldsymbol{T}^\mathsf{T}\boldsymbol{\lambda}^k, \boldsymbol{w}\rangle - f(\boldsymbol{w})\}$. Thanks to Theorem 3, this primal sequence $\{\boldsymbol{w}^k\}$ is guaranteed to converge to the optimal primal solution $\boldsymbol{w}^*$ of Problem (7) at the rate of $O(\frac{1}{k})$.

**Theorem 3.** *Let $\boldsymbol{w}^k = \arg\max_{\boldsymbol{w}}\{\langle \boldsymbol{T}^\mathsf{T}\boldsymbol{\lambda}^k, \boldsymbol{w}\rangle - f(\boldsymbol{w})\}$ with the dual sequence $\{\boldsymbol{\lambda}^k\}$ given by Procedure* `GWBI`. *$\boldsymbol{w}^*$ and $\boldsymbol{\lambda}^*$ are the optimal solution of Problem (7) and the optimal solution of Problem (8), respectively. We have,*

$$\|\boldsymbol{w}^k - \boldsymbol{w}^*\|_2 \leq \frac{\sqrt{l+u}}{\beta(k+1)}\|\boldsymbol{\lambda}^0 - \boldsymbol{\lambda}^*\|_2. \tag{15}$$

Motivated by [45], Theorem 3 establishes that the proposed method exhibits a sub-linear convergence rate, which represents a state-of-the-art result for optimization-based graph construction methods, accompanied by a global convergence guarantee. This improved convergence rate is primarily attributed to the utilization of the FISTA algorithm applied to the dual problem. Time complexity analysis is included in Appendix G.5. More results regarding the robustness of our method are discussed in Appendix D.

## 4 Experiments

In this section, we conduct numerical experiments on both synthetic and real-world datasets to demonstrate the advantages of our proposed BAGL method in terms of efficacy and convergence. Robustness analysis (Appendix G.3) and more experimental results are included in Appendix G.

### 4.1 Baseline Models

We choose the following graph construction methods in GSSL for comparison. Radial basis function kernel (RBF) [85] and kNN graph [17] are two classic methods. Smooth graph learning (SGL) [29] is a popular method in the graph signal processing domain. RGCLI [7] is another label-informed graph construction method. Anchor Graph Regularization (AGR) [40] deals with large-scale graph construction. GraphEBM [14] and BCAN [63] are two state-of-the-art methods. The former exploits the energy-based model while the latter constructs a bipartite graph.

All the hyper-parameters are fine-tuned with the grid search method. We repeat the experiment 20 times for each case and report the average result with optimal parameter setting in the efficacy analysis. Unless otherwise specified, the default label inference algorithm is LGC, and the label rate is ten labeled samples per class. More details on the experimental settings can be found in Appendix F.

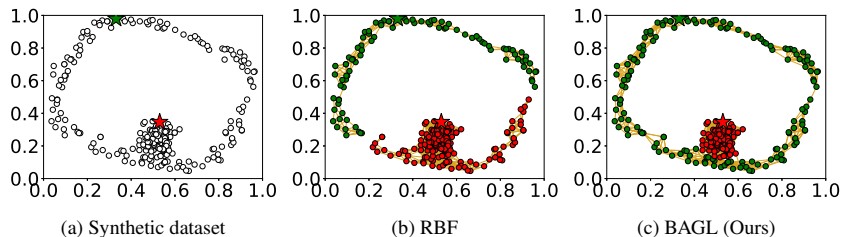

(a) Synthetic dataset    (b) RBF    (c) BAGL (Ours)

Figure 1: Visualization of classification results on the synthetic dataset.

Table 1: Description of datasets

| Dataset | #Samples $n$ | #Features $d$ | #Classes $c$ |
|---|---|---|---|
| ORHD | 5,620 | 64 | 10 |
| USPS | 9,298 | 256 | 10 |
| COIL100 | 7,200 | 1,024 | 100 |
| TDT2 | 9,394 | 36,771 | 30 |
| MNIST | 70,000 | 784 | 10 |
| EMNIST Letters | 145,600 | 784 | 20 |

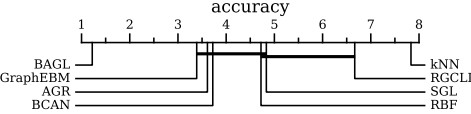

Figure 2: Comparison of BAGL against baseline models with the *Bonferroni-Dunn* test. CD=2.154 at 0.05 significance level.

## 4.2 Synthetic Dataset

We generate a synthetic dataset as shown in Fig. 1 (a). The constructed dataset contains two clusters, a dense Gaussian cluster surrounded by a sparse ring-like cluster. With only one labeled sample given in each cluster, we compare the result of our proposed BAGL method (Fig. 1 (c)) with the result of the most popular method RBF (Fig. 1 (b)). We use the coordinates as the node feature and set the width in the RBF kernel as $0.7$. For visualization purposes, we show the adjacency matrix with yellow line segments connecting the node pairs if the weight associated with the edge is greater than $0.5$ after normalization. The direction of the edge is ignored in Fig. 1 (c). For a fair comparison, we perform label propagation [85] on both constructed affinity graphs. We can see that BAGL can recover two ground-truth clusters much better. Unlike RBF, BAGL can improve the connectivity by the logarithm penalty term and reduce the inter-cluster links by the block-wise design.

## 4.3 Real-world Datasets

Classification tasks are implemented to assess the performance of BAGL against all graph construction baseline methods on six real-world datasets, listed in Table 1. **ORHD** (Optical Recognition of Handwritten Digits Data Set), **USPS**, **MNIST**, and **EMNIST Letters** are four popular digits image datasets. **COIL100** is an object image dataset. **TDT2** is a text dataset. We fix the number of anchor nodes as 1000 in four datasets (COIL100, USPS, ORHD and TDT2), while for the rest two datasets (MNIST, EMNIST-Letters), the number of anchors is fixed as 2000. We perform tf-idf and principal component analysis (PCA) as the pre-processing step on TDT2 dataset. The default label inference algorithm is LGC, with ten labels per class. Further details can be found in Appendix F.1.

### 4.3.1 Efficacy

We fix the number of labeled samples per class to ten and select three label inference methods for the second phase of GSSL. We report average results by performing 20 trials for each algorithm over all the settings in Table 2. Our algorithm outperforms all methods in the USPS, TDT2, and EMNIST-Letters datasets, indicating that BAGL can learn an optimal graph for label inference algorithms in the unified framework. Moreover, we perform the Friedman test with the Bonferroni-Dunn post hoc test for statistical significance analysis. Fig. 2 illustrates the critical difference (CD) diagram on the accuracy, where the average rank is marked along the axis with lower (better) ranks to the left. If the average rank difference between two models is greater than one CD, the relative performance is believed to be different. Accordingly, BAGL significantly outperforms all other baselines by a large margin. We also conduct experiments under low label rates with LGC fixed as the label inference method. Fig. 3 (a) and (b) demonstrate that BAGL performs relatively well with low label rates. This phenomenon can be attributed to the utilization of label information in BAGL. (Appendix G.1)

Table 2: Classification accuracy and standard deviation (%) on real-world datasets.

| | | RBF | kNN | SGL | RGCLI | AGR | GraphEBM | BCAN | BAGL |
|---|---|---|---|---|---|---|---|---|---|
| ORHD | GRF | 97.46±0.36 | 86.59±1.26 | 94.68±0.66 | 88.24±3.11 | 97.63±0.53 | 95.13±0.41 | 97.49±0.47 | **97.88±0.40** |
| | LGC | 97.56±0.29 | 87.61±2.30 | 95.75±0.74 | 89.32±2.58 | 96.90±0.47 | 95.78±0.53 | 97.27±0.63 | **98.04±0.71** |
| | GCN | 98.11±0.44 | 90.64±3.54 | 95.80±0.59 | 89.37±3.02 | **98.42±0.50** | 98.08±0.70 | 98.15±0.62 | |
| USPS | GRF | 94.53±0.65 | 81.42±0.98 | 87.67±0.40 | 84.15±1.85 | 93.64±0.62 | 94.26±0.36 | 93.98±0.60 | **96.56±0.93** |
| | LGC | 94.75±0.42 | 85.13±1.18 | 86.44±0.51 | 84.86±1.63 | 95.92±0.51 | 94.30±0.30 | 95.07±0.75 | **96.77±0.66** |
| | GCN | 94.98±0.21 | 86.20±2.03 | 90.31±0.32 | 86.09±1.72 | 95.78±0.49 | 95.81±0.48 | 95.79±0.55 | **97.20±0.64** |
| COIL100 | GRF | 94.40±0.19 | 81.24±1.64 | 92.65±0.82 | 87.48±2.30 | 86.54±0.40 | 85.22±0.57 | 84.51±0.59 | **94.78±0.53** |
| | LGC | **95.13±0.37** | 83.66±1.35 | 93.27±1.03 | 87.79±2.47 | 87.81±0.57 | 88.50±0.47 | 87.06±0.63 | 94.93±0.45 |
| | GCN | 94.31±0.25 | 87.64±1.72 | 93.52±0.91 | 89.30±1.65 | 94.63±0.51 | 90.15±0.39 | 90.02±0.48 | **94.99±0.88** |
| TDT2 | GRF | 89.22±0.79 | 80.09±2.69 | 92.13±0.99 | 86.51±3.42 | 94.47±0.79 | 93.64±0.74 | 95.95±0.60 | **96.01±0.91** |
| | LGC | 89.67±0.46 | 82.35±3.04 | 92.96±1.24 | 87.60±2.84 | 94.15±0.67 | 93.97±0.61 | 94.13±0.79 | **95.42±0.71** |
| | GCN | 92.89±0.68 | 85.77±2.41 | 94.39±0.83 | 89.94±3.15 | 95.36±0.85 | 94.78±0.69 | 96.30±0.77 | **96.33±0.85** |
| MNIST | GRF | 83.60±0.24 | 64.20±1.82 | 95.03±0.77 | 87.65±2.07 | 91.02±0.31 | 95.39±0.31 | 92.41±0.47 | **95.40±0.62** |
| | LGC | 84.12±0.17 | 68.86±1.63 | 94.40±0.52 | 88.22±2.36 | 94.79±0.37 | **95.43±0.47** | 93.55±0.58 | 95.42±0.51 |
| | GCN | 87.03±0.32 | 74.93±1.77 | 95.18±0.47 | 90.47±2.11 | 95.30±0.21 | **95.51±0.40** | 94.84±0.37 | 95.47±0.43 |
| EMNIST Letters | GRF | 50.74±0.16 | 41.85±1.35 | 62.34±0.58 | 54.04±1.92 | 64.38±0.65 | 65.03±0.27 | 67.69±0.39 | **67.81±0.40** |
| | LGC | 54.35±0.27 | 49.51±1.57 | 63.02±0.41 | 57.18±2.30 | 66.49±0.49 | 66.67±0.23 | 68.92±0.46 | **69.03±0.44** |
| | GCN | 59.28±0.24 | 51.48±1.44 | 66.51±0.37 | 58.82±1.74 | 67.21±0.71 | 68.56±0.19 | 68.97±0.50 | **69.14±0.47** |

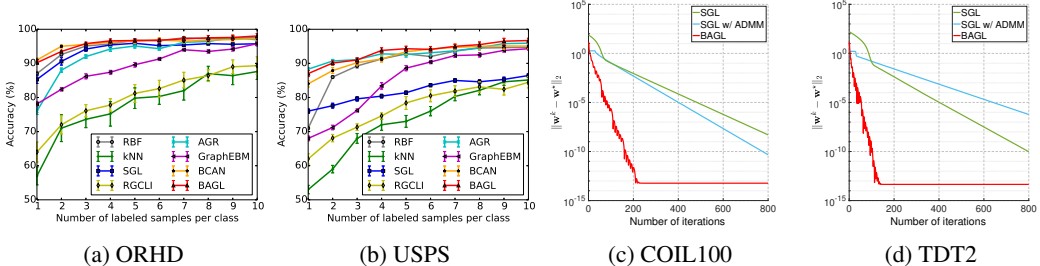

(a) ORHD  (b) USPS  (c) COIL100  (d) TDT2

Figure 3: (a) (b) Classification results and (c) (d) convergence results on real-world datasets.

### 4.3.2 Convergence

As an essential part of the overhead for BAGL, Procedure GWBI needs to be efficient for practice. Compared with two other optimization-based methods sharing a similar objective, SGL and its accelerated version by ADMM [61], BAGL enjoys the fastest convergence rate on both datasets. We sample 1% nodes in each dataset for the convenience of presentation in Fig. 3 (c) and (d). Its outstanding performance confirms the theoretical analysis in Theorem 3. (Appendix G.2).

### 4.3.3 Ablation Study

To obtain a better understanding of why the proposed BAGL works, we perform some ablation studies to empirically show how the key design of BAGL will potentially affect performance. We create three variants based on the original version of BAGL. First, to demonstrate the significance of the optimal asymmetric structure, we now do not differentiate labeled nodes and unlabeled nodes, and let any graph structure be the potential optimal structure without the constraints of $\boldsymbol{W}_{ul} = \boldsymbol{O}$ and $\boldsymbol{W}_{uu} = \boldsymbol{O}$. We call this variant *BAGL w/o optimal structure*. Second, to reveal the importance of the connectivity regularization term, we set $\alpha = 0$ in Procedure GWBI to remove the connectivity consideration. This variant is termed *BAGL w/o connectivity*. Third, to investigate the effects of sparsity control, we set $\beta = 10e - 5 \approx 0$ in Procedure GWBI to allow the sparsity of the constructed graph to vary arbitrarily. The last variant of BAGL is abbreviated as *BAGL w/o sparsity*. We conduct the experiments on the ORHD dataset with the same setting as Table 2 and report the classification accuracy results in Table 3. The proposed optimal asymmetric graph structure contributes most to the success of BAGL. The connectivity regularization term and the sparsity control term also matter since they together encourage a more sparse graph (the latter) but without disconnected components (the former), which is a more favorable graph for the second label inference step.

Table 3: Ablation study of BAGL on the ORHD dataset.

| | | BAGL w/o optimal structure | BAGL w/o connectivity | BAGL w/o sparsity | BAGL |
|---|---|---|---|---|---|
| ORHD | GRF | 95.71±0.84 | 96.71±0.62 | 96.05±0.73 | **97.88±0.40** |
| | LGC | 96.34±0.66 | 96.60±0.57 | 97.19±0.48 | **98.04±0.71** |
| | GCN | 97.29±0.59 | 97.89±0.64 | 97.74±0.53 | **98.15±0.62** |

## 5 Conclusion

In this paper, we propose a novel approach to graph construction for graph-based semi-supervised learning. Building upon the optimal asymmetric graph structure derived from theoretical insights, we develop an efficient block-wise graph construction method that guarantees faster convergence. Our approach combines theoretical insights with practical considerations to provide a more effective and reliable framework for the graph construction step in graph-based semi-supervised learning.

## 6 Limitations

BAGL is an optimization-based method for the graph construction step in graph-based semi-supervised learning. Graph Neural Networks (GNNs) excel in learning representations for graph-structured data [64, 54, 12, 49, 39, 78, 41, 75, 13, 53, 77, 51, 73, 52, 50, 38, 76, 11]. More recent graph structure learning methods aim to learn a clean graph structure from the given noisy graph so that the subsequent GNNs trained on this learned clean graph can obtain better performance. In GSSL, however, there is no given graph structure, and we need to learn the graph structure based on the node features only. Therefore, it is a more challenging task compared to graph structure learning. Therefore, we do not compare our method with other graph structure learning methods since their settings and goals are slightly different. We leave the investigation of graph structure learning for GSSL as future work since it is currently out of the scope of this work.

The other limitation of BAGL is it is only suitable for the transductive setting. If we have nodes or samples unseen in the training set, we have to construct the affinity graph again by executing BAGL again to infer their labels, which is often time-consuming and troublesome regarding efficiency. This is not desirable in real-world applications since we often come across new training samples after we build the affinity graph. We also leave the investigation of the inductive extension of BAGL as future work since this lack of inductive generalization is a well-known challenge in graph-based semi-supervised learning.

Even though BAGL is quite efficient in terms of convergence rate, it may still have computational issues when dealing with extremely large-scale datasets with billions of samples because the time spent on finishing one iteration during the optimization would increase dramatically when the number of training samples is extremely large. We leave the exploration of graph construction methods for extremely large-scale datasets as future work. Other potential applications of our method can be explored in the hyperbolic space [67, 70, 68, 71, 69, 66, 72] or in the natural language processing domain [32, 34, 33, 21, 20, 55, 42, 80, 81, 79].

## Acknowledgements

The work described in this paper was partially supported by the National Key Research and Development Program of China (No. 2018AAA0100204) and RGC General Research Funding Scheme (GRF) 14222922 (CUHK 2151185).

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
