# A  Notation Table

We summarize all the useful notations in this paper as Table 4. If $M$ is a matrix, unless otherwise specified, we use $m_i$ to denote the $i$-th row of $M$, $M_j$ to denote the $j$-th column of $M$, and $M_{ij}$, $M_{i,j}$ or $M_{ij}$ to denote the entity at the $i$-th row and $j$-th column.

Table 4: Important notations and corresponding descriptions.

| Notations | Descriptions |
|---|---|
| $n$ | Number of samples/nodes |
| $l$ | Number of labeled samples/nodes |
| $u$ | Number of unlabeled samples/nodes |
| $d$ | Dimension of sample's feature space |
| $c$ | Number of classes |
| $\boldsymbol{x}_i \in \mathbb{R}^d$ | $i$-th sample's feature vector |
| $y_i \in \{k \in \mathbb{N}^+ \mid 1 \leq k \leq c\}$ | $i$-th sample's ground-truth label value |
| $\boldsymbol{y}_i \in \{0,1\}^c$ | $i$-th sample's ground-truth label vector |
| $\boldsymbol{f}_i \in \mathbb{R}^c$ | $i$-th sample's soft label vector |
| $\boldsymbol{I}$ | Identity matrix |
| $\boldsymbol{X} \in \mathbb{R}^{n \times d}$ | Node feature matrix |
| $\boldsymbol{Y} \in \{0,1\}^{n \times c}$ | Ground-truth label matrix |
| $\boldsymbol{F} \in \mathbb{R}^{n \times c}$ | Soft label matrix |
| $\boldsymbol{A} \in \{0,1\}^{n \times n}$ | Adjacency matrix |
| $\boldsymbol{W} \in \mathbb{R}^{n \times n}$ | Weight matrix |
| $\boldsymbol{D} \in \mathbb{R}^{n \times n}$ | Degree matrix |
| $\boldsymbol{L} \in \mathbb{S}_+^n$ | (Combinatorial) Laplacian matrix |
| $\boldsymbol{\mathcal{L}} \in \mathbb{S}_+^n$ | Normalized Laplacian matrix |
| $\boldsymbol{S} \in \mathbb{S}_+^n$ | Smoothing matrix |
| $\boldsymbol{Z} \in \mathbb{R}_+^{n \times n}$ | Distance matrix |
| $\|\cdot\|_F$ | Frobenius norm |
| $\odot$ | Hadamard product |
| $\mathrm{vec}[\cdot]$ | Matrix vectorization |
| $\mathbb{I}_A$ | Indicator function of set $A$ |
| $f^*$ | Conjugate of function $f$ |

# B  Related Work

## B.1  Affinity Graph Construction in GSSL

A line of recent research works indicates that the key to the success of the GSSL is to construct high-quality graphs rather than more powerful label inference algorithms [36, 37]. Based on the review in [52], we can divide the existing graph construction methods in GSSL into two categories: distance metric-based methods and distance metric-based methods. The distance metric-based approaches compute the similarities between samples based on certain distance measurements, in which a smaller distance implies a higher level of similarity. Representative examples include the Radial basis function (RBF) kernel and kNN graphs [17].. b-matching graph [26] solves the issue of varying node degrees in the kNN graph. Smooth graph learning (SGL) [29] and its variants [19] are proposed from the view of graph signal processing with theoretical guarantees on convergence. More recent works focus on how to incorporate label information for robustness [6, 7], or probabilistic view for interpretability [14]. The data representation-based graph methods measure the similarities by the representation coefficients between samples and some chosen anchor nodes. Linear neighborhood propagation (LNP) [57, 58] reconstructs each sample using a convex combination of its k nearest neighbors. Anchor Graph Regularization (AGR) [40] enhances the scalability by introducing the anchor nodes. There are also several recent works utilizing label information to guide the graph construction step. ALGSSL adaptively updates the graph during the semi-supervised learning process [74]. SSLRR graph [86] adds the low-rank representation with coefficients between two labeled samples with different labels constrained to zero. Most recent works [24, 23, 63] further improve the efficiency via the bipartite graph construction.

## B.2 Unified Framework for Label Inference in GSSL

Most GSSL methods focus on the label inference step, where an affinity graph is assumed to have been constructed already. Gaussian Random Field (GRF) [85] is the first to propose the idea of label propagation by using the combinatorial Laplacian matrix as the smoothing matrix. It also tolerates no prediction errors on the labeled nodes. Local and Global Consistency (LGC) [82] relaxes the GRF method by eliminating the restriction of zero empirical risk on labeled nodes and exploits the normalized Laplacian matrix for smoothing instead. Graph Convolutional Networks (GCN) [31] and its variants [10, 22] are increasingly popular methods with the success of deep learning. Recent research work has shown its deep connection with the classic label propagation algorithm [27, 65, 59].

Following the regularization framework in graph-based semi-supervised learning [52], we can recover most of the label inference algorithms in GSSL with a unified framework as Eq. (16).

$$ \boldsymbol{F}^* = \arg\min_{\boldsymbol{F}} Q(\boldsymbol{F}) = \arg\min_{\boldsymbol{F}} \left\{ \mathrm{Tr}\left(\boldsymbol{F}^\mathsf{T}\boldsymbol{S}\boldsymbol{F}\right) + \mathrm{Tr}\left((\boldsymbol{F}-\boldsymbol{Y})^\mathsf{T}\boldsymbol{\Lambda}(\boldsymbol{F}-\boldsymbol{Y})\right) \right\}. \qquad (16) $$

Table 5: Summary of several representative GSSL methods under the optimization framework in Eq. (1).

| Method | $\boldsymbol{S}$ | $\boldsymbol{\Lambda}$ |
|---|---|---|
| Gaussian Random Field (GRF) [85] | $\boldsymbol{L}$ | $\lim_{\lambda\to\infty}\lambda\boldsymbol{I}$ |
| Local and Global Consistency (LGC) [82] | $\mathcal{L}$ | $\lambda\boldsymbol{I}$ |
| Random Walk Smoothing [83] | $\mathcal{L}'$ | $\lambda\boldsymbol{I}$ |
| Tikhonov Smoothing [4] | $\boldsymbol{L}$ | $\begin{pmatrix} \lambda\boldsymbol{I}_l & \boldsymbol{O} \\ \boldsymbol{O} & \boldsymbol{O} \end{pmatrix}$ |
| Hub & Authority Smoothing [84] | $(1-\gamma)\mathcal{L}_A + \gamma\mathcal{L}_H$ | $\lambda\boldsymbol{I}$ |
| Manifold Smoothing [5] | $\mathcal{K} + \gamma\boldsymbol{L}$ | $\begin{pmatrix} \lambda\boldsymbol{I}_l & \boldsymbol{O} \\ \boldsymbol{O} & \boldsymbol{O} \end{pmatrix}$ |
| Graph Convolutional Networks (GCN) [31] | $\boldsymbol{L} + \gamma\boldsymbol{L} + \gamma^2\boldsymbol{L}^2 + \cdots$ | $\begin{pmatrix} \lambda\boldsymbol{I}_l & \boldsymbol{O} \\ \boldsymbol{O} & \boldsymbol{O} \end{pmatrix}$ |
| $\ell_p$ Laplacian Regularization [1] | $\mathcal{L}^{(p)}$ | $\lambda\boldsymbol{I}$ |
| Poisson Learning [9] | $\mathcal{L}^u$ | $\begin{pmatrix} \lambda_0\boldsymbol{I}_l & \boldsymbol{O} \\ \boldsymbol{O} & \lambda_1\boldsymbol{I}_u \end{pmatrix}$ |

We summarize several representative works under this unified framework by listing the exact forms of two key matrices $\boldsymbol{S}$ and $\boldsymbol{\Lambda}$ in Table 5. Gaussian Random Field (GRF) [85] is the first to propose the idea of label propagation by using the combinatorial Laplacian matrix as the smoothing matrix. It also tolerates no prediction errors on the labeled nodes, so the trade-off parameter can be set to infinity. Local and Global Consistency (LGC) [82] relaxes the GRF method by eliminating the restriction of zero empirical risk on labeled nodes and exploits the normalized Laplacian matrix for smoothing instead. Random Walk Smoothing [83] extends LRC for directed graphs by indirectly operating LGC on a modified undirected graph with a new normalized Laplacian matrix $\mathcal{L}'$. Tikhonov Smoothing [4] only uses the labeled nodes in the quadratic error term. Hub & Authority Smoothing [84] proposes another random-walk-based strategy on directed graphs that is motivated by the hub and authority web model. Its smoothing matrix is more complex with two underlying Laplacian matrices $\mathcal{L}_A, \mathcal{L}_H$ for in-links and out-links. Manifold Smoothing [5] adds a Mercer kernel $\mathcal{K}$ in the smoothing matrix, which controls the smoothness of the predictions in the ambient space. Graph Convolutional Networks (GCN) [31] and its variants are increasingly popular methods with the success of deep learning. We present the result of GCN after removing all the non-linear activation functions. Recent research work has shown its deep connection with classic label propagation algorithm [27, 35, 65, 59]. $\ell_p$ Laplacian Regularization extends smoothness functional with $p > 2$ by manipulating $\boldsymbol{L}^{(p)}$. Poisson Learning [9] follows the idea in $\ell_p$ Laplacian Regularization by adding the Poisson equation on the graph.

## C   Pseudocodes for Proposed Method BAGL

In this algorithm, we first prepare the input for Procedure `GWBI` in lines 1-4, including the distance matrix $\boldsymbol{Z}$, two linear maps $\boldsymbol{T}_a, \boldsymbol{T}_b$ for getting out-degree and in-degree vectors, and step size settings.

Then we run Procedure GWBI for two blocks $\boldsymbol{W}_{ul}, \boldsymbol{W}_{uu}$ (lines 5-6), and recover the ultimate optimal affinity graph (lines 7-8).

---

**Algorithm 1:** Block-wise Affinity Graph Learning (BAGL)

**Input:** Labeled samples $\{\boldsymbol{x}\}_{i=1}^{l}$, unlabeled samples $\{\boldsymbol{x}\}_{i=l+1}^{l+u}$, balancing parameters $\alpha_1, \alpha_2$ and $\beta_1, \beta_2$, error tolerance parameter $\epsilon$.

**Output:** Weight matrix of the constructed affinity graph $\hat{\boldsymbol{W}}$.

1 Construct distance matrix $\boldsymbol{Z}$ with $Z_{ij} = \|\boldsymbol{x}_i - \boldsymbol{x}_j\|_2^2$;
2 Construct $\boldsymbol{T}_a$ such that $\boldsymbol{T}_a \text{vec}[\boldsymbol{W}_{ul}] = (\boldsymbol{W}_{ul}^\intercal, \boldsymbol{W}_{ul})\mathbf{1}, \forall \boldsymbol{W}_{ul} \in \mathbb{R}_+^{u \times l}$;
3 Construct $\boldsymbol{T}_b$ such that $\boldsymbol{T}_b \text{vec}[\boldsymbol{W}_{uu}] = (\boldsymbol{W}_{uu}^\intercal, \boldsymbol{W}_{uu})\mathbf{1}, \forall \boldsymbol{W}_{uu} \in \mathbb{R}_+^{u \times u}$;
4 Set the fixed step size $t_1 = \frac{2\beta_1}{l+u}$ and $t_2 = \frac{\beta_2}{u}$;
5 $\hat{\boldsymbol{w}}_{ul} \leftarrow \texttt{GWBI}(\text{vec}(\boldsymbol{Z}_{ul}), \boldsymbol{T}_a, \alpha_1, \beta_1, t_1, \epsilon)$;
6 $\hat{\boldsymbol{w}}_{uu} \leftarrow \texttt{GWBI}(\text{vec}(\boldsymbol{Z}_{uu}), \boldsymbol{T}_b, \alpha_2, \beta_2, t_2, \epsilon)$;
7 Reconstruct $\hat{\boldsymbol{W}}_{ul}, \hat{\boldsymbol{W}}_{uu}$ based on $\hat{\boldsymbol{w}}_{ul}, \hat{\boldsymbol{w}}_{uu}$ respectively;
8 **return** $\hat{\boldsymbol{W}} = \begin{pmatrix} \boldsymbol{O} & \boldsymbol{O} \\ \hat{\boldsymbol{W}}_{ul} & \hat{\boldsymbol{W}}_{uu} \end{pmatrix}$;

---

## D  More Theoretical Analysis

### D.1  Robustness Interpretation

Following the proposed framework [60], we show that the proposed block-wise graph learning framework for the optimal structure introduced in Proposition 1 can also enhance the robustness when compared with the popular methods [29, 18] that are designed for the whole graph. To combine Problem (5) and (6) into one optimization problem in matrix form, we may assume $\alpha = \alpha_1 = 2\alpha_2, \beta = \beta_1 = \beta_2$. Therefore, we can convert the optimization problem over an asymmetric weight matrix $\boldsymbol{W}$ into an equivalent Problem (17) over an undirected graph ($\frac{1}{2}(\boldsymbol{W} + \boldsymbol{W}^\intercal)$) for convenience.

$$\min_{\boldsymbol{W} \in \mathbb{R}_+^{n \times n}} \|\boldsymbol{W} \odot \boldsymbol{Z}\|_1 - \alpha \mathbf{1}^\intercal \log(\boldsymbol{W}\mathbf{1}) + \beta\|\boldsymbol{W}\|_F^2$$
$$\text{s.t.} \quad \boldsymbol{W}_{ll} = \boldsymbol{O}. \tag{17}$$

Following common practice [18, 44], we suppose that there exists a ground-truth Laplacian matrix $\boldsymbol{L}^*$ that admits the eigendecomposition $\boldsymbol{L}^* = \boldsymbol{\chi}\boldsymbol{\Lambda}^*\boldsymbol{\chi}^\intercal$. Further, the channel of node feature $\boldsymbol{X}_i$ ($i$-th column of $\boldsymbol{X}$) is independently and identically (i.i.d.) generated by the factor analysis model as $\boldsymbol{X}_i = \boldsymbol{\chi}\boldsymbol{h} + \boldsymbol{u}_i + \boldsymbol{\delta}$, where $\boldsymbol{h} \sim \mathcal{N}(\mathbf{0}, \boldsymbol{\Lambda}^{*\dagger})$ is the hidden variable, $\boldsymbol{u}_x$ is the mean of $\boldsymbol{X}_i$, and $\boldsymbol{\delta} \sim \mathcal{N}(\mathbf{0}, \delta_\epsilon^2 \boldsymbol{I})$ is the Gaussian noise with mean zero and covariance $\delta_\epsilon^2 \boldsymbol{I}$. We can further rewrite Problem (17) in terms of $\boldsymbol{L}$ since there exists a bijective mapping between the weight matrix $\boldsymbol{W}$ and the Laplacian matrix $\boldsymbol{L}$ for undirected graphs. Namely, we have $\|\boldsymbol{W} \odot \boldsymbol{Z}\|_1 = 2\text{Tr}(\boldsymbol{X}^\intercal \boldsymbol{L}\boldsymbol{X}) = 2d\sum_{i=1}^{d}\boldsymbol{X}_i^\intercal \boldsymbol{L}\boldsymbol{X}_i$, and there exists a function $R : \mathbb{S}_+^n \to \mathbb{R}$ such that $R(\boldsymbol{L}) = -\alpha\mathbf{1}^\intercal \log(\boldsymbol{W}\mathbf{1}) + \beta\|\boldsymbol{W}\|_F^2$. To get rid of the equality constraint, we transform the original problem into a bi-level optimization problem by introducing the ambiguity set $\mathcal{A}$.

$$\min_{\boldsymbol{L} \in \mathcal{S}_+^n} \left\{ \max_{q \in \mathcal{A}} \mathbb{E}_{\boldsymbol{X}_i \sim q(\boldsymbol{X}_i)}[\boldsymbol{X}_i^\intercal \boldsymbol{L}\boldsymbol{X}_i] + R(\boldsymbol{L}) \right\}, \tag{18}$$

where $\mathcal{A} = \{q \in \mathcal{P}(\boldsymbol{\mu}, \boldsymbol{\Sigma}) \mid (\boldsymbol{\mu} - \hat{\boldsymbol{\mu}}_d)^\intercal \boldsymbol{L}(\boldsymbol{\mu} - \hat{\boldsymbol{\mu}}_d) \leq \rho_1, \|\boldsymbol{\Sigma} - \hat{\boldsymbol{\Sigma}}_d\|_F^2 \leq \rho_2, \boldsymbol{\mu} \in \mathbb{R}^n, \boldsymbol{\Sigma} \in \mathbb{S}_+^n\}$. Here, $\mathcal{P}(\boldsymbol{\mu}, \boldsymbol{\Sigma})$ denotes the set of probability distributions with mean $\boldsymbol{\mu} \in \mathbb{R}^n$ and covariance $\boldsymbol{\Sigma} \in \mathbb{S}_+^n$. $\hat{\boldsymbol{\mu}}_d = \frac{1}{d}\sum_{i=1}^{d}\boldsymbol{X}_i$, $\hat{\boldsymbol{\Sigma}}_d = \frac{1}{d}\sum_{i=1}^{d}(\boldsymbol{X}_i - \hat{\boldsymbol{\mu}}_d)(\boldsymbol{X}_i - \hat{\boldsymbol{\mu}}_d)^\intercal$ are the empirical mean and empirical covariance of the observed features, respectively.

We can show that, with high probability, the ambiguity set $\mathcal{A}$ contains the ground-truth distribution $p(\boldsymbol{X}_i)$ with mean $\boldsymbol{\mu}^*$ and covariance $\Sigma^*$ when choosing $\rho_1, \rho_2$ according to Theorem 4 and 5. These results are directly adapted from [60].

**Theorem 4.** *For $\forall \xi \in (0, \frac{1}{e^2})$ and let $\rho_1 = \frac{c_1}{d} \ln^2(\frac{1}{\xi})$ with the constant $c_1 > 0$, we have,*

$$\mathbb{P}\left[(\boldsymbol{\mu}^* - \hat{\boldsymbol{\mu}}_d)^\mathsf{T} \boldsymbol{L} (\boldsymbol{\mu}^* - \hat{\boldsymbol{\mu}}_d) > \rho_1\right] \leq \xi.$$

**Theorem 5.** *For $\forall \xi \in (0, \frac{1}{e^2})$ and we set $\rho_2$ properly as $\rho_2 = \frac{c_2}{\sqrt{d}} \ln^{3/2}(\frac{4d^{3/2}}{\xi}) \|\Sigma^*\|_F + \frac{c_3}{d} \ln^2(\frac{2}{\xi})$ with constants $c_2 > 0, c_3 > 0$, we have,*

$$\mathbb{P}\left[\|\boldsymbol{\Sigma}^* - \hat{\boldsymbol{\Sigma}}_d\|_F > \rho_2\right] \leq \xi.$$

Theorem 4 and 5 guarantee that the hidden ground-truth distribution of the node feature will be contained in the introduced ambiguity set. Even if there exists some noise in the observed features, we can still recover the ground-truth distribution as long as the number of node feature channels $d$ is sufficiently large ($\rho_1 \to 0, \rho_2 \to 0$ when $d \to \infty$). However, previous works [29, 18] only use the empirical distribution $\hat{p}$ of the node feature as $\mathbb{E}_{\boldsymbol{X}_i \sim \hat{p}(\boldsymbol{X}_i)}[\boldsymbol{X}_i^\mathsf{T} \boldsymbol{L} \boldsymbol{X}_i]$, which may suffer from the issue of noise. Because they tend to yield an affinity graph where the given noisy node features $\{\boldsymbol{X}_i\}_{i=1}^d$ are smooth, but the clean node features generated by the actual hidden distribution $p$ are not. Our proposed block-wise graph learning framework can mitigate this problem by inexplicitly manipulating an approximation distribution $q$ of ground-truth $p$ in a relaxed ambiguity set. Consequently, from Theorem 4 and 5, the robustness of our method with respect to noisy node features is guaranteed with high probability.

Theorem 4 and 5 can be viewed as a theoretical interpretation or analysis of our proposed method when the underlying data generation model is assumed to be the factor analysis model. Under these assumptions, BAGL can guarantee that the hidden ground-truth distribution of the sample feature will be contained in an introduced ambiguity set. Even if there exists some noise in the observed features, we can still recover the ground-truth distribution as long as the number of node feature channels is sufficiently large.

We further validate the robustness to node feature noise in the proposed method via empirical experiments, where we add different levels of Gaussian noise to the given node features. More details can be found in Appendix G.3.

### D.2 Generalization Bond Improvement Interpretation

We further investigate how the proposed optimal graph structure can facilitate the downstream label inference algorithm through the lens of expected generalization error. In the transductive semi-supervised learning setting, we care about the generalization ability of the algorithm on the unlabeled data only, unlike the classic supervised setting. Therefore, we use the expected generalization error for theoretical analysis. In other words, we are interested in the generalization behavior of the learning algorithm compared to a properly defined optimal risk $\min_{\boldsymbol{F}} Q(\boldsymbol{F})$. Formally, we have Theorem 6 as follows. Note that, Theorem 6 can be viewed as an adaptation of Theorem 1 in [28] by instantiating the loss function as the least square loss. Theorem 1 in [28] holds for any affinity graph constructed in GSSL. We thereby follow the same assumptions regarding the label distributions as those made in [28] and omit the details here.

**Theorem 6.** *If the label inference algorithm in (1) is performed on the optimal graph structure proposed in Proposition 1, the expected generalization error over the labeled samples index set $D_l$, uniformly drawn without replacement from $D_n = \{1, 2, \cdots, n\}$, can be upper bounded as*

$$\mathbb{E}_{D_l}\left[\frac{1}{u} \sum_{i \in D_u} \mathbb{I}_{\hat{y}_i \neq y_i}\right] \leq 16 \min_{\boldsymbol{F}} \{Q(\boldsymbol{F})\} + \frac{\mathrm{Tr}(\boldsymbol{\Lambda}) \mathrm{Tr}(\boldsymbol{S}^{-1})}{nl}, \tag{19}$$

*with $D_u = D_n - D_l$, $Q(\boldsymbol{F}) = \mathrm{Tr}\left(\boldsymbol{F}^\mathsf{T} s(\boldsymbol{W}^*) \boldsymbol{F}\right) + \mathrm{Tr}\left((\boldsymbol{F} - \boldsymbol{Y})^\mathsf{T} \boldsymbol{\Lambda}(\boldsymbol{F} - \boldsymbol{Y})\right)$.*

Theorem 6 reveals that our method could enjoy better generalization capability due to a tighter error upper bound, when compared with other graph construction methods without the optimal structure. It analyzes the generalization capability of the fixed label inference algorithm on different graph construction methods. If we use our proposed graph construction method, we have the upper bound in Theorem 6. Indeed, we can also obtain an almost identical upper bound in Eq. (19) when other graph construction methods, like kNN graph, are employed as input for the label inference step, but $Q(\boldsymbol{F})$ changes to $Q'(\boldsymbol{F}) = \mathrm{Tr}\left(\boldsymbol{F}^\mathsf{T} s(\boldsymbol{W}) \boldsymbol{F}\right) + \mathrm{Tr}\left((\boldsymbol{F} - \boldsymbol{Y})^\mathsf{T} \boldsymbol{\Lambda}(\boldsymbol{F} - \boldsymbol{Y})\right)$. Note that the

additional second term is unchanged. Based on the definition of $\boldsymbol{W}^*$ in Definition 1, it is easy to see $Q(\boldsymbol{F}) \leq Q'(\boldsymbol{F})$ thanks to the optimality of $\boldsymbol{W}^*$. Hence, our proposed optimal structure can induce a tighter generalization error bound for GSSL methods, providing a stronger guarantee for the learning performance.

## E   Proofs

We present the detailed proofs in this section and some of them are adapted from [47, 53, 60, 28].

### E.1   Proof of Theorem 1

*Proof.* From Definition 1, we know
$$\boldsymbol{F} = (s(\boldsymbol{W}) + \boldsymbol{\Lambda})^{-1} \boldsymbol{\Lambda} \boldsymbol{Y},$$
so we can rearrange it and get
$$(s(\boldsymbol{W}) + \boldsymbol{\Lambda}) \boldsymbol{F} = \boldsymbol{\Lambda} \boldsymbol{Y},$$
and thus,
$$\boldsymbol{F}^{\mathsf{T}} s(\boldsymbol{W}) \boldsymbol{F} = \boldsymbol{F}^{\mathsf{T}} \boldsymbol{\Lambda} \boldsymbol{Y} - \boldsymbol{F}^{\mathsf{T}} \boldsymbol{\Lambda} \boldsymbol{F}. \tag{20}$$
By plugging Eq. (20) into Eq. (3), we can simplify Eq. (3) as
$$\begin{aligned}
\boldsymbol{W}^* &= \arg\min_{\boldsymbol{W}} \left\{ \mathrm{Tr}(\boldsymbol{F}^{\mathsf{T}} s(\boldsymbol{W}) \boldsymbol{F}) + \mathrm{Tr}\left((\boldsymbol{F} - \boldsymbol{Y})^{\mathsf{T}} \boldsymbol{\Lambda}(\boldsymbol{F} - \boldsymbol{Y})\right) \right\} \\
&= \arg\min_{\boldsymbol{W}} \left\{ \mathrm{Tr}\left(\boldsymbol{F}^{\mathsf{T}} \boldsymbol{\Lambda} \boldsymbol{Y} - \boldsymbol{F}^{\mathsf{T}} \boldsymbol{\Lambda} \boldsymbol{F}\right) + \mathrm{Tr}\left((\boldsymbol{F} - \boldsymbol{Y})^{\mathsf{T}} \boldsymbol{\Lambda}(\boldsymbol{F} - \boldsymbol{Y})\right) \right\} \\
&= \arg\min_{\boldsymbol{W}} \left\{ \mathrm{Tr}(\boldsymbol{F}^{\mathsf{T}} \boldsymbol{\Lambda} \boldsymbol{Y} - \boldsymbol{F}^{\mathsf{T}} \boldsymbol{\Lambda} \boldsymbol{F} + \boldsymbol{F}^{\mathsf{T}} \boldsymbol{\Lambda} \boldsymbol{F} - \boldsymbol{F}^{\mathsf{T}} \boldsymbol{\Lambda} \boldsymbol{Y} - \boldsymbol{Y}^{\mathsf{T}} \boldsymbol{\Lambda} \boldsymbol{F} + \boldsymbol{Y}^{\mathsf{T}} \boldsymbol{\Lambda} \boldsymbol{Y}) \right\} \\
&= \arg\min_{\boldsymbol{W}} \left\{ \mathrm{Tr}(\boldsymbol{Y}^{\mathsf{T}} \boldsymbol{\Lambda} \boldsymbol{Y}) - \mathrm{Tr}(\boldsymbol{Y}^{\mathsf{T}} \boldsymbol{\Lambda} \boldsymbol{F}) \right\}.
\end{aligned} \tag{21}$$
Note that the first term $\mathrm{Tr}(\boldsymbol{Y}^{\mathsf{T}} \boldsymbol{\Lambda} \boldsymbol{Y})$ is irrelevant to $\boldsymbol{W}$, and thus we have,
$$\boldsymbol{W}^* = \arg\max_{\boldsymbol{W}} \mathrm{Tr}(\boldsymbol{Y}^{\mathsf{T}} \boldsymbol{\Lambda} \boldsymbol{F}), \tag{22}$$
with $\boldsymbol{F} = (s(\boldsymbol{W}) + \boldsymbol{\Lambda})^{-1} \boldsymbol{\Lambda} \boldsymbol{Y}$.

By decomposing $\boldsymbol{Y}$ as $\boldsymbol{Y}^{\mathsf{T}} = (\ \boldsymbol{Y}_l^{\mathsf{T}} \ \ \boldsymbol{Y}_u^{\mathsf{T}} \ )$ with $\boldsymbol{Y}_l \in \{0, 1\}^{l \times c}$ being the first $l$ rows of $\boldsymbol{Y}$ and $\boldsymbol{Y}_u \in \{0, 1\}^{u \times c}$ being the last $u$ rows of $\boldsymbol{Y}$. Similarly, we have $\begin{pmatrix} \boldsymbol{\Lambda}_l & \boldsymbol{O} \\ \boldsymbol{O} & \boldsymbol{\Lambda}_u \end{pmatrix}$ and $\boldsymbol{F} = \begin{pmatrix} \boldsymbol{F}_l \\ \boldsymbol{F}_u \end{pmatrix}$. Also recall that we initialize $\boldsymbol{Y}_u = \boldsymbol{O}$, so we further simplify Eq. (22) as
$$\boldsymbol{W}^* = \arg\max_{\boldsymbol{W}} \mathrm{Tr}(\boldsymbol{Y}_l^{\mathsf{T}} \boldsymbol{\Lambda}_l \boldsymbol{F}_l). \tag{23}$$
From Eq. (23), if we recall that $\boldsymbol{F} \in [0, 1]^{n \times c}$ is a normalized predicted probability score matrix according to the definition of the soft label matrix and $\boldsymbol{\Lambda} \in \mathbb{S}_+^n$, we can easily show that $\boldsymbol{W}^*$ is the minimizer if and only if
$$\boldsymbol{Y}_l = \boldsymbol{F}_l,$$
which proves the necessity and the sufficiency of this condition. $\qquad\square$

**Remark.** *Theorem 1 is a necessary and sufficient condition on the optimality definition of $\boldsymbol{W}^*$ in Problem 3. If the affinity graph is optimal, then after we perform the label inference algorithm for GSSL on $\boldsymbol{W}^*$, the predicted soft labels for the labeled nodes would coincide with the ground truth precisely (i.e. $\boldsymbol{Y}_l = \boldsymbol{F}_l^*$, with $\boldsymbol{F}^* = (\boldsymbol{F}_l^*, \boldsymbol{F}_u^*) = (s(\boldsymbol{W}^*) + \boldsymbol{\Lambda})^{-1} \boldsymbol{\Lambda} \boldsymbol{Y}$).*

*The converse of this observation also holds true. If we find that $\boldsymbol{Y}_l = \boldsymbol{F}_l$, when running the label inference algorithm on some affinity graph, then this affinity graph satisfies the optimality definition in Definition 1. Theorem 1 serves as an equivalent statement of Definition 1. The proposed optimal structure in Proposition 1 actually derives from Theorem 1.*

Note that at the very beginning we establish Eq. (20) in which we substitute the regularization term $\boldsymbol{F}^{\mathsf{T}} s(\boldsymbol{W}) \boldsymbol{F}$ with $\boldsymbol{F}^{\mathsf{T}} \boldsymbol{\Lambda} \boldsymbol{Y} - \boldsymbol{F}^{\mathsf{T}} \boldsymbol{\Lambda} \boldsymbol{F}$, which does not include $s(\boldsymbol{W})$. That is how we avoid the appearance of $s(\boldsymbol{W})$ in the later derivation. Note that the starting point is the fact that $\boldsymbol{F}$ is fixed as $\boldsymbol{F} = (s(\boldsymbol{W}) + \boldsymbol{\Lambda})^{-1} \boldsymbol{\Lambda} \boldsymbol{Y}$. Therefore, our proposed optimal structure is irrelevant to the exact forms of $s(\boldsymbol{W})$ or $\boldsymbol{S}$.

## E.2 Proof of Proposition 1

*Proof.* We take the LRC algorithm [82] as an example for simplicity. Proofs for other label inference algorithms under the unified GSSL framework discussed in Appendix B.2 follow in the similar manner.

Note that we have,

$$\boldsymbol{F} = (s(\boldsymbol{W}) + \boldsymbol{\Lambda})^{-1}\boldsymbol{\Lambda}\boldsymbol{Y} = \lambda^2 (\frac{1}{\lambda}\boldsymbol{\mathcal{L}} + \boldsymbol{I})^{-1}\boldsymbol{Y}.$$

By ignoring the irrelevant leading coefficient $\lambda$ since the final prediction $\hat{y}_i = \hat{y}(\boldsymbol{f}_i) = \arg\max_{1 \le j \le c} F_{ij}^*$ is only dependant on the relative value in $\boldsymbol{F}$, and decomposing $(\frac{1}{\lambda}\boldsymbol{\mathcal{L}} + \boldsymbol{I})^{-1}$ as

$$\begin{pmatrix} \boldsymbol{I} + \frac{1}{\lambda}\boldsymbol{D}_{ll}^{-\frac{1}{2}}\left(\boldsymbol{D}_{ll} - \boldsymbol{W}_{ll}\right)\boldsymbol{D}_{ll}^{-\frac{1}{2}} & -\frac{1}{\lambda}\boldsymbol{D}_{ll}^{-\frac{1}{2}}\boldsymbol{W}_{lu}\boldsymbol{D}_{uu}^{-\frac{1}{2}} \\ -\frac{1}{\lambda}\boldsymbol{D}_{uu}^{-\frac{1}{2}}\boldsymbol{W}_{ul}\boldsymbol{D}_{ll}^{-\frac{1}{2}} & \boldsymbol{I} + \frac{1}{\lambda}\boldsymbol{D}_{uu}^{-\frac{1}{2}}\left(\boldsymbol{D}_{uu} - \boldsymbol{W}_{uu}\right)\boldsymbol{D}_{uu}^{-\frac{1}{2}} \end{pmatrix},$$

we can get,

$$\left(\boldsymbol{I} + \frac{1}{\lambda}\boldsymbol{D}_{ll}^{-\frac{1}{2}}\left(\boldsymbol{D}_{ll} - \boldsymbol{W}_{ll}\right)\boldsymbol{D}_{ll}^{-\frac{1}{2}}\right)\boldsymbol{F}_l - \frac{1}{\lambda}\boldsymbol{D}_{ll}^{-\frac{1}{2}}\boldsymbol{W}_{lu}\boldsymbol{D}_{uu}^{-\frac{1}{2}}\boldsymbol{F}_u = \boldsymbol{Y}_l.$$

Recall that if $\boldsymbol{W}$ is optimal in Definition 1, based on Theorem 1, we immediately have

$$\boldsymbol{D}_{ll}^{-\frac{1}{2}}\left(\boldsymbol{D}_{ll} - \boldsymbol{W}_{ll}\right)\boldsymbol{D}_{ll}^{-\frac{1}{2}}\boldsymbol{Y}_l = \boldsymbol{D}_{ll}^{-\frac{1}{2}}\boldsymbol{W}_{lu}\boldsymbol{D}_{uu}^{-\frac{1}{2}}\boldsymbol{F}_u. \tag{24}$$

A naive solution of Eq. (24) is to set both $\boldsymbol{W}_{ll} = \boldsymbol{O}$, and $\boldsymbol{W}_{lu} = \boldsymbol{O}$, which yields the optimal graph structure for $\boldsymbol{W}^*$ as

$$\boldsymbol{W}^* = \begin{pmatrix} \boldsymbol{O} & \boldsymbol{O} \\ \boldsymbol{W}_{ul} & \boldsymbol{W}_{uu} \end{pmatrix}.$$

$\square$

**Remark.** *From this lemma and the fact* $\boldsymbol{F} = \lambda^2 (\frac{1}{\lambda}\boldsymbol{\mathcal{L}} + \boldsymbol{I})^{-1}\boldsymbol{Y}$ *at the beginning of the proof, We can also easily show that under the optimal structure of $\boldsymbol{W}^*$, the predicted soft label matrix for has become*

$$\boldsymbol{F}_u = (\boldsymbol{I} - \mu\boldsymbol{W}_{uu})^{-1}\boldsymbol{W}_{ul}\boldsymbol{Y}_L,$$

*when absorbing all irrelevant coefficients into some constant $\mu$.*

We review the logic of Proposition 1 again. We first start from Problem (3) in Definition 1 to get an ad-hoc solution in Eq. (4), which is optimal under Definition 1. Note that Eq. (4) only specifies how optimal graph structure should be like (partitioned into four blocks according to the label indices with two zeros) and the exact entries in each block ($\boldsymbol{W}_{ul}, \boldsymbol{W}_{uu}$) of Eq. (4) will be further optimized. Hence, we have Eq.(7)-(8) to optimize the exact weights in each block. Note that Problem (5) and Problem (6) strictly follow the optimal structure in Eq. (4) by splitting it into two similar problems.

Some may wonder whether the asymmetric structure of $\boldsymbol{W}^*$ may lead to unclear positive semidefinite definition of the resulted smoothing matrix $s(\boldsymbol{W}^*)$ or $\boldsymbol{S}$. In fact, we do not even care about the exact definition of $s(\cdot)$ or $\boldsymbol{S}$ and the positive semi-definiteness of $\boldsymbol{S}$ when deriving an ad-hoc solution of Problem (3) in Definition 1 as long as $\boldsymbol{F}$ is fixed as $\boldsymbol{F} = (s(\boldsymbol{W}) + \boldsymbol{\Lambda})^{-1}\boldsymbol{\Lambda}\boldsymbol{Y}$. Note that we do not figure out all the optimal solutions to Problem (3). Instead, we only give an ad-hoc solution in Eq. (4) that does not depend on specific forms of $\boldsymbol{S}$ as long as $\boldsymbol{F}$ is fixed as $\boldsymbol{F} = (s(\boldsymbol{W}) + \boldsymbol{\Lambda})^{-1}\boldsymbol{\Lambda}\boldsymbol{Y}$. In this way, we can subtly circumvent the issue of positive semi-definiteness of $\boldsymbol{S}$ or the exact forms of $\boldsymbol{S}$.

## E.3 Proof of Dual Problem Derivation

**Proposition 2.** *The dual problem of Problem (7) is given as $\min_{\boldsymbol{\lambda}} F(\boldsymbol{\lambda}) + G(\boldsymbol{\lambda})$ equivalently, where $F(\boldsymbol{\lambda}) = f^*(\boldsymbol{T}^\intercal\boldsymbol{\lambda})$ and $G(\boldsymbol{\lambda}) = g^*(-\boldsymbol{\lambda})$.*

*Proof.* The Lagrangian function with the Lagrangian multipliers $\boldsymbol{\lambda} \in \mathbb{R}^n$ is given as $\mathcal{L}(\boldsymbol{w}, \boldsymbol{v}, \boldsymbol{\lambda}) = f(\boldsymbol{w}) + g(\boldsymbol{v}) - \langle \boldsymbol{\lambda}, \boldsymbol{Tw} - \boldsymbol{v} \rangle$. Therefore, the dual problem can be written as

$$
\begin{aligned}
&\max_{\boldsymbol{\lambda}} \min_{\boldsymbol{w}, \boldsymbol{v}} \mathcal{L}(\boldsymbol{w}, \boldsymbol{v}, \boldsymbol{\lambda}) \\
&= \max_{\boldsymbol{\lambda}} - \{ \max_{\boldsymbol{w}, \boldsymbol{v}} -\mathcal{L}(\boldsymbol{w}, \boldsymbol{v}, \boldsymbol{\lambda}) \} \\
&= \max_{\boldsymbol{\lambda}} - \left\{ \max_{\boldsymbol{w}} \{ \langle \boldsymbol{T}^\mathsf{T} \boldsymbol{\lambda}, \boldsymbol{w} \rangle - f(\boldsymbol{w}) \} + \max_{\boldsymbol{v}} \{ \langle -\boldsymbol{\lambda}, \boldsymbol{v} \rangle - g(\boldsymbol{v}) \} \right\} \\
&= \max_{\boldsymbol{\lambda}} - \{ f^*(\boldsymbol{T}^\mathsf{T} \boldsymbol{\lambda}) + g^*(-\boldsymbol{\lambda}) \} \\
&= - \min_{\boldsymbol{\lambda}} \{ f^*(\boldsymbol{T}^\mathsf{T} \boldsymbol{\lambda}) + g^*(-\boldsymbol{\lambda}) \}.
\end{aligned}
$$

$\square$

## E.4 Proof of Properties of Objective Functions

**Lemma 1.** $f(\boldsymbol{w}) = \boldsymbol{w}^\mathsf{T} \boldsymbol{z} + \beta \|\boldsymbol{w}\|_2^2 + \mathbb{I}_{\{\boldsymbol{w} \geq 0\}}$ *is $2\beta$-strongly convex.*

*Proof.* Since $\beta \|\boldsymbol{w}\|_2^2$ is twice continuously differentiable and $\nabla^2 \beta \|\boldsymbol{w}\|_2^2 = 2\beta$. Thus, $\beta \|\boldsymbol{w}\|_2^2$ is strongly convex with the parameter $2\beta$. Note that $\boldsymbol{w}^\mathsf{T} \boldsymbol{z} + \mathbb{I}_{\{\boldsymbol{w} \geq 0\}}$ is also convex. Then by the definition of the convexity and strong convexity, it is easy to see that $f(\boldsymbol{w})$ is strongly convex with the parameter $2\beta$. $\square$

**Lemma 2.** $F(\boldsymbol{\lambda}) = f^*(\boldsymbol{T}^\mathsf{T} \boldsymbol{\lambda})$ *has the following properties.*

*(1) $F(\boldsymbol{\lambda})$ is differentiable and $\nabla F(\boldsymbol{\lambda}) = \boldsymbol{T} \boldsymbol{x}^*$, where*

$$
\boldsymbol{x}^* = \arg\max_{\boldsymbol{x}} \{ (\boldsymbol{T}^\mathsf{T} \boldsymbol{\lambda})^\mathsf{T} \boldsymbol{x} - f(\boldsymbol{x}) \}
$$

.

*(2) $\nabla F(\boldsymbol{\lambda})$ is Lipschitz continuous with parameter $L = \frac{l+u}{2\beta}$.*

*Proof.*     (1) We know that

$$
\boldsymbol{x}^* = \arg\max_{\boldsymbol{x}} \{ (\boldsymbol{T}^\mathsf{T} \boldsymbol{\lambda})^\mathsf{T} \boldsymbol{x} - f(\boldsymbol{x}) \}
$$

if and only if

$$
\boldsymbol{T}^\mathsf{T} \boldsymbol{\lambda} \in \partial f(\boldsymbol{x}^*),
$$

which is equivalent to

$$
\boldsymbol{x}^* \in \partial f^*(\boldsymbol{T}^\mathsf{T} \boldsymbol{\lambda}).
$$

Since $\boldsymbol{x}^*$ is a unique maximizer, $\partial f^*(\boldsymbol{T}^\mathsf{T} \boldsymbol{\lambda})$ contains only one element, which means

$$
\partial f^*(\boldsymbol{T}^\mathsf{T} \boldsymbol{\lambda}) = \{ \nabla f^*(\boldsymbol{T}^\mathsf{T} \boldsymbol{\lambda}) \} = \{ \boldsymbol{x}^* \}.
$$

Therefore, this implies $F(\boldsymbol{\lambda})$ is differentiable and

$$
\nabla F(\boldsymbol{\lambda}) = \boldsymbol{T} \nabla f^*(\boldsymbol{T}^\mathsf{T} \boldsymbol{\lambda}) = \boldsymbol{T} \boldsymbol{x}^*.
$$

(2) From Lemma 1, we know $f(\boldsymbol{w})$ is $2\beta$-strongly convex. Thus, we have, $\forall \boldsymbol{x}, \boldsymbol{x}', \boldsymbol{T}^\mathsf{T} \boldsymbol{\lambda} \in \partial f(\boldsymbol{x}), \boldsymbol{T}^\mathsf{T} \boldsymbol{\lambda}' \in \partial f(\boldsymbol{x}')$,

$$
(\boldsymbol{T}^\mathsf{T} \boldsymbol{\lambda} - \boldsymbol{T}^\mathsf{T} \boldsymbol{\lambda}')^\mathsf{T} (\boldsymbol{x} - \boldsymbol{x}') \geq 2\beta \|\boldsymbol{x} - \boldsymbol{x}'\|_2^2. \tag{25}
$$

Also we have $\boldsymbol{x} \in \partial f^*(\boldsymbol{T}^\mathsf{T} \boldsymbol{\lambda})$ and $\boldsymbol{x}' \in \partial f^*(\boldsymbol{T}^\mathsf{T} \boldsymbol{\lambda}')$. By (1), we must further have $\boldsymbol{x} = \nabla f^*(\boldsymbol{T}^\mathsf{T} \boldsymbol{\lambda})$ and $\boldsymbol{x}' = \nabla f^*(\boldsymbol{T}^\mathsf{T} \boldsymbol{\lambda}')$. Substituting this into (25) gives

$$
\begin{aligned}
&(\boldsymbol{T}^\mathsf{T} \boldsymbol{\lambda} - \boldsymbol{T}^\mathsf{T} \boldsymbol{\lambda}')^\mathsf{T} (\nabla f^*(\boldsymbol{T}^\mathsf{T} \boldsymbol{\lambda}) - \nabla f^*(\boldsymbol{T}^\mathsf{T} \boldsymbol{\lambda}')) \\
&\geq 2\beta \| \nabla f^*(\boldsymbol{T}^\mathsf{T} \boldsymbol{\lambda}) - \nabla f^*(\boldsymbol{T}^\mathsf{T} \boldsymbol{\lambda}') \|_2^2.
\end{aligned} \tag{26}
$$

By Cauchy-Schwarz inequality, we have

$$
\begin{aligned}
&\| \nabla f^*(\boldsymbol{T}^\mathsf{T} \boldsymbol{\lambda}) - \nabla f^*(\boldsymbol{T}^\mathsf{T} \boldsymbol{\lambda}') \|_2 \cdot \| \boldsymbol{T}^\mathsf{T} \boldsymbol{\lambda} - \boldsymbol{T}^\mathsf{T} \boldsymbol{\lambda}' \|_2 \\
&\geq (\boldsymbol{T}^\mathsf{T} \boldsymbol{\lambda} - \boldsymbol{T}^\mathsf{T} \boldsymbol{\lambda}')^\mathsf{T} (\nabla f^*(\boldsymbol{T}^\mathsf{T} \boldsymbol{\lambda}) - \nabla f^*(\boldsymbol{T}^\mathsf{T} \boldsymbol{\lambda}')) \\
&\geq 2\beta \| \nabla f^*(\boldsymbol{T}^\mathsf{T} \boldsymbol{\lambda}) - \nabla f^*(\boldsymbol{T}^\mathsf{T} \boldsymbol{\lambda}') \|_2^2.
\end{aligned}
$$

Also note that $\|T\|_2$ is the spectral norm of $T$, we then obtain

$$\|\nabla f^*(T^\mathsf{T}\lambda) - \nabla f^*(T^\mathsf{T}\lambda')\|_2$$
$$\leq \frac{1}{2\beta}\|T^\mathsf{T}\lambda - T^\mathsf{T}\lambda'\|_2 \tag{27}$$
$$\leq \frac{\|T^\mathsf{T}\|_2}{2\beta}\|\lambda - \lambda'\|_2.$$

Further, we have

$$\|\nabla F(\lambda) - \nabla F(\lambda')\|_2$$
$$= \|T\nabla f^*(T^\mathsf{T}\lambda) - T\nabla f^*(T^\mathsf{T}\lambda')\|_2 \tag{28}$$
$$\leq \frac{\|T\|_2^2}{2\beta}\|\lambda - \lambda'\|_2.$$

Recall that we have $W\mathbf{1} = T_1 w$, which denotes that $T$ simply maps the vector $w$ to the out-degree vector of the weight matrix. Then it is easy to see

$$T_1 T_1^\mathsf{T} = lI.$$

Similarly, we have

$$T_2 T_2^\mathsf{T} = uI.$$

Recall that

$$T = \left( \begin{array}{c} T_1 \\ T_2 \end{array} \right),$$

and we can easily get

$$TT^\mathsf{T} = \left( \begin{array}{cc} lI_u & \mathbf{1}_u\mathbf{1}_l^\mathsf{T} \\ \mathbf{1}_l\mathbf{1}_u^\mathsf{T} & uI_l \end{array} \right).$$

So now we know its eigenvalues $\lambda$ can be then computed as the roots of the characteristic polynomial as

$$\det(TT^\mathsf{T} - \lambda I)$$
$$= \det((l - \lambda)I_u) \det\left( (u - \lambda)I_l - \mathbf{1}_l\mathbf{1}_u^\mathsf{T}(\frac{1}{l - \lambda}I_u)\mathbf{1}_u\mathbf{1}_l^\mathsf{T} \right)$$
$$= (l - \lambda)^u \det\left( (u - \lambda)I_l - \frac{u}{l - \lambda}\mathbf{1}_l\mathbf{1}_l^\mathsf{T} \right)$$
$$= (l - \lambda)^u(u - \lambda)^l \det(I_l - \frac{u}{(u - \lambda)(l - \lambda)}\mathbf{1}_l\mathbf{1}_l^\mathsf{T})$$
$$= (l - \lambda)^u(u - \lambda)^l \left( 1 + \mathrm{Tr}(\frac{-u}{(u - \lambda)(l - \lambda)}\mathbf{1}_l\mathbf{1}_l^\mathsf{T}) \right)$$
$$+ (l - \lambda)^u(u - \lambda)^l \det(\frac{-u}{(u - \lambda)(l - \lambda)}\mathbf{1}_l\mathbf{1}_l^\mathsf{T})$$
$$= (l - \lambda)^u(u - \lambda)^l(1 + \frac{-ul}{(u - \lambda)(l - \lambda)})$$
$$= \lambda(l - \lambda)^{u-1}(u - \lambda)^{l-1}(\lambda - l - u).$$

Let $\det(TT^\mathsf{T} - \lambda I) = 0$ and assume $u > l$, we immediately know $\lambda_1 = l + u > \lambda_2 = \cdots = \lambda_l = u > \lambda_{l+1} = \cdots = \lambda_{n-1} = l > \lambda_n = 0$. Therefore, $\|T\|_2^2 = \lambda_1 = l + u$. By substituting this into Eq. (28), we finally have

$$\|\nabla F(\lambda) - \nabla F(\lambda')\|_2 \leq \frac{l + u}{2\beta}\|\lambda - \lambda'\|_2.$$

$\square$

## E.5 Proof of $\bar{w}^k$ Derivation

$$\bar{w}^k = \arg\max_{w} \left\{ (T^\mathsf{T} \mu^k)^\mathsf{T} w - f(w) \right\}$$

$$= \arg\min_{w} \left\{ w^\mathsf{T} z + \beta \|w\|_2^2 - (T^\mathsf{T} \mu^k)^\mathsf{T} w + \mathbb{I}_{\{w \geq 0\}} \right\}$$

$$= \arg\min_{w} \left\{ \left\| w + \frac{z - T^\mathsf{T} \mu^k}{2\beta} \right\|^2 + \mathbb{I}_{\{w \geq 0\}} \right\}$$

$$= \arg\min_{w} \left\{ \frac{1}{2} \left\| w - \frac{T^\mathsf{T} \mu^k - z}{2\beta} \right\|^2 + \frac{1}{2} \mathbb{I}_{\{w \geq 0\}} \right\}$$

$$= \mathrm{prox}_{\frac{1}{2} \mathbb{I}_{\{w \geq 0\}}} \left( \frac{T^\mathsf{T} \mu^k - z}{2\beta} \right)$$

$$= P_{\mathbb{R}_+} \left( \frac{T^\mathsf{T} \mu^k - z}{2\beta} \right)$$

$$= \left[ \frac{T^\mathsf{T} \mu^k - z}{2\beta} \right]_+ .$$

## E.6 Proof of Updating Step of Dual Variables

**Proposition 3.** *The updating step* $\lambda^k = \mathrm{prox}_{tG} \left( \mu^k - t \nabla F(\mu^k) \right)$ *can be written equivalently as the following three steps.*

$$\bar{w}^k = \left[ \frac{T^\mathsf{T} \mu^k - z}{2\beta} \right]_+ ,$$

$$\bar{u}^k = \frac{(T\bar{w}^k - t^{-1}\mu^k) + \sqrt{(T\bar{w}^k - t^{-1}\mu^k) \odot (T\bar{w}^k - t^{-1}\mu^k) + 4\alpha t^{-1}\mathbf{1}}}{2},$$

$$\lambda^k = \mu^k - t(T\bar{w}^k - \bar{u}^k),$$

*where* $[\cdot]_+ = \max(0, \cdot)$ *and* $\odot$ *denotes element-wise multiplication.*

*Proof.* By Lemma 2, we know $\nabla F(\mu^k) = T\bar{w}^k$ with

$$\bar{w}^k = \arg\max_{w} \left\{ (T^\mathsf{T} \mu^k)^\mathsf{T} w - f(w) \right\}$$

$$= \arg\min_{w} \left\{ w^\mathsf{T} z + \beta \|w\|_2^2 - (T^\mathsf{T} \mu^k)^\mathsf{T} w + \mathbb{I}_{\{w \geq 0\}} \right\}$$

$$= \arg\min_{w} \left\{ \left\| w + \frac{z - T^\mathsf{T} \mu^k}{2\beta} \right\|^2 + \mathbb{I}_{\{w \geq 0\}} \right\}$$

$$= \arg\min_{w} \left\{ \frac{1}{2} \left\| w - \frac{T^\mathsf{T} \mu^k - z}{2\beta} \right\|^2 + \frac{1}{2} \mathbb{I}_{\{w \geq 0\}} \right\}$$

$$= \mathrm{prox}_{\frac{1}{2} \mathbb{I}_{\{w \geq 0\}}} \left( \frac{T^\mathsf{T} \mu^k - z}{2\beta} \right)$$

$$= P_{\mathbb{R}_+} \left( \frac{T^\mathsf{T} \mu^k - z}{2\beta} \right)$$

$$= \left[ \frac{T^\mathsf{T} \mu^k - z}{2\beta} \right]_+ .$$

Let $p^k = \mu^k - tT\bar{w}^k$. By the extended Moreau decomposition [43],

$$\mathrm{prox}_{\gamma h}(z) + \gamma \, \mathrm{prox}_{\gamma^{-1} h^*}(z/\gamma) = z, \quad \forall z.$$

We have
$$\operatorname{prox}_{tG}(\boldsymbol{p}^k) = \boldsymbol{p}^k - t\operatorname{prox}_{t^{-1}G^*}(t^{-1}\boldsymbol{p}^k)$$
$$= \boldsymbol{p}^k + t\operatorname{prox}_{t^{-1}g}(-t^{-1}\boldsymbol{p}^k)$$
$$= \boldsymbol{\mu}^k - t\boldsymbol{T}\bar{\boldsymbol{w}}^k + t\operatorname{prox}_{t^{-1}g}(\boldsymbol{T}\bar{\boldsymbol{w}}^k - t^{-1}\boldsymbol{\mu}^k)$$
$$= \boldsymbol{\mu}^k - t(\boldsymbol{T}\bar{\boldsymbol{w}}^k - \bar{\boldsymbol{u}}^k).$$

Here, $\bar{\boldsymbol{u}}^k = \operatorname{prox}_{t^{-1}g}(\boldsymbol{T}\bar{\boldsymbol{w}}^k - t^{-1}\boldsymbol{\mu}^k)$. Note that $g(\boldsymbol{v}) = -\alpha\mathbf{1}^\mathsf{T}\log(\boldsymbol{v})$ and by the definition of proximal mapping, it is easy to prove that

$$\operatorname{prox}_{t^{-1}g}(\boldsymbol{v}) = \frac{\boldsymbol{v} + \sqrt{\boldsymbol{v}\odot\boldsymbol{v} + 4\alpha t^{-1}\mathbf{1}}}{2}.$$

Therefore,

$$\bar{\boldsymbol{u}}^k = \frac{(\boldsymbol{T}\bar{\boldsymbol{w}}^k - t^{-1}\boldsymbol{\mu}^k) + \sqrt{(\boldsymbol{T}\bar{\boldsymbol{w}}^k - t^{-1}\boldsymbol{\mu}^k)\odot(\boldsymbol{T}\bar{\boldsymbol{w}}^k - t^{-1}\boldsymbol{\mu}^k) + 4\alpha t^{-1}\mathbf{1}}}{2},$$
$$\boldsymbol{\lambda}^k = \boldsymbol{\mu}^k - t(\boldsymbol{T}\bar{\boldsymbol{w}}^k - \bar{\boldsymbol{u}}^k),$$

$\square$

### E.7 Proof of Theorem 2

*Proof.* Recall that the dual sequence $\{\boldsymbol{\lambda}^k\}$ is generated by Procedure `GWBI` equivalently as the following iterations.

$$\boldsymbol{\mu}^k = \boldsymbol{\lambda}^{k-1} + \frac{k-2}{k+1}(\boldsymbol{\lambda}^{k-1} - \boldsymbol{\lambda}^{k-2}),$$
$$\boldsymbol{\lambda}^k = \operatorname{prox}_{tG}\left(\boldsymbol{\mu}^k - t\nabla F(\boldsymbol{\mu}^k)\right).$$

Let $\theta^k = \frac{2}{(k+1)}$ and we also introduce another intermediate variable $\boldsymbol{\nu}^k$. Initialize $\boldsymbol{\nu}^0 = \boldsymbol{\lambda}^0$. Then by substituting the expression for $\boldsymbol{\nu}^k$ in formula for $\boldsymbol{\mu}^k$, we can rewrite the iteration in Procedure `GWBI` as

$$\boldsymbol{\mu}^k = (1 - \theta^k)\boldsymbol{\lambda}^{k-1} + \theta^k\boldsymbol{\nu}^{k-1},$$
$$\boldsymbol{\lambda}^k = \operatorname{prox}_{tG}(\boldsymbol{\mu}^k - t\nabla F(\boldsymbol{\mu}^k)),$$
$$\boldsymbol{\nu}^k = \boldsymbol{\lambda}^{k-1} + \frac{1}{\theta^k}(\boldsymbol{\lambda}^k - \boldsymbol{\lambda}^{k-1}).$$

For notation simplicity, we let $\boldsymbol{\lambda}^+ = \boldsymbol{\lambda}^k$, $\boldsymbol{\lambda} = \boldsymbol{\lambda}^{k-1}$, $\boldsymbol{\mu}^+ = \boldsymbol{\mu}^k$, $\boldsymbol{\nu}^+ = \boldsymbol{\nu}^k$, $\boldsymbol{\nu} = \boldsymbol{\nu}^{k-1}$ and $\theta = \theta^k$. Then we have

$$\boldsymbol{\mu}^+ = (1 - \theta)\boldsymbol{\lambda} + \theta\boldsymbol{\nu}, \tag{29}$$
$$\boldsymbol{\lambda}^+ = \operatorname{prox}_{tG}(\boldsymbol{\mu}^+ - t\nabla F(\boldsymbol{\mu}^+)), \tag{30}$$
$$\boldsymbol{\nu} = \boldsymbol{\lambda} + \frac{\boldsymbol{\mu}^+ - \boldsymbol{\lambda}}{\theta}, \tag{31}$$
$$\boldsymbol{\nu}^+ = \boldsymbol{\lambda} + \frac{\boldsymbol{\lambda}^+ - \boldsymbol{\lambda}}{\theta}. \tag{32}$$

We can get the upper bound on $G$ from the definition of proximal mapping.

$$G(\boldsymbol{\lambda}^+) \le G(\boldsymbol{\lambda}') + \boldsymbol{\xi}^\mathsf{T}(\boldsymbol{\lambda}^+ - \boldsymbol{\lambda}') \quad \forall\boldsymbol{\lambda}', \tag{33}$$

where $\boldsymbol{\xi} \in \partial G(\boldsymbol{\lambda}^+)$. Since $\boldsymbol{\lambda}^+ = \operatorname{prox}_{tG}(\boldsymbol{\mu}^+ - t\nabla F(\boldsymbol{\mu}^+))$, then by the subgradient characterization, we have $\boldsymbol{\mu}^+ - t\nabla F(\boldsymbol{\mu}^+) - \boldsymbol{\lambda}^+ \in \partial(tG)(\boldsymbol{\lambda}^+) = t\partial G(\boldsymbol{\lambda}^+)$. So we can get

$$\frac{1}{t}\left(\boldsymbol{\mu}^+ - t\nabla F(\boldsymbol{\mu}^+) - \boldsymbol{\lambda}^+\right) \in \partial G(\boldsymbol{\lambda}^+).$$

From Eq. (33), for all $\boldsymbol{\lambda}'$, we have

$$G(\boldsymbol{\lambda}^+) \le G(\boldsymbol{\lambda}') + \frac{1}{t}\left[\boldsymbol{\mu}^+ - t\nabla F(\boldsymbol{\mu}^+) - \boldsymbol{\lambda}^+\right]^\mathsf{T}(\boldsymbol{\lambda}^+ - \boldsymbol{\lambda}')$$

$$= G(\boldsymbol{\lambda}') + \left[\nabla F(\boldsymbol{\mu}^+) + \frac{1}{t}(\boldsymbol{\lambda}^+ - \boldsymbol{\mu}^+)\right]^\mathsf{T}(\boldsymbol{\lambda}' - \boldsymbol{\lambda}^+)$$

$$= G(\boldsymbol{\lambda}') + \nabla F(\boldsymbol{\mu}^+)^\mathsf{T}(\boldsymbol{\lambda}' - \boldsymbol{\lambda}^+) + \frac{1}{t}(\boldsymbol{\lambda}^+ - \boldsymbol{\mu}^+)^\mathsf{T}(\boldsymbol{\lambda}' - \boldsymbol{\lambda}^+). \tag{34}$$

Recall that $\nabla F(\boldsymbol{\lambda})$ is Lipschitz continuous with parameter $L = \frac{l+u}{2\beta}$ from Lemma 2 and the step size is fixed as $t = \frac{1}{L} = \frac{2\beta}{l+u}$. By the upper bound from Lipschitz property, for all $\boldsymbol{\lambda}'$, we have,

$$F(\boldsymbol{\lambda}^+) = F(\boldsymbol{\lambda}') + \nabla F(\boldsymbol{\lambda}')^\mathsf{T}(\boldsymbol{\lambda}^+ - \boldsymbol{\lambda}')$$

$$+ \int_0^1 (\boldsymbol{\lambda}^+ - \boldsymbol{\lambda}')^\mathsf{T}\left(\nabla F(\boldsymbol{\lambda}' + \tau(\boldsymbol{\lambda}^+ - \boldsymbol{\lambda}')) - \nabla F(\boldsymbol{\lambda}')\right) d\tau$$

$$\le F(\boldsymbol{\lambda}') + \nabla F(\boldsymbol{\lambda}')^\mathsf{T}(\boldsymbol{\lambda}^+ - \boldsymbol{\lambda}') + \frac{L}{2}\|\boldsymbol{\lambda}' - \boldsymbol{\lambda}^+\|_2^2$$

$$= F(\boldsymbol{\lambda}') + \nabla F(\boldsymbol{\lambda}')^\mathsf{T}(\boldsymbol{\lambda}^+ - \boldsymbol{\lambda}') + \frac{1}{2t}\|\boldsymbol{\lambda}' - \boldsymbol{\lambda}^+\|_2^2. \tag{35}$$

Let $\boldsymbol{\lambda}' = \boldsymbol{\mu}^+$ in Eq. (35), we immediately have,

$$F(\boldsymbol{\lambda}^+) \le F(\boldsymbol{\mu}^+) + \nabla F(\boldsymbol{\mu}^+)^\mathsf{T}(\boldsymbol{\lambda}^+ - \boldsymbol{\mu}^+) + \frac{1}{2t}\|\boldsymbol{\mu}^+ - \boldsymbol{\lambda}^+\|_2^2. \tag{36}$$

Combining Eq. (34) and Eq. (36), we can obtain, for $\forall \boldsymbol{\lambda}'$

$$Q(\boldsymbol{\lambda}^+) = F(\boldsymbol{\lambda}^+) + G(\boldsymbol{\lambda}^+)$$

$$\le G(\boldsymbol{\lambda}') + F(\boldsymbol{\mu}^+) + \nabla F(\boldsymbol{\mu}^+)^\mathsf{T}(\boldsymbol{\lambda}' - \boldsymbol{\mu}^+) \tag{37}$$

$$+ \frac{1}{t}(\boldsymbol{\lambda}^+ - \boldsymbol{\mu}^+)^\mathsf{T}(\boldsymbol{\lambda}' - \boldsymbol{\lambda}^+) + \frac{1}{2t}\|\boldsymbol{\mu}^+ - \boldsymbol{\lambda}^+\|_2^2$$

$$\le G(\boldsymbol{\lambda}') + F(\boldsymbol{\lambda}') + \frac{1}{t}(\boldsymbol{\lambda}^+ - \boldsymbol{\mu}^+)^\mathsf{T}(\boldsymbol{\lambda}' - \boldsymbol{\lambda}^+) + \frac{1}{2t}\|\boldsymbol{\mu}^+ - \boldsymbol{\lambda}^+\|_2^2$$

$$= Q(\boldsymbol{\lambda}') + \frac{1}{t}(\boldsymbol{\lambda}^+ - \boldsymbol{\mu}^+)^\mathsf{T}(\boldsymbol{\lambda}' - \boldsymbol{\lambda}^+) + \frac{1}{2t}\|\boldsymbol{\lambda}^+ - \boldsymbol{\mu}^+\|_2^2. \tag{38}$$

Therefore, we have,

$$Q(\boldsymbol{\lambda}^+) - Q(\boldsymbol{\lambda}') \le \frac{1}{t}(\boldsymbol{\lambda}^+ - \boldsymbol{\mu}^+)^\mathsf{T}(\boldsymbol{\lambda}' - \boldsymbol{\lambda}^+) + \frac{1}{2t}\|\boldsymbol{\lambda}^+ - \boldsymbol{\mu}^+\|_2^2 \quad \forall \boldsymbol{\lambda}'. \tag{39}$$

By making a convex combination of the RHS of Eq.( 39) for $\boldsymbol{\lambda}' = \boldsymbol{\lambda}$ and $\boldsymbol{\lambda}' = \boldsymbol{\lambda}^*$. We have,

$$\theta\left(Q(\boldsymbol{\lambda}^+) - Q(\boldsymbol{\lambda}^*)\right) + (1 - \theta)\left(Q(\boldsymbol{\lambda}^+) - Q(\boldsymbol{\lambda})\right)$$

$$= Q(\boldsymbol{\lambda}^+) - \theta Q(\boldsymbol{\lambda}^*) - (1 - \theta)Q(\boldsymbol{\lambda})$$

$$\le \frac{1}{t}(\boldsymbol{\lambda}^+ - \boldsymbol{\mu}^+)^\mathsf{T}\left(\theta\boldsymbol{\lambda}^* + (1 - \theta)\boldsymbol{\lambda} - \boldsymbol{\lambda}^+\right) + \frac{1}{2t}\|\boldsymbol{\lambda}^+ - \boldsymbol{\mu}^+\|_2^2$$

$$= \frac{1}{2t}\left(\|\boldsymbol{\mu}^+ - (1 - \theta)\boldsymbol{\lambda} - \theta\boldsymbol{\lambda}^*\|_2^2 - \|\boldsymbol{\lambda}^+ - (1 - \theta)\boldsymbol{\lambda} - \theta\boldsymbol{\lambda}^*\|_2^2\right)$$

$$= \frac{\theta^2}{2t}\left(\|\boldsymbol{\nu} - \boldsymbol{\lambda}^*\|_2^2 - \|\boldsymbol{\nu}^+ - \boldsymbol{\lambda}^*\|_2^2\right). \tag{40}$$

From Eq.( 40), we have

$$Q(\boldsymbol{\lambda}^+) - \theta Q(\boldsymbol{\lambda}^*) - (1 - \theta)Q(\boldsymbol{\lambda}) \le \frac{\theta^2}{2t}\left(\|\boldsymbol{\nu} - \boldsymbol{\lambda}^*\|_2^2 - \|\boldsymbol{\nu}^+ - \boldsymbol{\lambda}^*\|_2^2\right).$$

Then we can get

$$Q(\boldsymbol{\lambda}^k) - Q(\boldsymbol{\lambda}^*) \le (1 - \theta^k)\left(Q(\boldsymbol{\lambda}^{k-1}) - Q(\boldsymbol{\lambda}^*)\right)$$

$$+ \frac{(\theta^k)^2}{2t}\left(\|\boldsymbol{\nu}^{k-1} - \boldsymbol{\lambda}^*\|_2^2 - \|\boldsymbol{\nu}^k - \boldsymbol{\lambda}^*\|_2^2\right). \tag{41}$$

Thus, we obtain,

$$\frac{t}{(\theta^k)^2}\left(Q(\boldsymbol{\lambda}^k) - Q(\boldsymbol{\lambda}^*)\right) + \frac{1}{2}\|\boldsymbol{\nu}^k - \boldsymbol{\lambda}^*\|_2^2$$

$$\leq \frac{(1-\theta^k)t}{(\theta^k)^2}\left(Q(\boldsymbol{\lambda}^{k-1}) - Q(\boldsymbol{\lambda}^*)\right) + \frac{1}{2}\|\boldsymbol{\nu}^{k-1} - \boldsymbol{\lambda}^*\|_2^2. \tag{42}$$

Moreover, recall that $\theta^k = \frac{2}{k+1}$ with $\theta^1 = 1$, and thus it is easy to prove that

$$\frac{1-\theta^k}{(\theta^k)^2} \leq \frac{1}{(\theta^{k-1})^2} \quad \forall k \geq 2.$$

Therefore, note that $\boldsymbol{\nu}^0 = \boldsymbol{\lambda}^0$ we obtain from Eq. (42),

$$\frac{t}{(\theta^k)^2}\left(Q(\boldsymbol{\lambda}^k) - Q(\boldsymbol{\lambda}^*)\right) + \frac{1}{2}\|\boldsymbol{\nu}^k - \boldsymbol{\lambda}^*\|_2^2$$

$$\leq \frac{(1-\theta^k)t}{(\theta^k)^2}\left(Q(\boldsymbol{\lambda}^{k-1}) - Q(\boldsymbol{\lambda}^*)\right) + \frac{1}{2}\|\boldsymbol{\nu}^{k-1} - \boldsymbol{\lambda}^*\|_2^2$$

$$\leq \frac{t}{(\theta^{k-1})^2}\left(Q(\boldsymbol{\lambda}^{k-1}) - Q(\boldsymbol{\lambda}^*)\right) + \frac{1}{2}\|\boldsymbol{\nu}^{k-1} - \boldsymbol{\lambda}^*\|_2^2$$

$$\leq \cdots$$

$$\leq \frac{(1-\theta^1)t}{(\theta^1)^2}\left(Q(\boldsymbol{\lambda}^0) - Q(\boldsymbol{\lambda}^*)\right) + \frac{1}{2}\|\boldsymbol{\nu}^0 - \boldsymbol{\lambda}^*\|_2^2$$

$$= \frac{1}{2}\|\boldsymbol{\lambda}^0 - \boldsymbol{\lambda}^*\|_2^2.$$

Finally, recall that $t = \frac{2\beta}{l+u}$, and we have

$$Q(\boldsymbol{\lambda}^k) - Q(\boldsymbol{\lambda}^*) \leq \frac{(\theta^k)^2}{2t}\|\boldsymbol{\lambda}^0 - \boldsymbol{\lambda}^*\|_2^2 = \frac{l+u}{\beta(k+1)^2}\|\boldsymbol{\lambda}^0 - \boldsymbol{\lambda}^*\|_2^2.$$

$\square$

## E.8   Proof of Theorem 3

*Proof.* Let us define

$$h_1(\boldsymbol{w}) = f(\boldsymbol{w}) - \langle \boldsymbol{T}^\mathsf{T}\boldsymbol{\lambda}, \boldsymbol{w}\rangle,$$
$$h_2(\boldsymbol{v}) = g(\boldsymbol{v}) + \langle \boldsymbol{\lambda}^k, \boldsymbol{v}\rangle.$$

Similar to the sequence $\boldsymbol{w}^k = \arg\max_{\boldsymbol{w}}\{\langle \boldsymbol{T}^\mathsf{T}\boldsymbol{\lambda}^k\rangle - f(\boldsymbol{w})\}$, we also construct another sequence $\boldsymbol{v}^k = \arg\max_{\boldsymbol{v}}\{\langle -\boldsymbol{\lambda}^k, \boldsymbol{v}\rangle - g(\boldsymbol{v})\}$. Then, we immediately have,

$$\mathcal{L}(\boldsymbol{w}, \boldsymbol{v}, \boldsymbol{\lambda}^k) = h_1(\boldsymbol{w}) + h_2(\boldsymbol{v}) \quad \forall \boldsymbol{w}, \forall \boldsymbol{v}, \tag{43}$$

$$\boldsymbol{w}^k = \arg\min_{\boldsymbol{w}} h_1(\boldsymbol{w}), \tag{44}$$

$$\boldsymbol{v}^k = \arg\min_{\boldsymbol{v}} h_2(\boldsymbol{v}). \tag{45}$$

By Lemma 1, it follows that $h_1(\boldsymbol{w})$ is also $2\beta$-strongly convex. Therefore, by Eq. (44) and the strong convexity of $h_1(\boldsymbol{w})$, we know that

$$h_1(\boldsymbol{w}) - h_1(\boldsymbol{w}^k) \geq \beta\|\boldsymbol{w} - \boldsymbol{w}^k\|_2^2 \quad \forall \boldsymbol{w}. \tag{46}$$

Similarly, by Eq. (45) and the convexity of $h_2(\boldsymbol{v})$, we also know that

$$h_2(\boldsymbol{v}) - h_2(\boldsymbol{v}^k) \geq 0 \quad \forall \boldsymbol{v}. \tag{47}$$

By summing Eq. (43), Eq. (44) and Eq. (45) on both sides, we then obtain

$$\mathcal{L}(\boldsymbol{w}, \boldsymbol{v}, \boldsymbol{\lambda}^k) - \mathcal{L}(\boldsymbol{w}^k, \boldsymbol{v}^k, \boldsymbol{\lambda}^k) \geq \beta\|\boldsymbol{w} - \boldsymbol{w}^k\|_2^2 \quad \forall \boldsymbol{w}, \forall \boldsymbol{v}. \tag{48}$$

By introducing the primal optimal solution $(\boldsymbol{w}^*, \boldsymbol{v}^*)$ and substituting $\boldsymbol{w} = \boldsymbol{w}^*$ and $\boldsymbol{v} = \boldsymbol{v}^* = \boldsymbol{T}\boldsymbol{w}^*$, we get

$$\mathcal{L}(\boldsymbol{w}^*, \boldsymbol{v}^*, \boldsymbol{\lambda}^k) - \mathcal{L}(\boldsymbol{w}^k, \boldsymbol{v}^k, \boldsymbol{\lambda}^k) \geq \beta \|\boldsymbol{w}^* - \boldsymbol{w}^k\|_2^2. \tag{49}$$

Additionally, based on the definition of $\boldsymbol{w}^k$ and $\boldsymbol{v}^k$, we have

$$\begin{aligned}
\mathcal{L}(\boldsymbol{w}^k, \boldsymbol{v}^k, \boldsymbol{\lambda}^k) &= - \left\{ \langle \boldsymbol{T}^\intercal \boldsymbol{\lambda}^k, \boldsymbol{w}^k \rangle - f(\boldsymbol{w}^k) + \langle -\boldsymbol{\lambda}^k, \boldsymbol{v}^k \rangle - g(\boldsymbol{v}^k) \right\} \\
&= -f^*(\boldsymbol{T}^\intercal \boldsymbol{\lambda}^k) - g^*(-\boldsymbol{\lambda}^k) \\
&= -F(\boldsymbol{\lambda}^k) - G(\boldsymbol{\lambda}^k) \\
&= -Q(\boldsymbol{\lambda}^k).
\end{aligned}$$

$$\begin{aligned}
\mathcal{L}(\boldsymbol{w}^*, \boldsymbol{v}^*, \boldsymbol{\lambda}^k) &= f(\boldsymbol{w}^*) + g(\boldsymbol{v}^*) - \langle \boldsymbol{\lambda}^k, \boldsymbol{T}\boldsymbol{w}^* - \boldsymbol{v}^* \rangle \\
&= f(\boldsymbol{w}^*) + g(\boldsymbol{T}\boldsymbol{v}^*) \\
&= -Q(\boldsymbol{\lambda}^*).
\end{aligned}$$

The last equality follows from the strong duality in Proposition 2. Therefore, from Eq. (49) and Theorem 2, we have

$$\beta \|\boldsymbol{w}^* - \boldsymbol{w}^k\|_2^2 \leq Q(\boldsymbol{\lambda}^k) - Q(\boldsymbol{\lambda}^*) \leq \frac{l+u}{\beta(k+1)^2} \|\boldsymbol{\lambda}^0 - \boldsymbol{\lambda}^*\|_2^2$$

. Finally, we obtain

$$\|\boldsymbol{w}^* - \boldsymbol{w}^k\|_2 \leq \frac{\sqrt{l+u}}{\beta(k+1)} \|\boldsymbol{\lambda}^0 - \boldsymbol{\lambda}^*\|_2$$

$\square$

**Remark.** *Our method applies the FISTA algorithm to the dual formulation (Eq. (8)) of the original problem (Eq. (7)). This will change the original convergence analysis of the FISTA algorithm when applied to the primal formulation directly. Theorem 2 follows the standard convergence analysis of the FISTA algorithm in terms of the dual variable $\boldsymbol{\lambda}$, while Theorem 3 analyzes the convergence result in terms of the primal variable $\boldsymbol{w}$, that we actually care about in Procedure GWBI. We need to use the result of Theorem 5 to prove Theorem 6. Our method achieves the SOTA global convergence rate with provable guarantees for the graph construction step in GSSL, as far as we know.*

### E.9 Proof of Moment Growth Condition

**Lemma 3.** *Assume that the channel of node feature $\boldsymbol{X}_i$ ($i$-th column of $\boldsymbol{X}$) is independently and identically (i.i.d.) generated by the factor analysis model as $\boldsymbol{X}_i = \boldsymbol{\chi}\boldsymbol{h} + \boldsymbol{u}_i + \boldsymbol{\delta}$, where $\boldsymbol{h} \sim \mathcal{N}(\boldsymbol{0}, \boldsymbol{\Lambda}^{*\dagger})$ is the hidden variable, $\boldsymbol{u}_x$ is the mean of $\boldsymbol{X}_i$, and $\boldsymbol{\delta} \sim \mathcal{N}(\boldsymbol{0}, \delta_\epsilon^2 \boldsymbol{I})$ is the Gaussian noise with mean zero and covariance $\delta_\epsilon^2 \boldsymbol{I}$. Then the probability distribution $p(\boldsymbol{X}_i)$ with mean $\boldsymbol{\mu}^* = \boldsymbol{u}_i$ must satisfy the moment growth condition [48]: there exists a constant $c > 0$ such that for $\forall r \geq 1$,*

$$\mathbb{E}_{\boldsymbol{X}_i \sim p(\boldsymbol{X}_i)} \left[ \|\boldsymbol{X}_i - \boldsymbol{\mu}^*\|_2^r \right] \leq (cr)^{r/2}.$$

*Proof.* Note that the latent variable h follows a degenerate zero-mean multivariate Gaussian distribution with the precision matrix defined as the eigenvalue matrix $\boldsymbol{\Lambda}^*$ of the graph Laplacian $\boldsymbol{L}^*$, i.e.,

$$\boldsymbol{h} \sim \mathcal{N}(\boldsymbol{0}, \boldsymbol{\Lambda}^{*\dagger}),$$

where $\boldsymbol{\Lambda}^{*\dagger}$ is the Moore-Penrose pseudoinverse of $\boldsymbol{\Lambda}$. Based on the generative model $\boldsymbol{X}_i = \boldsymbol{\chi}\boldsymbol{h} + \boldsymbol{u}_i + \boldsymbol{\delta}$ and the isotropic noise model $\boldsymbol{\delta} \sim \mathcal{N}(\boldsymbol{0}, \delta_\epsilon^2 \boldsymbol{I})$, the conditional probability of $\boldsymbol{X}_i$ given $\boldsymbol{h}$, and the probability of $\boldsymbol{X}_i$ as:

$$\boldsymbol{X}_i \mid \boldsymbol{h} \sim \mathcal{N}(\boldsymbol{\chi}\boldsymbol{h} + \boldsymbol{u}_i, \delta_\epsilon^2 \boldsymbol{I}),$$

$$\boldsymbol{X}_i \sim \mathcal{N}(\boldsymbol{u}_i, \boldsymbol{L}^{*\dagger} + \delta_\epsilon^2 \boldsymbol{I}).$$

Since $\boldsymbol{X}_i$ follows the Gaussian distribution, based on the probabilistic techniques developed in Lemma 5.5 in [56], we can show that there exists a constant $c > 0$ such that $\forall r \geq 1$, such that

$$\mathbb{E}_{\boldsymbol{X}_i \sim p(\boldsymbol{X}_i)} \left[ \|\boldsymbol{X}_i - \boldsymbol{\mu}^*\|_2^r \right] \leq (cr)^{r/2}.$$

$\square$

## E.10 Proof of Theorem 4

*Proof.* Suppose $\boldsymbol{L}$ is compact and, thus, we have $\max_{\boldsymbol{L}} \|\boldsymbol{L}\| < +\infty$. Together with Lemma 3, we obtain that there exists a constant $c > 0$ such that for all $r \geq 1$,

$$\max_{\boldsymbol{L}} \left\{ \boldsymbol{E}_{\boldsymbol{X}_i \sim p(\boldsymbol{X}_1)} \left[ \|\boldsymbol{L}^{1/2}(\boldsymbol{X}_i - \boldsymbol{\mu}^*)\|_2^r \right] \right\} \leq (cr)^{r/2}.$$

Let us define $\boldsymbol{\zeta}_i = \boldsymbol{L}^{1/2}(\boldsymbol{X}_i - \boldsymbol{\mu}^*)$ for $i = 1, 2, \cdots, d$, and let $\tau_1, \cdots, \tau_d$ be i.d.d. Bernoulli random variables. Since $\mathbb{E}_{\boldsymbol{X}_i \sim p(\boldsymbol{X}_i)}[\boldsymbol{\zeta}_i] = 0$ for $i = 1, \cdots, d$, by the convexity of $x \mapsto |x|^r$ on $\mathbb{R}_+$ for any $r \geq 1$, we have

$$\mathbb{E}_{\boldsymbol{X}_i \sim p(\boldsymbol{X}_i)} \left[ \left\| \sum_{i=1}^d \boldsymbol{\zeta}_i \right\|_2^r \right] \leq 2^r \mathbb{E} \left[ \left\| \sum_{i=1}^d \tau_i \boldsymbol{\zeta}_i \right\|_2^r \right].$$

Now by Jensen's Inequality, conditioned on $\boldsymbol{\zeta}_i$, we have for $r \geq 2$

$$\mathbb{E}_\tau \left[ \left\| \sum_{i=1}^d \tau_i \boldsymbol{\zeta}_i \right\|_2^r \right] < r^{r/2} \left( \sum_{i=1}^d \|\boldsymbol{\zeta}_i\|_2^2 \right)^{r/2} \leq d^{r/2-1} r^{r/2} \left( \sum_{i=1}^d \|\boldsymbol{\zeta}_i\|_2^r \right).$$

By Markov's inequality, for any $t > 0$ and $r \geq 2$, we obtain

$$\mathbb{P} \left[ \left\| \frac{1}{d} \sum_{i=1}^d \boldsymbol{\zeta}_i \right\|_2 > t \right] \leq t^{-r} d^{-r} \mathbb{E}_{\boldsymbol{X}_i \sim p(\boldsymbol{X}_i)} \left[ \left\| \sum_{i=1}^d \boldsymbol{\zeta}_i \right\|_2^r \right] \leq \frac{2^r c^{r/2} r^r}{t^r d^{r/2}}.$$

If we choose $t = e\sqrt{4c/d} \ln(1/\xi)$, $r = \frac{t}{e}\sqrt{4c/d} \geq 2$ (since we have $\xi \in (0, \frac{1}{e^2})$), we conclude that

$$\mathbb{P} \left[ (\boldsymbol{\mu}^* - \hat{\boldsymbol{\mu}}_d)^\mathsf{T} \boldsymbol{L}(\boldsymbol{\mu}^* - \hat{\boldsymbol{\mu}}_d) > \frac{4ce^2}{d} \ln^2(1/\xi) \right] = \mathbb{P} \left[ \left\| \frac{1}{d} \sum_{i=1}^d \boldsymbol{\zeta}_i \right\|_2 > \rho_1 \right] \leq \xi,$$

where $\rho_1 = \frac{c_1}{d} \ln^2(\frac{1}{\xi})$ and $c_1 = 4ce^2$. $\square$

## E.11 Proof of Theorem 5

*Proof.* We define

$$\tilde{\boldsymbol{\Sigma}}_d = \frac{1}{d} \sum_{i=1}^d (\boldsymbol{X}_i - \boldsymbol{\mu}^*)(\boldsymbol{X}_i - \boldsymbol{\mu}^*)^\mathsf{T},$$

and we want to first prove there exists some constant $c_2 > 0$ such that

$$\|\tilde{\boldsymbol{\Sigma}}_d - \boldsymbol{\Sigma}^*\|_F \leq \frac{c_2}{\sqrt{d}} \ln^{3/2}(4n^{3/2}/\xi) \|\boldsymbol{\Sigma}^*\|$$

will hold with high probability at least $1 - \xi/2$. For $i = 1, \cdots, d$, we let

$$\boldsymbol{Q}_i = \boldsymbol{\Sigma}^{*-1/2}(\boldsymbol{X}_i - \boldsymbol{\mu}^*)(\boldsymbol{X}_i - \boldsymbol{\mu}^*)^\mathsf{T} \boldsymbol{\Sigma}^{*-1/2} - \boldsymbol{I}_n.$$

It is easy to show that $\mathbb{E}_{\boldsymbol{X}_i \sim p(\boldsymbol{X}_i)}[\boldsymbol{Q}_i] = 0$ for $i = 1, \cdots, d$. Also from Lemma 3, we know that there exists a constant $c' > 0$ such that for all $r \geq 1$,

$$\mathbb{E}_{\boldsymbol{X}_i \sim p(\boldsymbol{X}_i)} \left[ \left\| \boldsymbol{\Sigma}^{*-1/2}(\boldsymbol{X}_i - \boldsymbol{\mu}^*) \right\|_2^r \right] \leq (c'r)^{r/2}.$$

Together with Proposition 5 in [48], we show that for any $r \geq 1$,

$$\mathbb{E}_{\boldsymbol{X}_i \sim p(\boldsymbol{X}_i)} \left[ \left\| \sum_{i=1}^d \boldsymbol{Q}_i \right\|_{S_r}^r \right] \leq 2^r d^{r/2} r^{r/2} (n + (2c'r)^r).$$

Here, we use $\|\boldsymbol{A}\|_{S_r}$ to denote the Schatten $r$-norm of $\boldsymbol{A} \in \mathbb{R}^{q_1 \times q_2}$; i.e., $\|\boldsymbol{A}\|_{S_r} := \|\sigma(\boldsymbol{A})\|_r$, where $\sigma(\boldsymbol{A})$ is the vector of singular values. Note that $\|\boldsymbol{v}\|_2 \leq q^{1/2} \|\boldsymbol{v}\|_r$ for any $\boldsymbol{v} \in \mathbb{R}^q$ and $r \geq 2$. By applying Markov's inequality, we get , for any $r \geq 2$ and $t > 0$, that

$$\mathbb{P} \left[ \left\| \frac{1}{d} \sum_{i=1}^d \boldsymbol{Q}_i \right\|_F > t \right] = \mathbb{P} \left[ \left\| \frac{1}{d} \sum_{i=1}^d \boldsymbol{Q}_i \right\|_{S_2}^r > t^r \right] \leq \frac{2^r r^{r/2} n^{1/2}}{t^r d^{r/2}} (n + (2c'r)^r).$$

By setting $c = \max\{c', \frac{1}{4}\}$, $t = \frac{c_2}{\sqrt{d}}\ln^{3/2}(4n^{3/2}/\xi)$, $c_2 = 4c(\frac{2e}{3})^{3/2}$, $r = \left(\frac{t\sqrt{d}}{4c^{3/2}}\right)^{2/3} \geq 2$ (since we have $\xi \leq \frac{1}{e^2}$), we have

$$\frac{2^r r^{r/2} n^{1/2}}{t^r d^{r/2}}(n + (2c'r)^r) = \frac{n^{3/2} + n^{1/2}(2c'r)^r}{e^{3r/2}(2cr)^r} \leq \frac{\xi}{2}.$$

Finally, recall that

$$\|\tilde{\boldsymbol{\Sigma}}_d - \boldsymbol{\Sigma}^*\|_F = \left\|\boldsymbol{\Sigma}^{*1/2}\left(\frac{1}{d}\sum_{i=1}^d \boldsymbol{Q}_i\right)\right\|_F,$$

we have

$$\mathbb{P}\left[\|\tilde{\boldsymbol{\Sigma}}_d - \boldsymbol{\Sigma}^*\|_F > \frac{c_2}{\sqrt{d}}\ln^{3/2}(4n^{3/2}/\xi)\|\boldsymbol{\Sigma}^*\|\right] \leq \frac{\xi}{2}.$$

It is easy to observe that $\tilde{\boldsymbol{\Sigma}}_d = \hat{\boldsymbol{\Sigma}}_d + (\hat{\boldsymbol{\mu}}_d - \boldsymbol{\mu}^*)(\hat{\boldsymbol{\mu}}_d - \boldsymbol{\mu}^*)^\mathsf{T}$. Hence, we obtain

$$\|\hat{\boldsymbol{\Sigma}}_d - \boldsymbol{\Sigma}^*\|_F$$
$$\leq \|\hat{\boldsymbol{\Sigma}}_d - \tilde{\boldsymbol{\Sigma}}_d\|_F + \|\tilde{\boldsymbol{\Sigma}}_d - \boldsymbol{\Sigma}^*\|_F$$
$$= (\hat{\boldsymbol{\mu}}_d - \boldsymbol{\mu}^*)^\mathsf{T}(\hat{\boldsymbol{\mu}}_d - \boldsymbol{\mu}^*) + \|\tilde{\boldsymbol{\Sigma}}_d - \boldsymbol{\Sigma}^*\|_F.$$

Similar to Theorem 4, we can deduce, there exists $c_3 > 0$ such that

$$\mathbb{P}\left[(\hat{\boldsymbol{\mu}}_d - \boldsymbol{\mu}^*)^\mathsf{T}(\hat{\boldsymbol{\mu}}_d - \boldsymbol{\mu}^*) > \frac{c_3}{d}\ln^2(2/\xi)\right] \leq \frac{\xi}{2}.$$

By union bound, we finally show that

$$\mathbb{P}\left[\|\boldsymbol{\Sigma}^* - \hat{\boldsymbol{\Sigma}}_d\|_F > \frac{c_2}{\sqrt{d}}\ln^{3/2}\left(\frac{4d^{3/2}}{\xi}\right)\|\Sigma^*\|_F + \frac{c_3}{d}\ln^2\left(\frac{2}{\xi}\right)\right] \leq \xi.$$

$\square$

### E.12 Proof of Theorem 6

*Proof.* We introduce the following notations. Let $i_{l+1} \neq i_1, \cdots, i_l$ be an integer randomly sampled from $D_u$ and let $D_{l+1} = D_l \cup \{i_{l+1}\}$. Let $\hat{\boldsymbol{F}}(D_{l+1})$ be the graph-based semi-supervised learning method in Eq. (1) using the training data in $D_{l+1}$ with $\boldsymbol{\Lambda} = \begin{pmatrix} \frac{1}{\lambda l}\boldsymbol{I}_l & \boldsymbol{O} \\ \boldsymbol{O} & \boldsymbol{O} \end{pmatrix}$, we have

$$\hat{\boldsymbol{F}}(D_{l+1}) = \arg\min_{\boldsymbol{F}}\left\{\frac{1}{l}\sum_{j \in D_{l+1}} \phi(\boldsymbol{f}_j, \boldsymbol{y}_j) + \lambda\,\mathrm{Tr}(\boldsymbol{F}^\mathsf{T}\boldsymbol{S}\boldsymbol{F})\right\},$$

where $\phi(\boldsymbol{f}_j - \boldsymbol{y}_j) = \sum_{k=1}^c \phi_0(\boldsymbol{f}_{j,k}, \boldsymbol{y}_{j,k})$ and $\phi_0(\boldsymbol{f}_{j,k}, \boldsymbol{y}_{j,k}) = (\boldsymbol{f}_{j,k} - \boldsymbol{y}_{j,k})^2$. It is easy to verify that $\phi_0(x, y)$ enjoys the following properties.

1. $\phi_0(x, y)$ is non-negative and convex in $x$.

2. When $y = 0, 1$ and $\phi_0(x, y) \leq \frac{1}{16}$, we have $|\nabla_x \phi_0(x, y)| \leq \frac{1}{2}$.

3. $\min_x\{\phi_0(x, 1) \leq \frac{1}{16}\} - \max_x\{\phi_0(x, 0) \leq \frac{1}{16}\} = \frac{1}{2}$.

We first show two lemmas.

**Lemma 4.** *For each $k = 1, \cdots, c$, we have*

$$|\hat{\boldsymbol{F}}(D_{l+1})_{i_{l+1},k} - \hat{\boldsymbol{F}}(D_l)_{i_l,k}| \leq \frac{\boldsymbol{S}_{i_{l+1},i_{l+1}}^{-1}}{2\lambda l}|\nabla_k\phi(\hat{\boldsymbol{F}}(D_{l+1})_{i_{l+1}}, \boldsymbol{y}_{l+1})|,$$

*where $\nabla_k\phi(\hat{\boldsymbol{F}}(D_{l+1})_{i_{l+1}}, \boldsymbol{y}_{l+1})$ denotes the gradient of $\phi(\hat{\boldsymbol{F}}(D_{l+1})_{i_{l+1}}, \boldsymbol{y}_{l+1})$ with respect to $\hat{\boldsymbol{F}}(D_{l+1})_{i_{l+1},k}$.*

*Proof.* We know that there exists a gradient $\nabla_k \phi$ such that the following first-order condition holds,

$$-2\lambda l \mathbf{S}^{-1} \hat{\mathbf{F}}(D_l)._{\cdot,k} = \sum_{h \in D_l} \nabla_k \phi(\hat{\mathbf{F}}(D_l)_j, \mathbf{y}_j) \mathbf{e}_j,$$

where $\mathbf{e}_j$ is the $n$-dimensional vector with all zeros except for the $j$-th component with value one. Similarly, we have,

$$-2\lambda l \mathbf{S}^{-1} \hat{\mathbf{F}}(D_{l+1})._{\cdot,k} = \sum_{h \in D_{l+1}} \nabla_k \phi(\hat{\mathbf{F}}(D_{l+1})_j, \mathbf{y}_j) \mathbf{e}_j.$$

Now, we let $\mathbf{G} = \hat{\mathbf{F}}(D_l)$ and $\mathbf{H} = \hat{\mathbf{F}}(D_{l+1})$. By subtracting the above two equations, and taking the inner product with $\mathbf{H}_{\cdot,k} - \mathbf{H}_{\cdot,k}$, we obtain,

$$
\begin{aligned}
&- 2\lambda l (\mathbf{H}_{\cdot,k} - \mathbf{G}_{\cdot,k})^\mathsf{T} \mathbf{S} (\mathbf{H}_{\cdot,k} - \mathbf{G}_{\cdot,k}) \\
&= \nabla_k \phi(\mathbf{H}_{i_{l+1}}, \mathbf{y}_{i_{l+1}})(\mathbf{H}_{i_{l+1},k} - \mathbf{G}_{i_{l+1},k}) \\
&+ \sum_{j \in D_l} \left( \nabla_k \phi(\mathbf{H}_j, \mathbf{y}_j) - \nabla_k \phi(\mathbf{G}_j, \mathbf{y}_j) \right) (\mathbf{H}_{j,k} - \mathbf{G}_{j,k}) \\
&\leq -\nabla_k \phi(\mathbf{H}_{i_{l+1}}, \mathbf{y}_{i_{l+1}})(\mathbf{H}_{i_{l+1},k} - \mathbf{G}_{i_{l+1},k}).
\end{aligned}
$$

The last inequality is based on the convexity of $\phi$. Using Cauchy-Schwarz inequality, we have,

$$
\begin{aligned}
&2\lambda l (\mathbf{H}_{i_{l+1},k} - \mathbf{G}_{i_{l+1},k})^2 \\
&= 2\lambda l \left( (\mathbf{H}_{\cdot,k} - \mathbf{G}_{\cdot,k})^\mathsf{T} \mathbf{e}_{i_{l+1}} \right)^2 \\
&\leq 2\lambda l (\mathbf{H}_{\cdot,k} - \mathbf{G}_{\cdot,k})^\mathsf{T} \mathbf{S} (\mathbf{H}_{\cdot,k} - \mathbf{G}_{\cdot,k}) \mathbf{e}_{i_{l+1}}^\mathsf{T} \mathbf{S}^{-1} \mathbf{e}_{i_{l+1}} \\
&\leq |\nabla_k \phi(\mathbf{H}_{i_{l+1}}, \mathbf{y}_{i_{l+1}})| \cdot |\mathbf{H}_{i_{l+1},k} - \mathbf{G}_{i_{l+1},k}| \mathbf{S}^{-1}_{i_{l+1},i_{l+1}}.
\end{aligned}
$$

Therefore, we have $|\mathbf{H}_{i_{l+1},k} - \mathbf{G}_{i_{l+1},k}| \leq \dfrac{\mathbf{S}^{-1}_{i_{l+1},i_{l+1}}}{2\lambda l} |\nabla_k \phi(\mathbf{H}_i), \mathbf{y}_{i_{l+1}}|$, which completes the proof. $\square$

**Lemma 5.**

$$\mathbb{I}_{\hat{y}_{i_{l+1}}(D_l) \neq y_{i_{l+1}}} \leq \max_{k=k_0,i_{l+1}} \left\{ 16\phi_0(\hat{\mathbf{F}}(D_{l+1})_{i_{l+1},k}, \mathbf{y}_{i_{l+1},k}) + \frac{1}{\lambda l} \mathbf{S}^{-1}_{i_{l+1},i_{l+1}} \right\}.$$

*Proof.* If $\hat{\mathbf{F}}(D_l)$ does not make an error on the $i_{l+1}$-th sample. That is

$$\mathbb{I}_{\hat{y}_{i_{l+1}}(D_l) \neq y_{i_{l+1}}} = 0,$$

and thus the inequality automatically hold.

Assume that $\hat{\mathbf{F}}(D_l)$ makes an error on the $i_{l+1}$-th sample, then we immediately have $\mathbb{I}_{\hat{y}_{i_{l+1}}(D_l) \neq y_{i_{l+1}}} = 1$. So there exists some $k_o \neq y_{i_{l+1}}$ such that $\hat{\mathbf{F}}(D_l)_{i_{l+1},y_{i_{l+1}}} \leq \hat{\mathbf{F}}(D_l)_{i_{l+1},k_0}$. This means that for any constant $d$, either we have $\hat{\mathbf{F}}(D_l)_{i_{l+1},y_{i_{l+1}}} \leq d$ or $\hat{\mathbf{F}}(D_l)_{i_{l+1},k_0} \geq d$. Let

$$d = \left( \min_x \{\phi_0(x,1) \leq \frac{1}{16}\} + \max_x \{\phi_0(x,0) \leq \frac{1}{16}\} \right) / 2,$$

and notice that we also have

$$\min_x \{\phi_0(x,1) \leq \frac{1}{16}\} - \max_x \{\phi_0(x,0) \leq \frac{1}{16}\} = \frac{1}{2}.$$

Therefore, we either have

$$\min_x \{\phi_0(x,1) \leq \frac{1}{16}\} - \hat{\mathbf{F}}(D_l)_{i_{l+1},y_{i_{l+1}}} \geq \frac{1}{4},$$

or we have

$$\hat{\mathbf{F}}(D_l)_{i_{l+1},k_0} - \max_x \{\phi_0(x,0) \leq \frac{1}{16}\} \geq \frac{1}{4}.$$

It follows that there exists $k = k_0$ or $k = y_{i_{l+1}}$ such that either

$$\phi_0(\hat{\boldsymbol{F}}(D_{l+1})_{i_{l+1},k}, \boldsymbol{y}_{i_{l+1},k}) \geq \frac{1}{16},$$

or

$$|\hat{\boldsymbol{F}}(D_{l+1})_{i_{l+1},k} - \hat{\boldsymbol{F}}(D_l)_{i_{l+1},k}| \geq \frac{1}{4}.$$

From Lemma 4, we have either

$$16\phi_0(\hat{\boldsymbol{F}}(D_{l+1})_{i_{l+1},k}, \boldsymbol{y}_{i_{l+1},k}) \geq 1,$$

or

$$\frac{\boldsymbol{S}^{-1}_{i_{l+1},i_{l+1}}}{2\lambda l} \geq \frac{1}{2}.$$

This implies that

$$16\phi_0(\hat{\boldsymbol{F}}(D_{l+1})_{i_{l+1},k}, \boldsymbol{y}_{i_{l+1},k}) + \frac{\boldsymbol{S}^{-1}_{i_{l+1},i_{l+1}}}{\lambda l} \geq 1 = \mathbb{I}_{\hat{y}_{i_{l+1}}(D_l) \neq y_{i_{l+1}}}.$$

$$\square$$

With these two lemmas in hand, we are ready to prove Theorem 6. We denote $D_{l+1}^{(j)}$ as the subset of $l$ samples in $D_{l+1}$ with the $j$-th sample left out. From Lemma 5, we have,

$$\mathbb{I}_{\hat{y}_{i_{l+1}}(D_l^{(j)}) \neq y_j} \leq 16\phi_0(\hat{\boldsymbol{F}}(D_{l+1})_{i_{l+1},k}, \boldsymbol{y}_j) + \frac{1}{\lambda l}\boldsymbol{S}^{-1}_{j,j}.$$

Therefore, we can get,

$$\mathbb{E}_{D_l}\left[\frac{1}{u}\sum_{j \in D_u}\mathbb{I}_{\hat{y}_i \neq y_i}\right]$$

$$= \mathbb{E}_{D_{l+1}}\left[\mathbb{I}_{\hat{y}(D_{l+1}^{l+1})_{l+1} \neq y_{l+1}}\right]$$

$$= \frac{1}{l+1}\mathcal{E}_{D_{l+1}}\left[\sum_{j \in D_{l+1}}\mathbb{I}_{\hat{y}(D_{l+1}^{(j)}) \neq y_j}\right]$$

$$\leq \frac{16l}{l+1}\mathbb{E}_{D_{l+1}}\left[\frac{1}{l}\sum_{j \in D_{l+1}}\phi(\boldsymbol{f}_j, \boldsymbol{y}_j) + \lambda\operatorname{Tr}(\boldsymbol{F}^{\mathsf{T}}\boldsymbol{S}\boldsymbol{F})\right] \tag{50}$$

$$+ \frac{1}{l+1}\mathbb{E}_{D_{l+1}}\left[\sum_{j \in D_{l+1}}\frac{1}{\lambda l}\boldsymbol{S}^{-1}_{jj}\right]$$

$$= 16\left\{\frac{1}{n}\sum_{j=1}^{n}\phi(\boldsymbol{f}_j, \boldsymbol{y}_j) + \frac{\lambda l}{l+1}\operatorname{Tr}(\boldsymbol{F}^{\mathsf{T}}\boldsymbol{S}\boldsymbol{F})\right\} + \frac{1}{\lambda l}\left\{\frac{1}{n}\sum_{j=1}^{m}\boldsymbol{S}^{-1}_{jj}\right\}$$

$$= 16\min_{\boldsymbol{F}}\{Q(\boldsymbol{F})\} + \frac{\operatorname{Tr}(\boldsymbol{\Lambda})\operatorname{Tr}(\boldsymbol{S}^{-1})}{nl}.$$

The last equality follows from the definition of $\boldsymbol{F}^*$ in Eq. (1) and $\boldsymbol{\Lambda} = \begin{pmatrix} \frac{1}{\lambda l}\boldsymbol{I}_l & \boldsymbol{O} \\ \boldsymbol{O} & \boldsymbol{O} \end{pmatrix}$. $\qquad \square$

# F  Experiment Settings

## F.1  Datasets Description

**ORHD** [1] (Optical Recognition of Handwritten Digits Data Set), **USPS** [2], **MNIST** [3], and **EMNIST Letters** [4] are four popular digits image datasets. **COIL100** [5] is an object image dataset. **TDT2** [6] is a text dataset. We fix the number of anchor nodes as 1000 in four datasets (COIL100, USPS, ORHD, and TDT2), while for the rest two datasets (MNIST, EMNIST-Letters), the number of anchors is fixed as 2000 instead. We also perform tf-idf and principal component analysis (PCA) as the pre-processing step on the TDT2 dataset to reduce the running time. The features in the other five datasets are normalized with the popular Zscore method.

## F.2  Implementation Details

All the experiments are conducted on a hardware configuration with a 3.8 GHz 8-Core Intel Core i7 CPU with 32 GB 2667 MHz DDR4 RAM. The software configuration is MATLAB with the R2021b version.

All the hyper-parameters are fine-tuned with the grid search method. We list all the common hyper-parameters settings in all baseline methods and the optimal hyper-parameters settings in our proposed method on the six datasets. Other uncovered parameters are fixed as their original papers suggest. We repeat the experiment 20 times for each case and report the average result with optimal parameter setting in efficacy and robustness analysis. Unless otherwise specified, the default label inference algorithm is LGC, and the label rate is ten labeled samples per class.

**RBF**   Because it is not straightforward to find an adequate value for the kernel bandwidth when the labeled examples are scarce. We estimate its value by a third of the average distance between each sample and its $k$-th nearest neighbor, as suggested in [26].

**kNN**   The sparsification parameter $k$ is chosen from the range {1,2,4,5,8,10,20,50}.

**SGL**   The trade-off parameters $\alpha, \beta$ are chosen from two ranges, {0.01,0.02,0.05,0.1,0.2,0.5,1} and {0.001,0.002,0.005,0.01,0.02,0.05,0.1}, respectively.

**RGCLI**   The number of neighbors considered in kNN $k_e$ is fixed at 50 while the number of neighbors considered in RGCLI $k_i$ is chosen from the range {5,10,20,30,40,50}.

**AGR**   The k-means clustering centers are taken as anchor nodes. The number of the nearest anchors $s$ for each sample is fixed as 3.

**GraphEBM**   The label-level threshold $\gamma_l$ to restrict the choice of neighborhood is chosen from the range {5, 10, 15, 20, 25}. The feature-level threshold is chosen from the range {0.1,0.2,0.3,0.4,0.5,0.6,0.7,0.8}.

**BCAN**   The trade-off parameter that balances the graph and label inference $\alpha$ is chosen from the range {0.001,0.01,0.1,1,10,100,1000}.

**BAGL**   We have 4 main hyper-parameters. The trade-off parameters $\alpha_1, \alpha_2$ are chosen from the range {0.01,0.02,0.05,0.1,0.2,0.5,1} while the trade-off parameters $\beta_1, \beta_2$ are chosen from the range {0.001,0.002,0.005,0.01,0.02,0.05,0.1}.

We summarize all the optimal hyper-parameter settings for BAGL on the six datasets as Table 6.

---

[1] http://archive.ics.uci.edu/ml/datasets/Optical+Recognition+of+Handwritten+Digits
[2] https://www.csie.ntu.edu.tw/~cjlin/libsvmtools/datasets/multiclass.html\#usps
[3] http://yann.lecun.com/exdb/mnist/
[4] https://www.nist.gov/itl/products-and-services/emnist-dataset
[5] https://www.cs.columbia.edu/CAVE/software/softlib/coil-100.php
[6] http://www.cad.zju.edu.cn/home/dengcai/Data/TextData.html

Table 6: Hyperparameter settings for optimal classification accuracy on the six datasets.

|  |  | $\alpha_1$ | $\beta_1$ | $\alpha_2$ | $\beta_2$ |
|---|---|---|---|---|---|
| ORHD | GRF | 0.2 | 0.01 | 0.1 | 0.01 |
|  | LGC | 0.5 | 0.02 | 0.1 | 0.005 |
|  | GCN | 0.2 | 0.005 | 0.1 | 0.01 |
| USPS | GRF | 0.5 | 0.02 | 0.2 | 0.01 |
|  | LGC | 0.5 | 0.01 | 0.2 | 0.01 |
|  | GCN | 0.5 | 0.02 | 0.1 | 0.01 |
| COIL100 | GRF | 0.2 | 0.01 | 0.02 | 0.002 |
|  | LGC | 0.1 | 0.005 | 0.05 | 0.002 |
|  | GCN | 0.1 | 0.005 | 0.05 | 0.005 |
| TDT2 | GRF | 0.2 | 0.02 | 0.05 | 0.02 |
|  | LGC | 0.2 | 0.01 | 0.1 | 0.01 |
|  | GCN | 0.2 | 0.01 | 0.1 | 0.02 |
| MNIST | GRF | 0.2 | 0.02 | 0.05 | 0.02 |
|  | LGC | 0.5 | 0.05 | 0.05 | 0.005 |
|  | GCN | 0.5 | 0.02 | 0.1 | 0.005 |
| EMNIST Letters | GRF | 0.2 | 0.02 | 0.02 | 0.01 |
|  | LGC | 0.2 | 0.02 | 0.02 | 0.05 |
|  | GCN | 0.2 | 0.02 | 0.02 | 0.05 |

## G  Experiment Results

### G.1  Classification Results on Other Datasets

We still conduct experiments under extremely low label rates with LGC fixed as the label inference method on other datasets. Fig. 4 demonstrates that BAGL performs relatively well with varying label rates, especially on the TDT2 and EMNIST Letters datasets.

### G.2  Convergence Results on Other Datasets

We continue to show the faster convergence rate of BAGL on other large-scale datasets. We still only sample 1% nodes in each dataset for the convenience of presentation in Fig. 5. Its superiority in terms of the convergence rate becomes apparent when constructing large-scale graphs.

### G.3  Robustness Analysis

For starters, to validate the robustness of BAGL, we add Gaussian noise $\mathcal{N}(0, \gamma\sigma_i)$ on $i$-th feature channel with the corresponding variance $\sigma_i$ and $\gamma$ ranging from $\{0, 1, 2, 4\}$. Fig. 6 reveals that BAGL is exceptionally robust to feature noise, especially on the TDT2 and MNIST datasets.

### G.4  Sensitivity Analysis

Finally, we explore the sensitivity of the model performance with regard to the parameters $\alpha_1, \alpha_2$ and $\beta_1, \beta_2$. $\alpha_1, \alpha_2$ are chosen from the range $\{0.01,0.02,0.05,0.1,0.2,0.5,1\}$ and $\beta_1, \beta_2$ are chosen from the range $\{0.001,0.002,0.005,0.01,0.02,0.05,0.1\}$. Fig. 7 displays the performance with respect to variations of them. Overall, BAGL is relatively insensitive to a fixed range of parameter variations. There will be a performance drop if either the connectivity penalty term or the sparsity penalty term dominates the objective function.

### G.5  Scalability Analysis

#### G.5.1  Time Complexity Analysis

We now rigorously analyze the time complexity of Algorithm 1. The first step to construct the distance matrix takes $O(n^2)$ trivially. Then it takes $O(l + u) = O(n)$ to construct the linear mapping matrix $\boldsymbol{T}_a$ since it is a sparse matrix and it only takes constant time to construct each row in $\boldsymbol{T}_a$

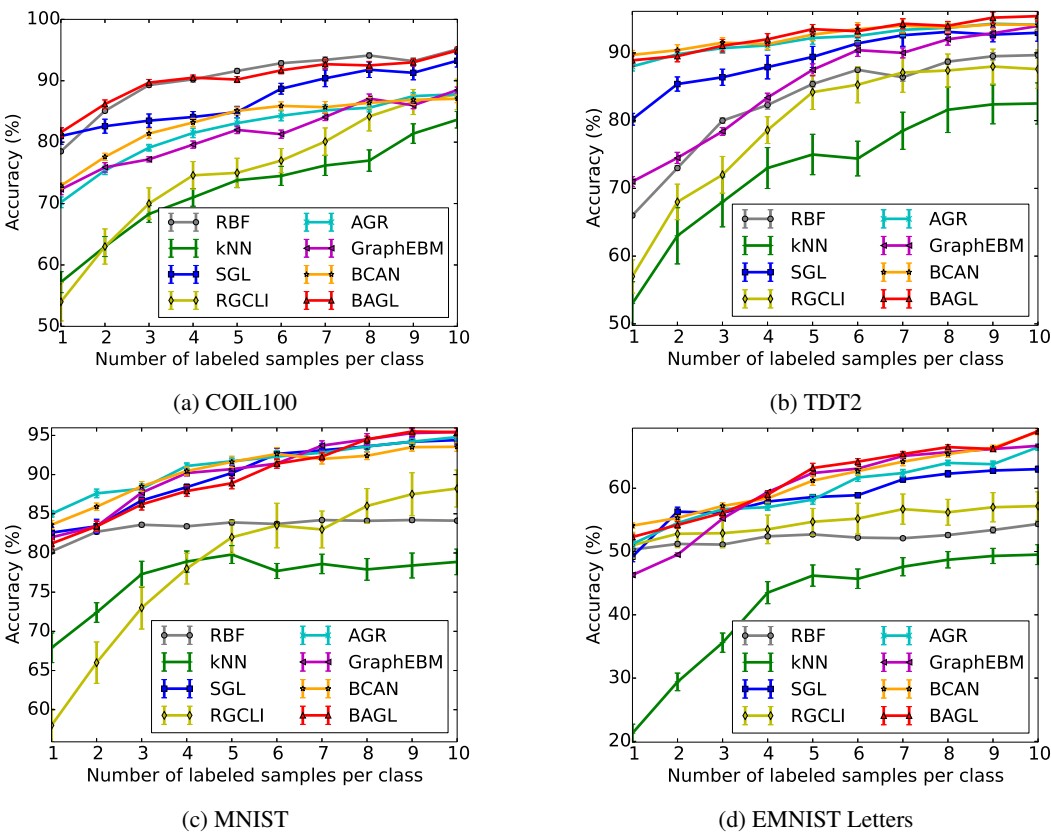

Figure 4: Classification results on real-world datasets with different label rates (Cont.).

given the matrix vectorization operation. Similarly, it costs $O(u + u) = O(u)$ to recover $\boldsymbol{T}_b$. We can neglect the assignment for the step size since it also takes constant time. The last two steps also only cost constant time to complete.

The major overhead now lies in the two intermediate procedure calls, and thus we need to dive into Procedure GWBI. For each iteration, the update for $\boldsymbol{\mu}^k$ costs $O(n)$ according to the dimension of $\boldsymbol{\lambda} \in \mathbb{R}^n$. It is easy to see that the dominant overhead in the next three update steps is the multiplication of a spare matrix $\boldsymbol{T}$ ($\boldsymbol{T}^\intercal$) with a vector. Since $\boldsymbol{T}$ has $ul + ul = 2ul$ none-zero entities, the computational cost of matrix-vector multiplication is $o(ul)$ in the next three steps. More precisely, the update for $\bar{\boldsymbol{w}}^k$ takes $O(ul + ul) = O(ul)$. The updates for $\bar{\boldsymbol{u}}^k$ and $\boldsymbol{\lambda}^k$ both take $O(ul + n)$. Therefore, the computational cost for each iteration is $O(ul + n)$. For the current optimization-based graph construction methods in GSSL, which are often based on the primal-dual method [29] or ADMM [18, 61], they all need $O(n^2)$ in each iteration since they operate on the whole graph. Note that $n = u + l$ and $O(ul + n) \leq O(\frac{1}{4}n^2 + n) = O(n^2)$, our method can reduce the time complexity of each iteration. Furthermore, thanks to the faster convergence rate, the number of iterations $O(\frac{1}{\epsilon})$ to obtain the optimal solution is also significantly reduced. Here, $\epsilon$ denotes the user-defined precision value.

In summary, the total time complexity for the proposed GWBI procedure is $O(\frac{1}{\epsilon}(ul + n))$. The total time complexity for the proposed BAGL algorithm is $O(\frac{1}{\epsilon}un)$

### G.5.2 Running Time Comparison

Finally, we also compare the actual running time for each algorithm with the hardware and software configurations provided. Table 7 summarizes the time costs of all the compared models. The best results are bolded, and the second-best results are italicized. It is easy to check that our BAGL algorithm is faster than all the optimization-based methods on all datasets, resulting in a large extent

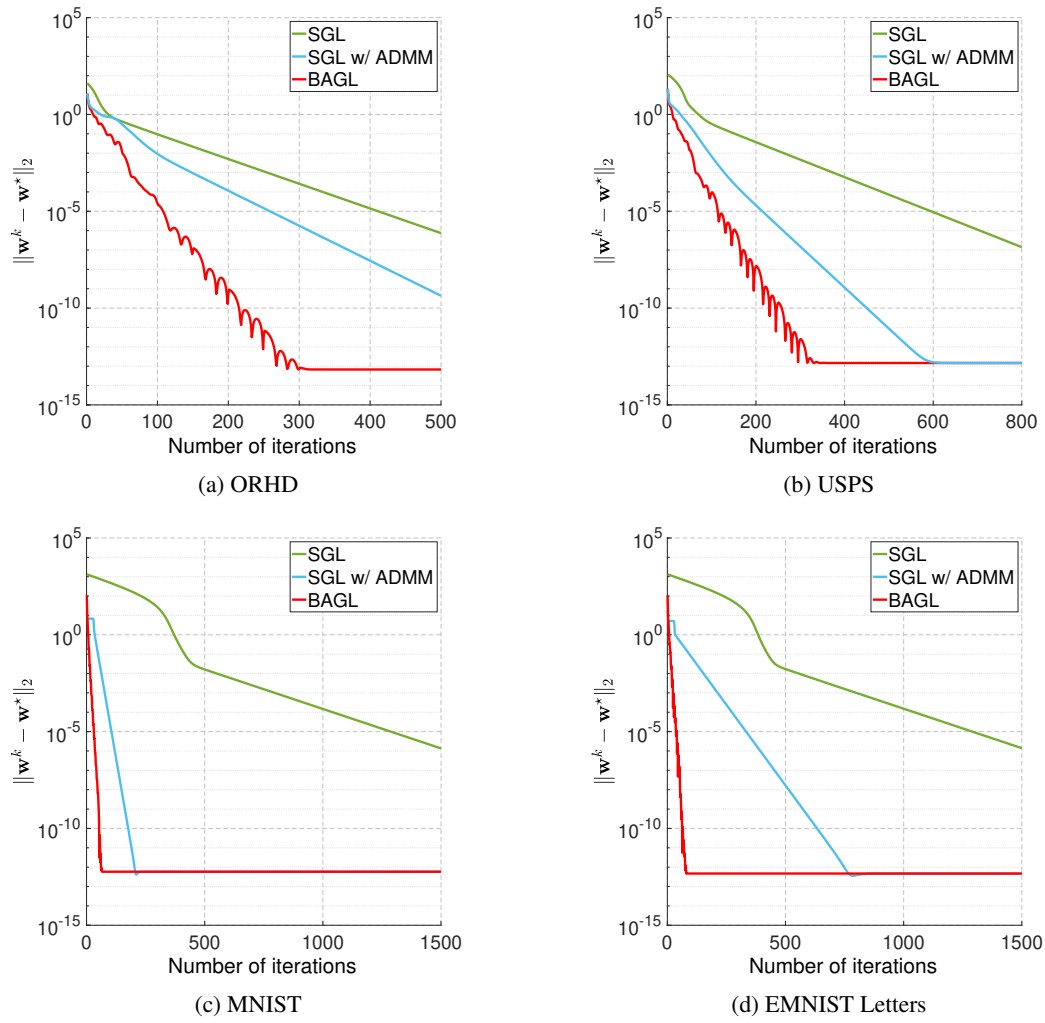

(a) ORHD

(b) USPS

(c) MNIST

(d) EMNIST Letters

Figure 5: Convergence results on real-world datasets (Cont.).

of improvement. Such advantage is essential to scale for large-scale datasets and is also in agreement with the time complexity analysis above.

Table 7: Average time cost (seconds) of compared algorithms on all real-world datasets.

| | Optimization-based | | | Others | | |
|---|---|---|---|---|---|---|
| | SGL | SGL w/ADMM | BAGL | kNN | AGR | BCAN |
| ORHD | 3.31 | 1.20 | *0.79* | **0.02** | 5.35 | 1.52 |
| USPS | 3.86 | 2.97 | *1.24* | **0.04** | 9.14 | 3.07 |
| COIL100 | 7.12 | 6.56 | *2.93* | **0.95** | 9.35 | 4.16 |
| TDT2 | 8.57 | 7.45 | *3.88* | **0.03** | 9.81 | 8.23 |
| MNIST | 102.65 | 48.52 | *35.41* | **1.24** | 70.50 | 82.48 |
| EMNIST Letters | 478.24 | 134.39 | *102.35* | **4.77** | 372.19 | 297.61 |

# H  Broader Impacts

Since our method focuses on the graph construction step to enhance the overall performance of graph-based semi-supervised learning, BAGL makes broader impacts for social benefits. As we all know, graph-based semi-supervised learning has substantial implications across multiple disciplines and industry sectors, given its proficiency in learning from sparse labels within large, unstructured

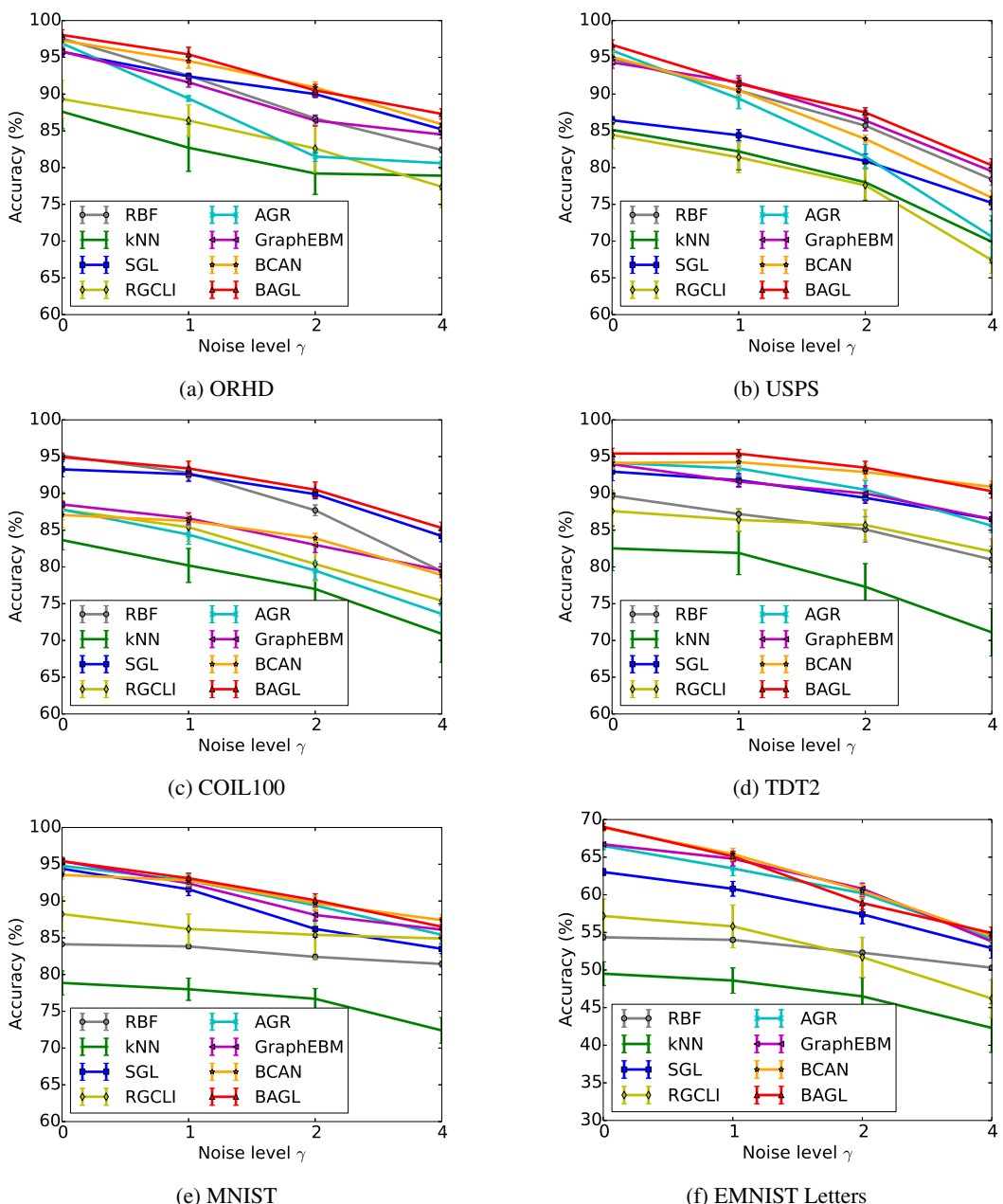

Figure 6: Accuracy on two datasets with different noise levels of features.

datasets. Its applications span from healthcare and drug discovery to social media and web services. In each case, the ability to exploit the structure of data can dramatically improve performance over traditional, non-graph-based learning methods.

For example, in the field of healthcare and drug discovery, with graph-based semi-supervised learning, researchers can predict disease progression or drug responses by analyzing biological networks, such as gene regulatory or protein-protein interaction networks. This can facilitate personalized medicine by identifying patients' specific needs based on their genetic profiles or disease history. The prediction power could also aid in discovering potential drug targets or optimizing drug combinations, significantly accelerating the drug discovery process.

In the applications of social media and web services, graph-based semi-supervised learning offers superior performance in understanding the vast and intricate connections within social networks.

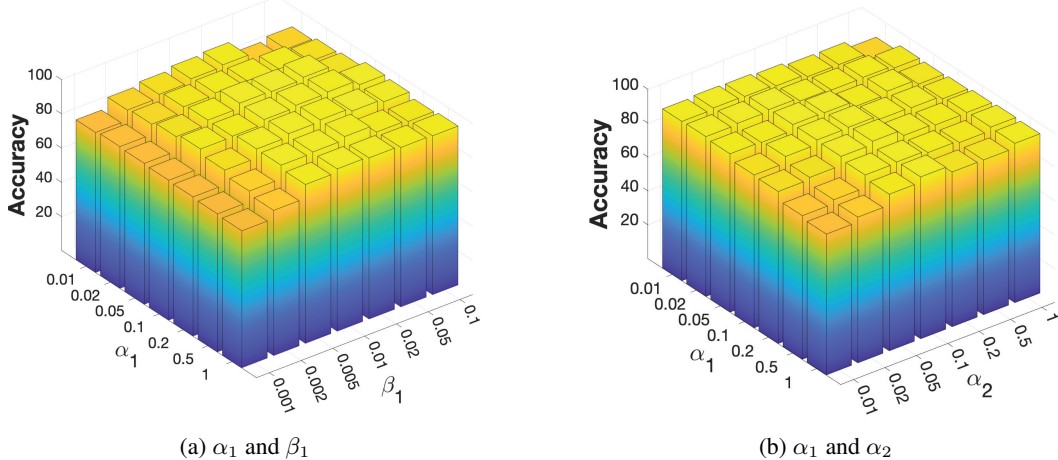

(a) $\alpha_1$ and $\beta_1$          (b) $\alpha_1$ and $\alpha_2$

Figure 7: Accuracy versus different hyper-parameters on ORHD.

It can be used to predict user behaviors, preferences, or community structures, leading to more personalized and efficient services. For instance, it could enable better content recommendation algorithms or improve user-targeted advertising. Besides, the learning framework could identify and mitigate fake news or harmful content by analyzing propagation patterns.

While these impacts are largely positive, it's important to consider potential ethical, privacy, and security challenges. Misuse of this powerful tool could lead to issues like privacy invasion, if used to infer sensitive personal information from social networks, or discrimination if the tool incorporates and perpetuates biased patterns from the training data. Also, the higher accuracy in predicting individuals' behaviors could be exploited for manipulative purposes. Therefore, it's crucial to establish robust policies and mechanisms to ensure that the use of graph-based semi-supervised learning is guided by ethical considerations and respect for privacy.