# OpenReview forum: "Optimal Block-wise Asymmetric Graph Construction for Graph-based Semi-supervised Learning"
_NeurIPS.cc/2023/Conference — NeurIPS 2023 poster_

### Official Review · Reviewer_1FFQ · 2023-06-30

**Soundness:** 3 good
**Presentation:** 3 good
**Contribution:** 3 good
**Rating:** 7
**Confidence:** 5

**Summary:**

The paper presents an optimal asymmetric graph structure for the label inference phase in graph-based semi-supervised learning (GSSL). The key motivation or intuition proposed by the authors is that we need to differentiate the roles of labeled and unlabeled nodes. Therefore, the authors design an efficient block-wise graph learning algorithm with a global convergence guarantee. The proposed method is shown to be superior to SOTA graph construction methods in GSSL through extensive experiments on synthetic and real-world datasets. The paper addresses the challenge of constructing a high-quality graph, which significantly influences label prediction performance, and proposes a solution with theoretical motivations and benefits, such as enhanced robustness to noisy node features.

**Strengths:**

1.	Quality: The work is of high quality, with rigorous theoretical motivations and comprehensive experiments. First, the motivation is strongly supported by some theoretical analysis. The proposed asymmetric optimal graph structure is deduced from Definition 1 rigorously. The authors provide a comprehensive explanation of the optimization problem and the structure of the optimal affinity graph. Second, they also present a detailed implementation of the block-wise graph learning algorithm BAGL. The paper is well-referenced, indicating a thorough understanding of the existing literature. Third, the provided convergence analysis on BAGL is also rigorous. The global sublinear convergence rate in Theorem 3 makes sense to the reviewer. Forth, the experiments are comprehensive, including several comparisons from the aspects of prediction accuracy, efficiency, and convergence rate.
2.	Clarity: The paper is well-structured and clear, with each section logically leading to the next. The authors provide clear definitions and explanations of complex concepts, making the paper accessible to readers with varying levels of expertise in the field. However, some parts of the appendix are suggested to move to the main body to give more background context on graph-based semi-supervised learning. The use of mathematical notation and diagrams further enhances the clarity of the paper.
3.	Significance: The reviewer thinks the work makes some significant contributions to the GSSL field. First, the investigated problem is significant since most GSSL literature only focuses on the label inference step. This paper focuses on the neglected graph construction step instead, as the quality of the graph affects the subsequent step a lot. Second, the proposed method achieves the SOTA global convergence rate, contributing to the GSSL field significantly.


**Weaknesses:**

There are several potential improvements in this paper.
1.	The background knowledge of the unified label inference framework can be elaborated. The details behind Eq.(1) should be provided to give readers without GSSL backgrounds more context. For example, Appendix B.2 should be added to the main body. Table 5 is very informative.

2.	The recent graph structure learning method should be discussed or compared since the goal of GSL is also similar to the task investigated in this paper.

3.	The conclusion part is short. More future work can be added.

4.	Other comments in questions section.


**Questions:**

1.	While the paper provides a detailed explanation of the block-wise graph learning algorithm, it would be beneficial to have a more explicit discussion on the computational complexity of the algorithm. How does the algorithm scale with the size of the dataset? What are the implications of using this algorithm on large-scale datasets?
2.	The paper could benefit from a discussion on potential real-world applications of the proposed method. In what specific domains or scenarios would this method be particularly useful?
3.	It would be interesting to hear the authors' perspectives on the limitations of their proposed method and how they plan to address these in future work. Are there any specific challenges or difficulties that arose during the development of the method?
4.	The paper mentions that the proposed method enhances the robustness of subsequent label inference algorithms. Could the authors elaborate on this? How does the method handle noise? Some theoretical analysis may be provided.


**Limitations:**

Please refer to the weaknesses and questions sections.

---

> ### Author Rebuttal · Authors · 2023-08-10
>
> Thank you for your very thoughtful review with constructive suggestions. We appreciate the recognition of our optimal graph construction approach in GSSL. We are glad to know that you find our paper novel, high-quality, rigorous with solid theoretical insights, well-organized with good writing, and contributing to the GSSL field. We would also like to thank you for your suggestions for improvement and have addressed each of your points below. We hope these responses will address your concerns appropriately.
>
>
> ## 1. **Background knowledge of the unified label inference framework**
>
> Thanks for your suggestion. **The relevant background knowledge of the unified label inference framework can be found in Appendix B.2**. We will consider moving some text and Table 5 into the main body for better presentation.
>
> ## 2. **Graph structure learning**
>
> More recent graph structure learning methods aim to learn a clean graph structure from the given noisy graph so that the subsequent GNNs trained on this learned clean graph can obtain better performance. In GSSL, however, there is no given graph structure, and we need to learn the graph structure based on the node features only. Therefore, it is a more challenging task compared to graph structure learning. **Therefore, we do not compare our method with other graph structure learning methods since their settings and goals are slightly different. We leave the investigation of graph structure learning for GSSL as future work since it is currently out of the scope of this work.**
>
>
> ## 3. **Short conclusion**
>
> We will elaborate on the conclusion section. More future work discussion in Appendix H will be added.
>
>
> ## 4. **Discussion on the computational complexity**
>
> Thanks for your suggestion. **Appendix G.5.1 gives a formal analysis of the time complexity, and Appendix G.5.2 presents a running time comparison with other baselines.** These results show our proposed method is efficient.
>
>
> ## 5. **A discussion on potential real-world applications**
>
> Thanks for your suggestion.  **The potential impact of the proposed method with some real-world applications, like social media and web services, can be found in Appendix I.**
>
>
> ## 6. **Limitations of the proposed method**
>
> Thanks for your suggestion. **We discuss some limitations of the proposed method with future work in Appendix H**. For instance, even though BAGL is quite efficient in terms of convergence rate, it may still have computational issues when dealing with extremely large-scale datasets with billions of samples. We plan to reduce the time spent on finishing one iteration during the optimization to solve this issue in future work.
>
>
> ## 7. **Robustness analysis of the proposed method**
>
> Thanks for your suggestion. **We include a theoretical analysis of the proposed method in terms of the robustness interpretation in Appendix D.1.** Intuitively speaking, the proposed BAGL method can guarantee that the hidden ground-truth distribution of the sample feature will be contained in an introduced ambiguity set. Even if there exists some noise in the observed features, we can still recover the ground-truth distribution as long as the number of node feature channels is sufficiently large.

---

> > ### Comment · Reviewer_1FFQ · 2023-08-16
> >
> > Thank you for your response. However, after reading the other reviews and some related works, I found that the novelty of this work is actually limited compared with some existing works. Sorry, I will lower my score to borderline reject.

---

> > > ### Author Response · Authors · 2023-08-17
> > > **Response to Reviewer 1FFQ for limited novelty concerns**
> > >
> > > We appreciate your recognition of the novelty in the first round of reviews. We also fully understand your concerns regarding the novelty after reading other reviewers' comments. We agree that the optimization algorithm for learning the graph weights is built on top of the phenomenal work [1]. However, according to NeurIPS 2023 reviewer guidelines [2], work that uses a novel combination of well-known techniques can be valuable! (Review Form -> Strengths and Weaknesses -> Originality) Therefore, we believe using the existing well-known method like [1] in our method can also have its originality based on the following new insights or contributions to the GSSL domain.
> > >
> > > First, [1] builds a symmetric graph without differentiating the labeled and unlabeled nodes, while ours builds an asymmetric graph that takes the different roles labeled and unlabeled nodes play, following the proposed optimal graph structure. This key difference between [1] and our method directly leads to superior empirical performance in ours compared to [1]. Table 2 supports this claim, and our method BAGL outperforms the method SGL used in [1] by large margins.
> > >
> > > Second, even though the formulated optimization problems in [1] and our method (when focusing on one block) are similar (Eq.(5)(6) v.s. Eq.(17) in [1]), the optimization algorithms used to solve the respective problems are totally different. [1] uses the off-the-shelf primal-dual algorithm, while ours cleverly applies the FISTA algorithm to the dual formulation of the original problem. This key difference leads to the next different properties of the two methods.
> > >
> > > Third, [1] does not even have a convergence rate guarantee, while ours enjoys the global sub-linear convergence rate guarantee, which is the SOTA result in the GSSL domain. The key difference comes from the optimization algorithm we use, which is also novel, utilizing the power of the FISTA algorithm.
> > >
> > > Fourth, we also give the interpretation of our method from the aspect of robustness (Appendix D.1) and generalization bound(Appendix D.2), while [1] does not have these theoretical interpretations. All these interpretation of our method is also novel and show the theoretical advantages of our method.
> > >
> > > To sum up, we believe that our method has novel contributions to the GSSL field even though it is partially developed based on [1].
> > >
> > > [1] Kalofolias, Vassilis. "How to learn a graph from smooth signals." Artificial intelligence and statistics. PMLR, 2016.
> > >
> > > [2] https://neurips.cc/Conferences/2023/ReviewerGuidelines
> > >
> > > **We sincerely hope that the reviewer can re-evaluate the novelty of our work based on our response.**

---

> > > > ### Comment · Reviewer_1FFQ · 2023-08-18
> > > >
> > > > Thank you for your response. Now all my concerns have been addressed.

---

> > > > > ### Author Response · Authors · 2023-08-18
> > > > >
> > > > > Thank you so much for your strong support, and we highly appreciate your updated rating score. We will add a detailed discussion on the difference between [23] and our work so that the novelty of our work is highlighted.

---

### Official Review · Reviewer_n8ra · 2023-07-03

**Soundness:** 3 good
**Presentation:** 3 good
**Contribution:** 3 good
**Rating:** 6
**Confidence:** 1

**Summary:**

The paper proposes a method for graph construction stage of graph based semi-supervised learning. They further evaluate their method with experimental results.

**Strengths:**

The authors present strong theoretical results.

**Weaknesses:**

The experimental results for the proposed method in Table 2 are only marginally better than the baselines.

**Questions:**

NA

---

> ### Author Rebuttal · Authors · 2023-08-10
>
> Thanks for your recognition of our work! We highly appreciate your feedback!
>
> **We did a statistical significance test in the experiment section**. Specifically, we perform the Friedman test with the Bonferroni-Dunn post hoc test for statistical significance analysis. Figure 2 illustrates the critical difference (CD) diagram on accuracy, where the average rank is marked along the axis with lower (better) ranks to the left. If the average rank difference between two models is greater than one CD, the relative performance is believed to be different. **Accordingly, our proposed method BAGL significantly outperforms all other baselines by a large margin.** Please refer to Figure 2 for more details.

---

### Official Review · Reviewer_HpMF · 2023-07-05

**Soundness:** 3 good
**Presentation:** 3 good
**Contribution:** 3 good
**Rating:** 6
**Confidence:** 3

**Summary:**

This paper proposes an efficient and effective method for constructing affinity graphs in Graph-based Semi-supervised Learning (GSSL), with a focus on the distinct roles of labeled and unlabeled nodes. The authors present a formulation for the GSSL problem, comprising two steps: graph construction and label inference. They investigate the optimal construction of the affinity graph in the first phase to facilitate enhanced performance in the second label inference phase. The paper offers four main contributions: a succinct definition of the optimality of the affinity graph in GSSL, a block-wise graph learning framework to infer the weights in the optimal graph structure, proof of a global sub-linear convergence rate for the proposed method, and extensive experiments on synthetic and real-world datasets to demonstrate the effectiveness and efficiency of the proposed method.

**Strengths:**

Originality:
The paper presents a novel approach to constructing affinity graphs in GSSL, with a focus on the distinct roles of labeled and unlabeled nodes. The proposed method is based on an asymmetric structure and a block-wise graph learning framework, which are different from existing methods. The paper also offers a succinct definition of the optimality of the affinity graph in GSSL, which is a unique contribution to the field. Overall, the paper is highly original in its approach to graph construction in GSSL.

Quality:
The paper is well-written and presents a rigorous derivation of the proposed method. The authors provide a clear explanation of the problem formulation and the proposed solution, as well as a detailed analysis of the benefits of the proposed method. The experiments are extensive and well-designed, with results that demonstrate the effectiveness and efficiency of the proposed method. The paper also includes a thorough review of related work, which adds to the quality of the paper. Overall, the paper is of high quality.

Clarity:
The paper is well-organized and easy to follow. The authors provide clear explanations of the concepts and methods used in the paper, and the figures and tables are well-designed and easy to understand. The paper also includes a summary of the contributions and a conclusion that summarizes the main findings. Overall, the paper is highly clear and well-presented.

Significance:
The paper makes a significant contribution to the field of GSSL by proposing an efficient and effective method for constructing affinity graphs. The proposed method is based on an asymmetric structure and a block-wise graph learning framework, which are different from existing methods. The paper also offers a succinct definition of the optimality of the affinity graph in GSSL, which is a unique contribution to the field. The experiments demonstrate the effectiveness and efficiency of the proposed method, which has the potential to improve the performance of GSSL algorithms in a wide range of applications. Overall, the paper is highly significant in its contribution to the field of GSSL.

**Weaknesses:**

1. Lack of comparison with more recent state-of-the-art methods: While the paper compares the proposed method with several existing methods, some of these methods are relatively old and may not represent the current state-of-the-art in GSSL. It would be useful to compare the proposed method with more recent methods to provide a more comprehensive evaluation.

2. Limited discussion of the limitations of the proposed method: While the paper discusses the benefits of the proposed method, there is limited discussion of its limitations. It would be useful to discuss the situations in which the proposed method may not be effective and to provide guidance on when to use the proposed method versus other methods.

3. Lack of real-world applications: While the paper includes experiments on synthetic and real-world datasets, there is limited discussion of real-world applications of the proposed method. It would be useful to provide examples of how the proposed method could be applied in real-world scenarios and to discuss the potential impact of the proposed method on these applications.

Overall, while the paper presents a novel approach to constructing affinity graphs in GSSL, there are some weaknesses that could be addressed to improve the paper's impact and relevance.

**Questions:**

Questions:
 How does the proposed method compare to more recent state-of-the-art methods in GSSL?
 Can you provide more guidance on when to use the proposed method versus other methods?
 Can you provide examples of how the proposed method could be applied in real-world scenarios?

Suggestions:
1. Consider comparing the proposed method with more recent state-of-the-art methods in GSSL to provide a more comprehensive evaluation.
2. Provide more discussion of the limitations of the proposed method and when it may not be effective.
3. Provide examples of how the proposed method could be applied in real-world scenarios to demonstrate its potential impact and relevance.
4. Consider including a discussion of the computational complexity of the proposed method and how it compares to other methods.
5. Consider including a more detailed explanation of the block-wise graph learning framework used in the proposed method to help readers better understand the approach.

**Limitations:**

The paper does not explicitly discuss the limitations and potential negative societal impact of the proposed method. While the paper does discuss the benefits of the proposed method, it does not provide a comprehensive discussion of its limitations or potential negative consequences.

It is important for authors to consider the potential limitations and negative consequences of their work, as this can help to ensure that the benefits of the work outweigh any potential negative impacts. In particular, authors should consider the ethical implications of their work and how it may impact society as a whole.

Therefore, it can be said that the authors have not adequately addressed the limitations and potential negative societal impact of their work.

---

> ### Author Rebuttal · Authors · 2023-08-10
>
>
> Thank you for your very thoughtful and constructive review. We appreciate the recognition of our optimal graph construction approach in GSSL. We are glad to know that you find our paper novel, well-written, rigorous, well-organized, and making a significant contribution to the field of GSSL. We would also like to thank you for your suggestions for improvement and have addressed each of your points below. We hope these responses will address your concerns appropriately.
>
>
>
>
> ## 1. **SOTA methods comparison**
>
> Thanks for your suggestion. **We chose some old graph construction methods in GSSL, like kNN and RBF, mainly because they are still the most simple yet quite effective methods, and they are included in relevant baseline papers as well.** The investigation of the graph construction step in GSSL has been overlooked for quite a long time, and this is why only a few recent SOTA methods are coming out in the past few years. However, **we do include GraphEBM [1], published in 2020, and BCAN [2], published in 2022, as the most recent SOTA methods as baselines.** We believe most of the recent important SOTA methods on graph construction methods in GSSL are covered in our experiments. We also welcome suggestions on other SOTA methods on graph construction steps for GSSL in recent years with references.
>
>
>
> [1] Zhijie Chen, Hongtai Cao, and Kevin Chen-Chuan Chang. Graphebm: Energy-based graph 371 construction for semi-supervised learning. In ICDM, pages 62–71. IEEE, 2020.
>
> [2] Zhen Wang, Long Zhang, Rong Wang, Feiping Nie, and Xuelong Li. Semi-supervised learning 445 via bipartite graph construction with adaptive neighbors. IEEE Transactions on Knowledge and 446 Data Engineering, pages 1–1, 2022.
>
>
>
>
> ## 2. **Discussion of the limitations**
>
> Thanks for your suggestion. **We actually include the discussions of the limitations in Appendix H.** For instance, one limitation of BAGL is it is only suitable for the transductive setting, and it may still have computational issues when dealing with extremely large-scale datasets with billions of samples. We leave the investigation of these issues as future work since it is out of the scope of this paper. We will add more discussion of limitations in the main body.
>
>
>
>
> ## 3. **Lack of real-world applications**
>
>
> Thanks for your suggestion. In fact, **we indeed run the experiments on some open real-world datasets, and the task is image classification. We assume this is an example of a real-world application.** Please refer to Appendix F.1 for the details of the real-world datasets. The experimental results show how the proposed method could be applied in this real-world scenario. **The potential impact of the proposed method can be found in Appendix I, with a focus on social benefits**.
>
>
>
> ## 4. **A discussion of the computational complexity**
>
> Thanks for your suggestion. **Appendix G.5.1 gives a formal analysis of the time complexity, and Appendix G.5.2 presents a running time comparison with other baselines.** These results show our proposed method is efficient.
>
>
> ## 5. **A more detailed explanation of the proposed method**
>
> Thanks for your suggestion. For the block-wise graph learning framework,  we build on top of the well-known method [1]. We learn each block in the optimal graph structure via a similar method in [1] but with quite different optimization algorithms. We apply the FISTA algorithm to the dual formulation of the proposed learning framework. **We will polish the text to make the paper easier to understand.**
>
>
> [1] Kalofolias, Vassilis. "How to learn a graph from smooth signals." Artificial intelligence and statistics. PMLR, 2016.

---

### Official Review · Reviewer_CLLx · 2023-07-05

**Soundness:** 2 fair
**Presentation:** 2 fair
**Contribution:** 3 good
**Rating:** 4
**Confidence:** 3

**Summary:**

This paper proposes a novel methodology for graph-based semi-supervised learning by leveraging a asymetric graph construction technique. The main contribution of the paper is the design of a block-wise graph learning framework to estimate the weights of a graph.

**Strengths:**

The main strengths of the paper are:
- The derivation of the structure of the optimal affinity graph
- The derivation of an optimization algorithm for the implementation of the block-wise graph learning algorithm
- The thorough experimental evaluation to assess the performance of the proposed algorithm in different scenarios


**Weaknesses:**

- While the derivation of the optimal affinity graph is novel, the optimization algorithm for learning the graph weights are heavily influenced by prior works, and therefore not much novel.

- The plots (c) and (d) in Figure 3 are not as helpful as the x-axis represent number of iterations where instead the authors should use computational time. Therefore, from Figures 3c and 3d we cannot conclude much about the computational efficiency of the proposed algorithm.

**Questions:**

- I am not convinced regarding the novelty of the framework. Can the authors elaborate how the development of section 3.2 is different from the work of [23] apart from the well-known application of the FISTA step?

**Limitations:**

The authors properly addressed the limitations of their proposed framework.

---

> ### Author Rebuttal · Authors · 2023-08-09
>
>
>
> Thanks for your detailed review. We appreciate the recognition of our derivation of the optimal affinity graph with thorough experimental evaluation. We would like to thank you for your suggestions for improvement and have addressed each of your points below. We hope these responses will address your concerns appropriately.
>
>
>
> ## 1. **Novelty**
>
> We appreciate your recognition of the novelty in the derivation of the optimal affinity graph in our method, confirmed by other reviewers as well. We also fully understand your concerns regarding the novelty of the optimization part or implementation of the proposed BAGL algorithm. **We agree that the optimization algorithm for learning the graph weights is built on top of the phenomenal work [1]. However, according to NeurIPS 2023 reviewer guidelines [2], work that uses a novel combination of well-known techniques can be valuable! (Review Form -> Strengths and Weaknesses -> Originality) Therefore, we believe using the existing well-known method like [1] in our method can also have its originality based on the following new insights or contributions to the GSSL domain.**
>
>
> First, **[1] builds a symmetric graph without differentiating the labeled and unlabeled nodes, while ours builds an asymmetric graph that takes the different roles labeled and unlabeled nodes play, following the proposed optimal graph structure.** This key difference between [1] and our method directly leads to superior empirical performance in ours compared to [1]. Table 2 supports this claim, and **our method BAGL outperforms the method SGL used in [1] by large margins.**
>
> Second, even though the formulated optimization problems in [1] and our method (when focusing on one block) are similar (Eq.(5)(6) v.s. Eq.(17) in [1]), **the optimization algorithms used to solve the respective problems are totally different.** [1] uses the off-the-shelf primal-dual algorithm, while ours cleverly applies the FISTA algorithm to the dual formulation of the original problem. This key difference leads to the next different properties of the two methods.
>
> Third, **[1] does not even have a convergence rate guarantee, while ours enjoys the global sub-linear convergence rate guarantee, which is the SOTA result in the GSSL domain.** The key difference comes from the optimization algorithm we use, which is also novel, utilizing the power of the FISTA algorithm.
>
> Fourth, **we also give the interpretation of our method from the aspect of robustness (Appendix D.1) and generalization bound(Appendix D.2), while [1] does not have these theoretical interpretations.** All these interpretation of our method is also novel and show the theoretical advantages of our method.
>
> To sum up, we believe that our method has novel contributions to the GSSL field even though it is partially developed based on [1].
>
> [1] Kalofolias, Vassilis. "How to learn a graph from smooth signals." Artificial intelligence and statistics. PMLR, 2016.
>
> [2] https://neurips.cc/Conferences/2023/ReviewerGuidelines
>
>
>
> Further, **other reviewers also appreciate the novelty of our work**. Reviewer HpMF says, "The paper presents a novel approach to constructing affinity graphs in GSSL, with a focus on the distinct roles of labeled and unlabeled nodes."  Reviewer 1FFQ comments, "Even though this method is based on the well-known FISTA optimization algorithm, it is applied to the dual problem, which is also new since the convergence rate will be affected compared to the one applied to the primal problem directly."
>
>
> **We sincerely hope that the reviewer can re-evaluate the novelty of our work based on our response.**
>
>
>
> ## 2. **Regarding Figure 3**
>
> We understand your concerns related to Figure 3. In fact, **Figure 3 is not for the computational efficiency comparison of different methods but for the convergence rate comparison of different methods. We do not cover computational efficiency in Sec. 4.3.2, and we only focus on the convergence rate here.**
>
> In fact, when we compare the convergence rates of different algorithms in the optimization domain, we usually set the x-axis as the number of iterations, instead of the actual running time. In this way, we can easily spot how quickly each algorithm approaches its limits. One of the most direct ways to measure this is by observing how the solution improves (e.g., how the $l_2$ distance between the current solution and the limit solution decreases) with each successive iteration. Please refer to the axes of Figure 3(c)(d) for details. In this way, **setting the x-axis as the number of iterations can better align with the definition of convergence rate**. It's easy to see that fewer iterations to achieve a similar error is generally better.
>
>
> However, it's important to note that the computational cost of one iteration can differ dramatically between algorithms. Therefore, it may not always be a direct measure of the computational effort. Therefore, your suggestion for computational time comparison is insightful.
>
>
> In fact, **we have already included a computational efficiency comparison in Appendix G.5.2** titled  "Running time comparison." Now in this section, we indeed compare the actual running time of each method. **Table 7 shows that our method is quite efficient compared with other optimization-based graph construction methods**.
>
>
> ## 3. **Final remarks**
>
> We sincerely respect and appreciate the time and effort you have dedicated to reviewing our work. We understand that every work should undergo meticulous scrutiny to ensure the highest standards, and we truly value your feedback. However, **we humbly hope you can reconsider certain aspects of our work that we believe it possesses significant novelty and includes computational time comparison in Appendix.** If there are specific areas of ambiguity or contention, we are willing to address them with further clarification. We kindly ask for an opportunity to emphasize the potential impact our work can bring to the broader this community.

---

> > ### Comment · Reviewer_CLLx · 2023-08-17
> >
> > Thanks for the rebuttal. First of all, the question I asked was straightforward and I was expecting a direct-to-the-point answer rather than a lengthy essay with bold-face quotations all over. Based on that and on other’s reviewers views about how the current paper falls short to clearly explain how’s it different from [23], I’ll maintain my score as is.

---

> > > ### Author Response · Authors · 2023-08-18
> > >
> > > Thanks for your suggestion! We format our response into a direct-to-the-point answer without bold text.
> > >
> > > ### 1. Novelty
> > > We list the major differences between our work and [23], showing the novelty and contribution of our work.  We will add a detailed discussion of their differences.
> > >
> > > |                                          |           Ours          |          [23]         |
> > > |:----------------------------------------:|:-----------------------:|:---------------------:|
> > > |              Graph structure             |     Asymmetric graph    |    Symmetric graph    |
> > > |             Label information            |           Use           |        Not use        |
> > > | Optimal for label inference step in GSSL |           Yes           |           No          |
> > > |          Optimization algorithm          |  FISTA on dual problem  | Primal-dual algorithm |
> > > |             Convergence rate             |   Globally sub-linear   |      No guarantee     |
> > > |       Time complexity per iteration      | $O(N_u \times N_l + N)$ |    $O(N \times N)$    |
> > > |      Generalization bond improvement     |           Yes           |           No          |
> > >
> > > Besides, NeurIPS 2023 reviewer guidelines indicate work that uses a novel combination of well-known techniques can be valuable.
> > >
> > > ### 2. Figure 3
> > >
> > > Figure 3 is for convergence rate comparison, so we set the x-axis as the number of iterations. The running time comparison can be found in Appendix G.5.2.
> > >
> > > ### 3. Final remarks
> > >
> > > We sincerely hope the reviewer can re-evaluate the novelty of our work based on our compact response and the latest reviews from other reviewers!

---

> ### Comment · Area_Chair_YrbC · 2023-08-17
> **Please provide additional feedback**
>
> Hi,
>
> Could you please acknowledge that you have read the rebuttal and let the reviewers know if you still have any concerns or not?

---

### Official Review · Reviewer_7nqE · 2023-07-07

**Soundness:** 4 excellent
**Presentation:** 4 excellent
**Contribution:** 3 good
**Rating:** 7
**Confidence:** 4

**Summary:**

The authors proposes to solve graph-based semi-supervised learning (GSSL) problems by first finding the "optimal graph" for SSL. The optimal graph has edges only from labeled to unlabeled nodes, or between unlabeled nodes. These edge weights are computed through FISTA algorithm in the dual space, and theoretical guarantees are provided for sub-linear convergence rates. Experiments are conducted using both synthetic and real-world datasets.

**Strengths:**

This paper is very well written and nicely presented. The clear writing made it easy to follow.

I did not carefully read through the proofs in the appendix, but the motivation for the framework and derivation of the algorithm seem correct. I appreciate that this paper did solid work on all aspects - problem formulation, clever optimization algorithm, theoretical convergence analysis, and numerical experiments.

**Weaknesses:**

Please see the questions below.

**Questions:**

1. When comparing methods such as RBF and KNN to BAGL, were RBF and KNN used to also generate asymmetric (and not symmetric) graphs? This is not made immediately clear in the text, and I'm wondering if the superior performance mainly comes from keeping $W_{lu}, W_{ll}$ all zero matrices.

2. Figure 1 c is visually appealing, but I'm having trouble understanding why that solution is any better or more probable than the RBF solution in 1b.

3. There's a mismatch between appendix section labels and their references in the main text. This should be fixed since the appendix includes important time complexity and runtime analysis.

4. How does BAGL react to class imbalance, either in the labeled nodes or in the entire classification problem?

**Limitations:**

Yes, in the appendix.

---

> ### Author Rebuttal · Authors · 2023-08-09
>
> Thank you for your very thoughtful and constructive review. We appreciate the recognition of our optimal graph construction approach in GSSL. We are glad to know that you find our paper solid in all aspects - problem formulation, clever optimization algorithm, theoretical convergence analysis, and numerical experiments. We would also like to thank you for your suggestions for improvement and have addressed each of your points below. We hope these responses will address your concerns appropriately.
>
>
> ## 1.  Keeping $W_{lu}$ and $W_{ll}$ zero matrices.
>
> This suggestion is quite interesting and valuable! In fact, when comparing existing graph construction methods like kNN or RBF with our proposed method BAGL (Table 2), we still stick to the original symmetric graph structure in these baselines and do not change the constructed graphs in these baselines into the same optimal asymmetric graph structure as the one used in our proposed method BAGL. **We choose not to do so, mainly because we strictly follow the original graph structures used in these baselines, which are all symmetric graphs, for a fair comparison with our method.** Regarding the question of whether the superior performance mainly comes from keeping $W_{lu}, W_{ll}$ all zero matrices, **we actually did an ablation study to investigate the influence of keeping $W_{lu}, W_{ll}$ all zero matrices for the proposed BAGL method in Table 3 in Sec. 4.3.3**. If we remove the constraints of $W_{lu} = \mathbf{O}, W_{ll} = \mathbf{O}$, we find that there is a most significant performance drop compared with other variants of BAGL in Table 3. **This result shows that the proposed optimal asymmetric graph structure contributes most to the success of BAGL.** However, we agree that this proposed asymmetric optimal graph structure can be easily incorporated into existing graph construction methods like kNN and RBF. Therefore, we add another experiment where we also convert the graphs constructed by existing baselines into the same proposed asymmetric optimal graph structure via setting $W_{lu} = \mathbf{O}, W_{ll} = \mathbf{O}$. The results are as follows.
>
> |                                  |  RBF  |  kNN  |  SGL  | RGCLI |  AGR  | GraphEBM |  BAGL |
> |:--------------------------------:|:-----:|:-----:|:-----:|:-----:|:-----:|:--------:|:-----:|
> | w/o $W_{lu},W_{ll} = \mathbf{O}$ | 97.46 | 86.59 | 94.68 | 88.24 | 97.63 |   95.13  | 95.71 |
> |  w/ $W_{lu},W_{ll} = \mathbf{O}$ | 97.51 | 87.66 | 96.72 | 89.31 | 95.40 |   95.21  | 97.88 |
> |          improvement (%)         |  0.05 |  1.23 |  2.15 |  1.21 | -2.28 |   0.08   |  2.26 |
>
> We can have some observations from this table. The proposed optimal asymmetric graph structure via setting $W_{lu},W_{ll} = \mathbf{O}$ does have some positive effects on almost all graph construction methods in GSSL. These empirical findings also support the theoretical motivations in Sec 3.1.2. Also, some optimization-based methods like SGL and BAGL seem to benefit more from this optimal asymmetric graph structure compared to some other baselines like kNN. Maybe this is because these methods tend to handle the optimization problem in Eq.(3) from the optimization perspective directly or indirectly, aligned with the merits of the derivation of why we set $W_{lu},W_{ll} = \mathbf{O}$ in Proposition 1. Besides, we have to say that this optimal asymmetric is not a universal positive strategy for all graph construction baselines. It has negative effects on AGR. We suspect setting $W_{lu},W_{ll} = \mathbf{O}$ may break the anchor node connections in AGR, thus leading to sub-optimal performance. **In summary, even though the proposed optimal structure is quite simple and may be incorporated into many existing graph construction baselines, we can only get the most significant performance improvement when used in optimization-based methods like our proposed method BAGL.**
>
>
>
> ## 2. Figure 1.
>
> A simple but not quite accurate intuition for the superiority of BAGL over RBF (Figure 1) is as follows. BAGL inexplicitly uses the label information via the optimal graph structure (setting $W_{lu},W_{ll} = \mathbf{O}$). Therefore, it is easier for BAGL to learn that the given red and the green labeled point (Figure 1a) are from two different classes or clusters (one is a ring-like cluster, and the other is a dense one). But RBF is only based on the distance of the points without any label information; thus, its performance is worse.
>
>
> ## 3. Mismatch between main and appendix.
>
> Thanks for spotting this mismatch. We will fix this issue.
>
>
> ## 4. Class imbalance
>
> We handcraft a label-imbalanced version of the ORHD dataset, where the labeled nodes are sampled until the overall imbalance ratio (max(#labeled class nodes)/ min(#labeled class nodes)) reaches 20. The results of BAGL with different subsequent label inference algorithms are as follows.
>
> |                          |  GRF  |  LGC  |  GCN  |
> |:------------------------:|:-----:|:-----:|:-----:|
> |  Balanced dataset (Acc)  | 97.88 | 98.04 | 98.15 |
> | Imbalanced dataset (Acc) | 94.75 | 95.30 | 96.28 |
> |   Performance Drop (%)   |  3.19 |  2.79 |  1.90 |
>
> **We can see that the imbalance in label classes can lead to a great performance drop in BAGL. Because BAGL only uses the information of whether the node is labeled or not in the graph construction step, instead of the information of the exact label of the node. This makes the BAGL unaware of the labeled class imbalance issue during training.** We will add this limitation of BAGL. However, from the results, we can see that the subsequent label inference algorithms also react to the labeled class imbalance case differently (say, GCN is more robust.) Therefore, we leave the investigation of graph construction methods that are also robust to label class imbalance as future work since it is out of the scope of this work. Thanks for your insights on this future direction. We will continue to conduct research on this new problem setting.

---

> > ### Comment · Reviewer_7nqE · 2023-08-17
> >
> > Thank you for the additional experimental results. Overall, I am happy with the paper and the authors' rebuttal comments.
> > However, it seems that the details in the appendix is *absolutely necessary* to address many of the reviewers' and future readers' concerns; in fact, I had some of those myself before I looked through the appendix. I'll keep my original score, but it is possible that a journal that allows for more thorough and longer manuscripts would be a better fit for this paper. Based on the reviewers' comments, at least a more detailed discussion on why this proposed work is different from [23], along with discussion of the proposed method's limitations, should be added to the main text.

---

> > > ### Author Response · Authors · 2023-08-17
> > >
> > > Thanks again for your strong support and insightful suggestions for our work! We will move some contents from the appendix to the main body so that the readers will have a better understanding of our work. More importantly, we will add a detailed discussion on the difference between [23] and our work, along with the limitations of our work. We will definitely consider extending this work to a journal as your valuable suggestions!

---

### Comment · Area_Chair_YrbC · 2023-08-17
**Comment by the AC**

Dear Authors and Reviewers

Thank you for your detailed answers and the effort that you put in your rebuttal.

Could the reviewers who haven't responded to the authors please do this as soon as possible.

---

### Decision · Program_Chairs · 2023-09-21

**Decision:**

Accept (poster)

**Comment:**

The majority of the reviewers recommend accepting the paper with average score 6.
I recommend accepting the paper as a poster.